# Variational Reasoning for Language Models

**Xiangxin Zhou**[*,1,2,3], **Zichen Liu**[1,4], **Haonan Wang**[1,4], **Chao Du**[1], **Min Lin**[1],
**Chongxuan Li**[5], **Liang Wang**[2,3], **Tianyu Pang**[*,‡,1]
[1]Sea AI Lab   [2]UCAS   [3]CASIA   [4]NUS   [5]RUC   ⌂ **Code Link**

## Abstract

We introduce a **variational reasoning** framework for language models that treats thinking traces as latent variables and optimizes them through variational inference. Starting from the evidence lower bound (ELBO), we extend it to a multi-trace objective for tighter bounds and propose a forward-KL formulation that stabilizes the training of the variational posterior. We further show that rejection sampling finetuning and binary-reward RL, including GRPO, can be interpreted as local forward-KL objectives, where *an implicit weighting by model accuracy* naturally arises from the derivation and reveals a previously unnoticed bias toward easier questions. We empirically validate our method on the Qwen 2.5 and Qwen 3 model families across a wide range of reasoning tasks. Overall, our work provides a principled probabilistic perspective that unifies variational inference with RL-style methods and yields stable objectives for improving the reasoning ability of language models.

## 1 Introduction

Reasoning has recently become a central focus for large language models (LLMs), driving advances in tasks such as mathematics, coding, and scientific problem solving (Jaech et al., 2024; Comanici et al., 2025; Guo et al., 2025). A common strategy is to let models generate explicit thinking traces before producing final answers. To train such reasoning abilities, two dominant approaches are widely used: supervised finetuning (SFT) (Guha et al., 2025; Muennighoff et al., 2025) and reinforcement learning (RL) (Yu et al., 2025a; Liu et al., 2025; Zeng et al., 2025), both showing strong empirical success.

Despite this progress, each approach faces limitations. SFT often relies on curated long-thinking traces, which are costly to collect and, as an offline method, may struggle to generalize (Chu et al., 2025) or suffer from catastrophic forgetting (Shenfeld et al., 2025). Recent RL methods typically depend on verifiable rewards to mitigate reward hacking, yet training can be unstable and output diversity may collapse (Cheng et al., 2025; Cui et al., 2025b). As a result, correct answers to harder questions become increasingly rare, leading to lower Pass@K accuracy than base models (Yue et al., 2025a). These challenges motivate the search for a more principled objective for training reasoning models.

To this end, we propose to view reasoning through the lens of probabilistic modeling, where thinking traces are treated as *latent variables*. Variational inference (Kingma & Welling, 2013) provides a natural way to optimize the log-likelihood of producing correct answers. This perspective offers several advantages: it replaces the intractable marginalization over thinking traces with tractable lower bounds, enables multi-trace extensions that tighten the objective, and introduces a variational posterior that can sample thinking paths more likely to yield correct answers. In this way, it provides a principled objective for training reasoning models, while remaining compatible with verifiable rewards.

Building on this perspective, we develop a **variational reasoning** framework for language models in Section 2. The core idea is to decompose reasoning into a thinking trace and an answer, leading to the *maximum log-likelihood estimation (MLE)* objective. To make this optimization tractable, we introduce an evidence lower bound (ELBO) and extend it to an IWAE-style multi-trace formulation (Burda et al., 2015), which tightens with more rollouts. To further stabilize the training of the variational posterior, we propose a forward-KL objective that prevents collapse and makes better use of answer hints. Together, these components form a unified training pipeline (as shown in Algorithm 1) that jointly improves the reasoning model and the variational posterior.

Beyond the method itself, our framework also helps interpret existing approaches, as described in Section 3. Rejection-sampling finetuning (RFT) (Dong et al., 2023; Touvron et al., 2023) can

---

*Equal contribution. ‡Correspondence to Tianyu Pang.

be re-expressed as forward-KL optimization weighted by model accuracy, and binary-reward RL, including GRPO (Shao et al., 2024), admits a similar form. Our analysis shows that this **weighting by accuracy** arises implicitly and produces a systematic bias toward easier questions, an effect that has not been explicitly recognized before. By placing these methods under a shared probabilistic view, our framework provides principled objectives and clarifies the behavior of widely used methods.

We validate our framework on the Qwen2.5 and Qwen3 model families (Yang et al., 2024; Team, 2025b) and observe consistent improvements over strong baselines across diverse reasoning benchmarks, including MATH500, AIME24&25, OlympiadBench, LiveCodeBench, GPQA-Diamond, and MMLU-Pro. Due to space constraints, a detailed discussion of related work is deferred to Appendix B.

## 2 VARIATIONAL REASONING

Let $\mathcal{V}^*$ denote the set of all prompt strings over the vocabulary $\mathcal{V}$. Given an input question $\boldsymbol{x} \in \mathcal{V}^*$, a reasoning model $\pi_\theta(\boldsymbol{z}, \boldsymbol{y}|\boldsymbol{x})$ generates both a thinking process $\boldsymbol{z} \in \mathcal{V}^*$ and a predicted answer $\boldsymbol{y} \in \mathcal{V}^*$. The joint probability can be written as $\pi_\theta(\boldsymbol{z}, \boldsymbol{y}|\boldsymbol{x}) = \pi_\theta(\boldsymbol{y}|\boldsymbol{x}, \boldsymbol{z}) \cdot \pi_\theta(\boldsymbol{z}|\boldsymbol{x})$. Following a standard format template (Guo et al., 2025), these two conditional terms are computed as[1]

$$\pi_\theta(\boldsymbol{z}|\boldsymbol{x}) = \pi_\theta\big([\boldsymbol{z}, \texttt{</think>}] \,|\, [\boldsymbol{x}, \texttt{<think>}]\big);$$

$$\pi_\theta(\boldsymbol{y}|\boldsymbol{x}, \boldsymbol{z}) = \pi_\theta\big([\boldsymbol{y}, \texttt{</answer>}] \,|\, [\boldsymbol{x}, \texttt{<think>}, \boldsymbol{z}, \texttt{</think>}, \texttt{<answer>}]\big),$$

(1)

where $\texttt{</think>}$ and $\texttt{</answer>}$ serve as the end-of-sequence markers for $\boldsymbol{z}$ and $\boldsymbol{y}$, respectively. We define the marginal distribution $P_\theta(\boldsymbol{y}|\boldsymbol{x}) = \sum_{\boldsymbol{z}} \pi_\theta(\boldsymbol{z}, \boldsymbol{y}|\boldsymbol{x}) = \sum_{\boldsymbol{z}} \pi_\theta(\boldsymbol{y}|\boldsymbol{x}, \boldsymbol{z})\pi_\theta(\boldsymbol{z}|\boldsymbol{x})$, where the notation $P_\theta(\boldsymbol{y}|\boldsymbol{x})$ highlights that this distribution is *induced* by $\pi_\theta$. This is different from the non-thinking probability $\pi_\theta(\boldsymbol{y}|\boldsymbol{x})$, which does not marginalize over possible thinking traces.

### 2.1 EVIDENCE LOWER BOUND

Let $\mathcal{Y}_{\boldsymbol{x}} \subset \mathcal{V}^*$ denote the oracle set (possibly infinite) of correct answers to the question $\boldsymbol{x}$. The marginal probability that $\pi_\theta$ generates a correct answer is $P_\theta(\mathcal{Y}_{\boldsymbol{x}}|\boldsymbol{x}) = \sum_{\boldsymbol{y} \in \mathcal{Y}_{\boldsymbol{x}}} P_\theta(\boldsymbol{y}|\boldsymbol{x})$. Maximizing this probability gives the *maximum log-likelihood estimation (MLE)* objective: $\boxed{\max_\theta \log P_\theta(\mathcal{Y}_{\boldsymbol{x}}|\boldsymbol{x})}$. However, this MLE objective is intractable because computing $P_\theta(\boldsymbol{y}|\boldsymbol{x})$ requires summing over all possible thinking traces $\boldsymbol{z}$. To make learning feasible, we apply variational inference (Kingma & Welling, 2013) to derive an *evidence lower bound (ELBO)*:

$$\begin{aligned}
\log P_\theta(\mathcal{Y}_{\boldsymbol{x}}|\boldsymbol{x}) &= \log \sum_{\boldsymbol{z}} \pi_\theta(\mathcal{Y}_{\boldsymbol{x}}|\boldsymbol{x}, \boldsymbol{z})\pi_\theta(\boldsymbol{z}|\boldsymbol{x}) \\
&= \log \mathbb{E}_{q_\phi(\boldsymbol{y}')}\mathbb{E}_{q_\phi(\boldsymbol{z}|\boldsymbol{x}, \boldsymbol{y}')}\left[\frac{\pi_\theta(\mathcal{Y}_{\boldsymbol{x}}|\boldsymbol{x}, \boldsymbol{z})\pi_\theta(\boldsymbol{z}|\boldsymbol{x})}{q_\phi(\boldsymbol{z}|\boldsymbol{x}, \boldsymbol{y}')}\right] \\
&\geq \mathbb{E}_{q_\phi(\boldsymbol{y}')}\underbrace{\left[\mathbb{E}_{q_\phi(\boldsymbol{z}|\boldsymbol{x}, \boldsymbol{y}')}\left[\log \pi_\theta(\mathcal{Y}_{\boldsymbol{x}}|\boldsymbol{x}, \boldsymbol{z})\right] - \mathbb{D}_{\mathrm{KL}}\left(q_\phi(\boldsymbol{z}|\boldsymbol{x}, \boldsymbol{y}')||\pi_\theta(\boldsymbol{z}|\boldsymbol{x})\right)\right]}_{\boxed{\mathcal{L}_{\mathrm{ELBO}}(\boldsymbol{x}, \mathcal{Y}_{\boldsymbol{x}}, \boldsymbol{y}'; \pi_\theta, q_\phi)}}.
\end{aligned}$$

(2)

In this expression, $\pi_\theta(\mathcal{Y}_{\boldsymbol{x}}|\boldsymbol{x}, \boldsymbol{z})$ denotes the probability of producing a correct answer given the question $\boldsymbol{x}$ and a particular thinking trace $\boldsymbol{z}$. The distribution $q_\phi(\boldsymbol{z}|\boldsymbol{x}, \boldsymbol{y}')$ is the **variational posterior**, which conditions not only on the question $\boldsymbol{x}$ but also on an auxiliary **answer hint** $\boldsymbol{y}'$:

$$q_\phi(\boldsymbol{z}|\boldsymbol{x}, \boldsymbol{y}') = q_\phi\big([\boldsymbol{z}, \texttt{</think>}] \,|\, [\boldsymbol{x}, \texttt{<hint>}, \boldsymbol{y}', \texttt{</hint>}, \texttt{<think>}]\big). \quad (3)$$

Here, $\texttt{<hint>}$ and $\texttt{</hint>}$ are shown as example delimiters; in experiments, we ablate different special tokens to wrap the hint $\boldsymbol{y}'$ and concatenate it after $\boldsymbol{x}$. Conditioning on $\boldsymbol{y}'$ encourages the variational posterior to generate thinking traces $\boldsymbol{z}$ that are more likely to yield correct answers. A simple yet effective design choice is to let $\boldsymbol{y}'$ come directly from the oracle set, that is, $\mathrm{supp}[q_\phi(\boldsymbol{y}')] \subset \mathcal{Y}_{\boldsymbol{x}}$. In practice, $\boldsymbol{y}'$ may be a rephrasing of a reference answer or any correct expression sampled from $\mathcal{Y}_{\boldsymbol{x}}$.

We can further show (detailed in Appendix A.1) that maximizing the ELBO objective w.r.t. $q_\phi$ in Eq. (2) is equivalent to minimizing the *reverse* KL divergence between $q_\phi(\boldsymbol{z}|\boldsymbol{x}, \boldsymbol{y}')$ and $P_\theta(\boldsymbol{z}|\boldsymbol{x}, \mathcal{Y}_{\boldsymbol{x}})$:

$$\mathcal{L}_{\mathrm{ELBO}}(\boldsymbol{x}, \mathcal{Y}_{\boldsymbol{x}}, \boldsymbol{y}'; \pi_\theta, q_\phi) = \log P_\theta(\mathcal{Y}_{\boldsymbol{x}}|\boldsymbol{x}) - \mathbb{D}_{\mathrm{KL}}\left(q_\phi(\boldsymbol{z}|\boldsymbol{x}, \boldsymbol{y}')||P_\theta(\boldsymbol{z}|\boldsymbol{x}, \mathcal{Y}_{\boldsymbol{x}})\right). \quad (4)$$

Here $\boxed{P_\theta(\boldsymbol{z}|\boldsymbol{x}, \mathcal{Y}_{\boldsymbol{x}}) = \frac{\pi_\theta(\mathcal{Y}_{\boldsymbol{x}}|\boldsymbol{x}, \boldsymbol{z})\pi_\theta(\boldsymbol{z}|\boldsymbol{x})}{P_\theta(\mathcal{Y}_{\boldsymbol{x}}|\boldsymbol{x})}}$ is the **true posterior**. Compared with the prior distribution $\pi_\theta(\boldsymbol{z}|\boldsymbol{x})$, this posterior distribution re-weights thinking traces by $\pi_\theta(\mathcal{Y}_{\boldsymbol{x}}|\boldsymbol{x}, \boldsymbol{z})$, thus favoring $\boldsymbol{z}$ that are more likely to produce correct answers. According to Eq. (4), we know that the optimal solution for $\max_{q_\phi} \mathcal{L}_{\mathrm{ELBO}}(\boldsymbol{x}, \mathcal{Y}_{\boldsymbol{x}}, \boldsymbol{y}'; \pi_\theta, q_\phi)$ is: $\boxed{\forall \boldsymbol{y}' \sim q_\phi(\boldsymbol{y}'), \text{ there is } q_\phi^*(\boldsymbol{z}|\boldsymbol{x}, \boldsymbol{y}') = P_\theta(\boldsymbol{z}|\boldsymbol{x}, \mathcal{Y}_{\boldsymbol{x}})}$.

---

[1]We will omit special tokens such as $\texttt{</think>}$ and $\texttt{</answer>}$ in the formulas without ambiguity.

## 2.2 Extension to IWAE-style lower bound

In reinforcement learning (RL), it is now common practice to perform parallel rollouts of multiple thinking traces $\boldsymbol{z}$ and answers $\boldsymbol{y}$ for a given question $\boldsymbol{x}$ (Shao et al., 2024). This naturally motivates us to extend the single-trace ELBO in Eq. (4) to an importance-weighted autoencoder (IWAE) style bound (Burda et al., 2015). By leveraging multiple $K$ traces, this approach yields a strictly tighter lower bound. Specifically, we obtain the following *IWAE-style lower bound* for $\log P_\theta(\mathcal{Y}_{\boldsymbol{x}}|\boldsymbol{x})$:

$$\mathcal{L}_{\text{ELBO}}^K(\boldsymbol{x}, \mathcal{Y}_{\boldsymbol{x}}, \boldsymbol{y}'; \pi_\theta, q_\phi) = \mathbb{E}_{\boldsymbol{z}_{1:K} \sim q_\phi(\boldsymbol{z}|\boldsymbol{x}, \boldsymbol{y}')} \left[ \log \frac{1}{K} \sum_{k=1}^K \frac{\pi_\theta(\boldsymbol{z}_k, \mathcal{Y}_{\boldsymbol{x}}|\boldsymbol{x})}{q_\phi(\boldsymbol{z}_k|\boldsymbol{x}, \boldsymbol{y}')} \right]. \tag{5}$$

These IWAE-style bounds satisfy $\mathcal{L}_{\text{ELBO}}^K \leq \mathcal{L}_{\text{ELBO}}^{K+1} \leq \log P_\theta(\mathcal{Y}_{\boldsymbol{x}}|\boldsymbol{x})$ for any $K \in \mathbb{N}^+$, which means the bound becomes tighter as $K$ increases (the proof is similar to that of Burda et al. (2015)). The single-trace ELBO objective in Eq. (4) corresponds to the special case of $K = 1$, i.e., $\mathcal{L}_{\text{ELBO}} = \mathcal{L}_{\text{ELBO}}^1$.

**Gradient estimation.** We now derive the gradient of $\mathcal{L}_{\text{ELBO}}^K(\boldsymbol{x}, \mathcal{Y}_{\boldsymbol{x}}, \boldsymbol{y}'; \pi_\theta, q_\phi)$ w.r.t. the model parameters $\theta$ (see Appendix A.2 for the gradient w.r.t. the variational parameters $\phi$, i.e., $\nabla_\phi \mathcal{L}_{\text{ELBO}}^K$):

$$\nabla_\theta \mathcal{L}_{\text{ELBO}}^K(\boldsymbol{x}, \mathcal{Y}_{\boldsymbol{x}}, \boldsymbol{y}'; \pi_\theta, q_\phi) = \mathbb{E}_{\boldsymbol{z}_{1:K} \sim q_\phi(\boldsymbol{z}|\boldsymbol{x}, \boldsymbol{y}')} \left[ \sum_{k=1}^K \widetilde{\rho}_k \nabla_\theta \log \pi_\theta(\boldsymbol{z}_k, \mathcal{Y}_{\boldsymbol{x}}|\boldsymbol{x}) \right],$$

$$\text{where} \quad \widetilde{\rho}_k = \frac{\rho_k}{\sum_{j=1}^K \rho_j} \quad \text{and} \quad \rho_k = \frac{\pi_\theta(\boldsymbol{z}_k, \mathcal{Y}_{\boldsymbol{x}}|\boldsymbol{x})}{q_\phi(\boldsymbol{z}_k|\boldsymbol{x}, \boldsymbol{y}')}. \tag{6}$$

**Estimating $\rho_k$.** The weight $\rho_k$ in Eq. (6) can be decomposed as $\boxed{\rho_k = \frac{\pi_\theta(\boldsymbol{z}_k|\boldsymbol{x})}{q_\phi(\boldsymbol{z}_k|\boldsymbol{x}, \boldsymbol{y}')} \cdot \pi_\theta(\mathcal{Y}_{\boldsymbol{x}}|\boldsymbol{x}, \boldsymbol{z}_k)}$, where the first term, $\frac{\pi_\theta(\boldsymbol{z}_k|\boldsymbol{x})}{q_\phi(\boldsymbol{z}_k|\boldsymbol{x}, \boldsymbol{y}')}$, is the *likelihood ratio of the thinking trace* $\boldsymbol{z}_k$, and the second term, $\pi_\theta(\mathcal{Y}_{\boldsymbol{x}}|\boldsymbol{x}, \boldsymbol{z}_k)$, is the probability of producing a correct answer given $\boldsymbol{x}$ and $\boldsymbol{z}_k$. In reasoning models, a single trace $\boldsymbol{z}_k$ may contain thousands of tokens. Directly computing the likelihood ratio over such long sequences often leads to high variance, a phenomenon also reported in concurrent studies (Cetin et al., 2025; Zheng et al., 2025). To mitigate this issue, we use the geometric mean $\left( \frac{\pi_\theta(\boldsymbol{z}_k|\boldsymbol{x})}{q_\phi(\boldsymbol{z}_k|\boldsymbol{x}, \boldsymbol{y}')} \right)^{1/|\boldsymbol{z}_k|}$ as a surrogate for the likelihood ratio of $\boldsymbol{z}_k$. This per-token normalization reduces variance at the cost of introducing some bias, effectively spreading the ratio evenly across the thinking tokens.

As for computing $\pi_\theta(\mathcal{Y}_{\boldsymbol{x}}|\boldsymbol{x}, \boldsymbol{z}_k)$, we consider two unbiased estimators: **(i) likelihood-based estimator** is $\pi_\theta(\mathcal{Y}_{\boldsymbol{x}}|\boldsymbol{x}, \boldsymbol{z}) = |\mathcal{Y}_{\boldsymbol{x}}| \cdot \mathbb{E}_{\boldsymbol{y} \sim \mathcal{U}(\mathcal{Y}_{\boldsymbol{x}})} [\pi_\theta(\boldsymbol{y}|\boldsymbol{x}, \boldsymbol{z})]$, where $|\mathcal{Y}_{\boldsymbol{x}}|$ is cardinality of $\mathcal{Y}_{\boldsymbol{x}}$ and $\mathcal{U}(\mathcal{Y}_{\boldsymbol{x}})$ is the uniform distribution on $\mathcal{Y}_{\boldsymbol{x}}$; **(ii) accuracy-based estimator** is $\pi_\theta(\mathcal{Y}_{\boldsymbol{x}}|\boldsymbol{x}, \boldsymbol{z}) = \mathbb{E}_{\boldsymbol{y} \sim \pi_\theta(\boldsymbol{y}|\boldsymbol{x}, \boldsymbol{z})} [\mathbb{1}(\boldsymbol{y} \in \mathcal{Y}_{\boldsymbol{x}})]$, where $\mathbb{1}(\cdot)$ is the indicator function. When $|\mathcal{Y}_{\boldsymbol{x}}| = 1$, i.e., there is a unique correct answer expression $\boldsymbol{y}^*$, Zhou et al. (2025) show that the likelihood-based estimator has lower variance (in fact, zero) compared to the accuracy-based one. We now extend this comparison to general cases when $|\mathcal{Y}_{\boldsymbol{x}}| > 1$:

**Theorem 1.** *(Proof in Appendix A.3) For $|\mathcal{Y}_{\boldsymbol{x}}| > 1$, the worst-case variances of the likelihood-based estimator and the accuracy-based estimator over all possible $\pi_\theta$ (under fixed $\pi_\theta(\mathcal{Y}_{\boldsymbol{x}}|\boldsymbol{x}, \boldsymbol{z})$) are*

$$\max_{\pi_\theta} Var_{like} = (|\mathcal{Y}_{\boldsymbol{x}}| - 1) \cdot \pi_\theta(\mathcal{Y}_{\boldsymbol{x}}|\boldsymbol{x}, \boldsymbol{z})^2; \quad \max_{\pi_\theta} Var_{acc} = \pi_\theta(\mathcal{Y}_{\boldsymbol{x}}|\boldsymbol{x}, \boldsymbol{z}) \cdot (1 - \pi_\theta(\mathcal{Y}_{\boldsymbol{x}}|\boldsymbol{x}, \boldsymbol{z})). \tag{7}$$

*Therefore, the accuracy-based estimator has lower worst-case variance, i.e., $\max_{\pi_\theta} Var_{acc} \leq \max_{\pi_\theta} Var_{like}$, whenever the model accuracy (conditional on $\boldsymbol{x}, \boldsymbol{z}$) satisfies $\pi_\theta(\mathcal{Y}_{\boldsymbol{x}}|\boldsymbol{x}, \boldsymbol{z}) \geq \frac{1}{|\mathcal{Y}_{\boldsymbol{x}}|}$.*

Note that for many practical questions, the space of correct answers can be quite flexible, so typically $|\mathcal{Y}_{\boldsymbol{x}}| \gg 1$. In this regime, the accuracy-based estimator enjoys much lower worst-case variance. Based on this insight, in our experiments we estimate the weight $\rho_k$ as

$$\rho_k^{\text{est}} = \left( \frac{\pi_\theta(\boldsymbol{z}_k|\boldsymbol{x})}{q_\phi(\boldsymbol{z}_k|\boldsymbol{x}, \boldsymbol{y}')} \right)^{1/|\boldsymbol{z}_k|} \cdot \mathbb{E}_{\boldsymbol{y} \sim \pi_\theta(\boldsymbol{y}|\boldsymbol{x}, \boldsymbol{z}_k)} [\mathbb{1}(\boldsymbol{y} \in \mathcal{Y}_{\boldsymbol{x}})], \tag{8}$$

where the expectation $\mathbb{E}_{\boldsymbol{y} \sim \pi_\theta(\boldsymbol{y}|\boldsymbol{x}, \boldsymbol{z}_k)} [\mathbb{1}(\boldsymbol{y} \in \mathcal{Y}_{\boldsymbol{x}})]$ is approximated by sampling multiple candidate answers for each thinking trace $\boldsymbol{z}_k$, similar to the implementation in Qi et al. (2025).

**Estimating $\nabla_\theta \log \pi_\theta(\boldsymbol{z}_k, \mathcal{Y}_{\boldsymbol{x}}|\boldsymbol{x})$.** When evaluating $\nabla_\theta \mathcal{L}_{\text{ELBO}}^K$ in Eq. (6), we need the gradient $\nabla_\theta \log \pi_\theta(\boldsymbol{z}_k, \mathcal{Y}_{\boldsymbol{x}}|\boldsymbol{x}) = \nabla_\theta \log \pi_\theta(\boldsymbol{z}_k|\boldsymbol{x}) + \nabla_\theta \log \pi_\theta(\mathcal{Y}_{\boldsymbol{x}}|\boldsymbol{x}, \boldsymbol{z}_k)$. The first term, $\nabla_\theta \log \pi_\theta(\boldsymbol{z}_k|\boldsymbol{x})$, is straightforward to calculate. For the second term, $\nabla_\theta \log \pi_\theta(\mathcal{Y}_{\boldsymbol{x}}|\boldsymbol{x}, \boldsymbol{z}_k)$, we also adopt an *accuracy-based estimator*: $\nabla_\theta \log \pi_\theta(\mathcal{Y}_{\boldsymbol{x}}|\boldsymbol{x}, \boldsymbol{z}_k) = \frac{\mathbb{E}_{\boldsymbol{y} \sim \pi_\theta(\boldsymbol{y}|\boldsymbol{x}, \boldsymbol{z}_k)} [\mathbb{1}(\boldsymbol{y} \in \mathcal{Y}_{\boldsymbol{x}}) \nabla_\theta \log \pi_\theta(\boldsymbol{y}|\boldsymbol{x}, \boldsymbol{z}_k)]}{\mathbb{E}_{\boldsymbol{y} \sim \pi_\theta(\boldsymbol{y}|\boldsymbol{x}, \boldsymbol{z}_k)} [\mathbb{1}(\boldsymbol{y} \in \mathcal{Y}_{\boldsymbol{x}})]}$. In practice, the expectations w.r.t. $\pi_\theta(\boldsymbol{y}|\boldsymbol{x}, \boldsymbol{z}_k)$ are approximated using the same samples drawn to estimate $\rho_k^{\text{est}}$.

---

**Algorithm 1** Training pipeline of variational reasoning

---

**Inputs:** An initial reasoning model $\pi_{\theta_0}(\boldsymbol{z}, \boldsymbol{y}|\boldsymbol{x})$, variational posterior $q_\phi(\boldsymbol{z}|\boldsymbol{x}, \boldsymbol{y}')$, question-answer dataset $\{\boldsymbol{x}, \boldsymbol{y}_{\boldsymbol{x}}^*\} \in \mathcal{X}$, where $\boldsymbol{y}_{\boldsymbol{x}}^* \in \mathcal{Y}_{\boldsymbol{x}}$ is one of the reference answers corresponding to $\boldsymbol{x}$
**Inputs:** Rollout numbers $K$ and $M$, training rounds $T$, steps per round $S_\theta$ and $S_\phi$, optimizer $\mathcal{O}$
**Outputs:** The trained model parameters $\theta_T$ and variational parameters $\phi_T$

1: **Initialize** $q_{\phi_0}(\boldsymbol{z}|\boldsymbol{x}, \boldsymbol{y}') \xleftarrow{\phi_0 \text{ copy } \theta_0} \pi_{\theta_0}([\boldsymbol{z}, \texttt{</think>}] | [\boldsymbol{x}, \texttt{<hint>}, \boldsymbol{y}', \texttt{</hint>}, \texttt{<think>}])$
2: **Construct** $\mathcal{Y}_{\boldsymbol{x}}$ (or its subset by rephrasing $\boldsymbol{y}_{\boldsymbol{x}}^*$) and rule-based/model-based verifier $\mathbb{1}(\boldsymbol{y} \in \mathcal{Y}_{\boldsymbol{x}})$
3: **for** $t = 1$ **to** $T$ **do**
        ## Updating variational parameters $\phi_t$ with $\boxed{\nabla_\phi \mathcal{L}_{\text{forward}}^M}$ in Eq. (9); initializing $\phi_t \leftarrow \phi_{t-1}$
4:      **for** $s = 1$ **to** $S_\phi$ **do**
5:          Sample a training batch of questions $\mathcal{B} \subset \mathcal{X}$
6:          **for all** questions $\boldsymbol{x} \in \mathcal{B}$ **do**      ## Collecting $\boldsymbol{z}_{1:M}$ and compute weights $\widetilde{w}_m$ for each $m$
7:             **Rollout** $\boldsymbol{z}_{1:M} \sim \pi_{\theta_{t-1}}(\boldsymbol{z}|\boldsymbol{x})$, $\boldsymbol{y}' \sim q_\phi(\boldsymbol{y}') = \mathcal{U}(\mathcal{Y}_{\boldsymbol{x}})$
8:             **Compute** $w_m = \mathbb{E}_{\boldsymbol{y} \sim \pi_{\theta_{t-1}}(\boldsymbol{y}|\boldsymbol{x}, \boldsymbol{z}_m)}[\mathbb{1}(\boldsymbol{y} \in \mathcal{Y}_{\boldsymbol{x}})]$ and $\widetilde{w}_m = \frac{w_m}{\sum_{j=1}^M w_j}$
9:          **Update** $\phi_t \leftarrow \mathcal{O}.\text{step}\left(\phi_t, \boxed{\frac{1}{|\mathcal{B}|} \sum_{\boldsymbol{x} \in \mathcal{B}} \sum_{m=1}^M \widetilde{w}_m \nabla_{\phi_t} \log q_{\phi_t}(\boldsymbol{z}_m|\boldsymbol{x}, \boldsymbol{y}')}\right)$
        ## Updating model parameters $\theta_t$ with $\boxed{\nabla_\theta \mathcal{L}_{\text{ELBO}}^K}$ in Eq. (6); initializing $\theta_t \leftarrow \theta_{t-1}$
10:      **for** $s = 1$ **to** $S_\theta$ **do**
11:          Sample a training batch of questions $\mathcal{B} \subset \mathcal{X}$
12:          **for all** questions $\boldsymbol{x} \in \mathcal{B}$ **do**     ## Collecting $\boldsymbol{z}_{1:K}$ and compute weights $\widetilde{\rho}_k$ for each $k$
13:             **Rollout** $\boldsymbol{z}_{1:K} \sim q_{\phi_t}(\boldsymbol{z}|\boldsymbol{x}, \boldsymbol{y}')$, $\boldsymbol{y}' \sim q_\phi(\boldsymbol{y}') = \mathcal{U}(\mathcal{Y}_{\boldsymbol{x}})$     ## Estimate $\rho_k^{\text{est}}$ by Eq. (8)
14:             **Compute** $\rho_k^{\text{est}} = \left(\frac{\pi_{\theta_t}(\boldsymbol{z}_k|\boldsymbol{x})}{q_{\phi_t}(\boldsymbol{z}_k|\boldsymbol{x}, \boldsymbol{y}')}\right)^{1/|\boldsymbol{z}_k|} \cdot \mathbb{E}_{\boldsymbol{y} \sim \pi_{\theta_t}(\boldsymbol{y}|\boldsymbol{x}, \boldsymbol{z}_k)}[\mathbb{1}(\boldsymbol{y} \in \mathcal{Y}_{\boldsymbol{x}})]$ and $\widetilde{\rho}_k = \frac{\rho_k^{\text{est}}}{\sum_{j=1}^K \rho_j^{\text{est}}}$
15:             **Compute** $\nabla_{\theta_t} \log \pi_{\theta_t}(\mathcal{Y}_{\boldsymbol{x}}|\boldsymbol{x}, \boldsymbol{z}_k) = \frac{\mathbb{E}_{\boldsymbol{y} \sim \pi_{\theta_t}(\boldsymbol{y}|\boldsymbol{x}, \boldsymbol{z}_k)}[\mathbb{1}(\boldsymbol{y} \in \mathcal{Y}_{\boldsymbol{x}}) \nabla_{\theta_t} \log \pi_{\theta_t}(\boldsymbol{y}|\boldsymbol{x}, \boldsymbol{z}_k)]}{\mathbb{E}_{\boldsymbol{y} \sim \pi_{\theta_t}(\boldsymbol{y}|\boldsymbol{x}, \boldsymbol{z}_k)}[\mathbb{1}(\boldsymbol{y} \in \mathcal{Y}_{\boldsymbol{x}})]}$
16:          **Update** $\theta_t \leftarrow \mathcal{O}.\text{step}\left(\theta_t, \boxed{\frac{1}{|\mathcal{B}|} \sum_{\boldsymbol{x} \in \mathcal{B}} \sum_{k=1}^K \widetilde{\rho}_k \nabla_{\theta_t}(\log \pi_{\theta_t}(\boldsymbol{z}_k|\boldsymbol{x}) + \log \pi_{\theta_t}(\mathcal{Y}_{\boldsymbol{x}}|\boldsymbol{x}, \boldsymbol{z}_k))}\right)$
17: **return** $\theta_T$ and $\phi_T$

---

## 2.3   OPTIMIZING THE VARIATIONAL POSTERIOR VIA FORWARD KL DIVERGENCE

While Eq. (5) provides IWAE-style bounds that yield tighter optimization of the MLE objective w.r.t. the model parameters $\theta$ (through $\nabla_\theta \mathcal{L}_{\text{ELBO}}^K$), our pilot experiments show unexpected behavior for the optimization of the variational parameters $\phi$ (through $\nabla_\phi \mathcal{L}_{\text{ELBO}}$ or $\nabla_\phi \mathcal{L}_{\text{ELBO}}^K$). Recall from Eq. (4) that the ELBO objective minimizes the *reverse* KL divergence $\mathbb{D}_{\text{KL}}(q_\phi(\boldsymbol{z}|\boldsymbol{x}, \boldsymbol{y}')||P_\theta(\boldsymbol{z}|\boldsymbol{x}, \mathcal{Y}_{\boldsymbol{x}}))$, where both the expectation and Monte Carlo samples are taken under $q_\phi(\boldsymbol{z}|\boldsymbol{x}, \boldsymbol{y}')$. In practice, however, the policy model $\pi_\theta(\boldsymbol{z}|\boldsymbol{x})$ is often already well-trained due to pretraining of base LLMs (Liu et al., 2025), while the variational posterior $q_\phi(\boldsymbol{z}|\boldsymbol{x}, \boldsymbol{y}')$ may *struggle to effectively use hints $\boldsymbol{y}'$ from correct answers without collapsing into shortcut reasoning* (e.g., directly leaking answer tokens into the thinking trace). To address this imbalance, we propose to optimize $q_\phi(\boldsymbol{z}|\boldsymbol{x}, \boldsymbol{y}')$ using the *forward* KL divergence $\mathbb{D}_{\text{KL}}(P_\theta(\boldsymbol{z}|\boldsymbol{x}, \mathcal{Y}_{\boldsymbol{x}})||q_\phi(\boldsymbol{z}|\boldsymbol{x}, \boldsymbol{y}'))$, whose gradient w.r.t. $\phi$ can be written as:[2]

$$\nabla_\phi \mathbb{D}_{\text{KL}}(P_\theta(\boldsymbol{z}|\boldsymbol{x}, \mathcal{Y}_{\boldsymbol{x}})||q_\phi(\boldsymbol{z}|\boldsymbol{x}, \boldsymbol{y}')) \simeq \mathbb{E}_{\boldsymbol{z}_{1:M} \sim \pi_\theta(\boldsymbol{z}|\boldsymbol{x})}\left[\sum_{m=1}^M \widetilde{w}_m \nabla_\phi \log q_\phi(\boldsymbol{z}_m|\boldsymbol{x}, \boldsymbol{y}')\right] \triangleq \nabla_\phi \mathcal{L}_{\text{forward}}^M,$$
(9)

$$\text{where} \quad \widetilde{w}_m = \frac{w_m}{\sum_{j=1}^M w_j} \quad \text{and} \quad w_m = \pi_\theta(\mathcal{Y}_{\boldsymbol{x}}|\boldsymbol{x}, \boldsymbol{z}_m) = \mathbb{E}_{\boldsymbol{y} \sim \pi_\theta(\boldsymbol{y}|\boldsymbol{x}, \boldsymbol{z}_m)}[\mathbb{1}(\boldsymbol{y} \in \mathcal{Y}_{\boldsymbol{x}})].$$

This approximation, $\nabla_\phi \mathcal{L}_{\text{forward}}^M$, follows a derivation similar to Bornschein & Bengio (2015) (see Appendix A.4), with the sample size $M$ not necessarily equal to $K$ used in $\nabla_\theta \mathcal{L}_{\text{ELBO}}^K$. Unlike IWAE, this objective is an approximation rather than a lower bound. Optimizing Eq. (9) can be viewed as a weighted supervised finetuning (SFT) for $q_\phi(\boldsymbol{z}|\boldsymbol{x}, \boldsymbol{y}')$, where training data is sampled from $\pi_\theta(\boldsymbol{z}|\boldsymbol{x})$.

We summarize the overall training pipeline of our variational reasoning method in Algorithm 1. In our experiments, we train for only a single round ($T = 1$), leaving multi-round training as an interesting direction for future work.

---

[2]We assume that $q_\phi(\boldsymbol{y}')$ is a fixed prior distribution and does not involve gradients.

## 3 CONNECTION TO OTHER METHODS

In prior work, the thinking trace $z$ and the final answer $y$ are often treated together as the full output in the formulations (Shao et al., 2024; Guo et al., 2025; Liu et al., 2025; Wu et al., 2025). By explicitly decomposing the output into a thinking process $z$ and an answer $y$, as we have done above, we can gain fresh perspectives on how our formulation relates to other mainstream methods.

**Connection to rejection sampling finetuning (RFT).** RFT methods (Dong et al., 2023; Touvron et al., 2023) generate multiple candidate outputs for each input $x$ using a reference model $\pi_{\text{ref}}$, and then select the candidate with the highest reward. The reference model may be a strong teacher model or identical to the learner $\pi_\theta$ (i.e., $\pi_{\text{ref}} = \pi_\theta^{\text{sg}}$, with sg denoting stop-gradient). Formally, the gradient of RFT training objective, *focusing only on the learning of the thinking trace $z$*, can be written as:

$$
\begin{aligned}
\nabla_\theta \mathcal{L}_{\text{RFT}}(\boldsymbol{x}, \pi_\theta) &\triangleq \nabla_\theta \mathbb{E}_{\pi_{\text{ref}}(\boldsymbol{z}|\boldsymbol{x})} \mathbb{E}_{\pi_{\text{ref}}(\boldsymbol{y}|\boldsymbol{x},\boldsymbol{z})} \left[ \mathbb{1}(\boldsymbol{y} \in \mathcal{Y}_{\boldsymbol{x}}) \cdot \log \pi_\theta(\boldsymbol{z}, \boldsymbol{y}|\boldsymbol{x}) \right] \\
&= \nabla_\theta \mathbb{E}_{\pi_{\text{ref}}(\boldsymbol{z}|\boldsymbol{x})} \left[ \pi_{\text{ref}}(\mathcal{Y}_{\boldsymbol{x}}|\boldsymbol{x}, \boldsymbol{z}) \cdot (\log \pi_\theta(\boldsymbol{z}|\boldsymbol{x}) + \log \pi_\theta(\boldsymbol{y}|\boldsymbol{x}, \boldsymbol{z})) \right] \\
&\xrightarrow[\pi_\theta(\boldsymbol{z}|\boldsymbol{x})]{\text{only w.r.t.}} \nabla_\theta \mathbb{E}_{\pi_{\text{ref}}(\boldsymbol{z}|\boldsymbol{x})} \left[ \pi_{\text{ref}}(\mathcal{Y}_{\boldsymbol{x}}|\boldsymbol{x}, \boldsymbol{z}) \cdot \log \pi_\theta(\boldsymbol{z}|\boldsymbol{x}) \right] \\
&= -P_{\text{ref}}(\mathcal{Y}_{\boldsymbol{x}}|\boldsymbol{x}) \cdot \nabla_\theta \mathbb{D}_{\text{KL}}(P_{\text{ref}}(\boldsymbol{z}|\boldsymbol{x}, \mathcal{Y}_{\boldsymbol{x}}) || \pi_\theta(\boldsymbol{z}|\boldsymbol{x})).
\end{aligned}
\tag{10}
$$

Here $P_{\text{ref}}(\mathcal{Y}_{\boldsymbol{x}}|\boldsymbol{x}) = \sum_{\boldsymbol{y} \in \mathcal{Y}_{\boldsymbol{x}}} P_{\text{ref}}(\boldsymbol{y}|\boldsymbol{x})$ denotes the *model accuracy* on instruction $x$, and the true posterior of the reference model is $P_{\text{ref}}(\boldsymbol{z}|\boldsymbol{x}, \mathcal{Y}_{\boldsymbol{x}}) = \frac{\pi_{\text{ref}}(\mathcal{Y}_{\boldsymbol{x}}|\boldsymbol{x},\boldsymbol{z})\pi_{\text{ref}}(\boldsymbol{z}|\boldsymbol{x})}{P_{\text{ref}}(\mathcal{Y}_{\boldsymbol{x}}|\boldsymbol{x})}$. As seen, the RFT objective can be viewed as maximizing a forward KL divergence weighted by $P_{\text{ref}}(\mathcal{Y}_{\boldsymbol{x}}|\boldsymbol{x})$, with the optimal solution $\boxed{\pi_\theta^*(\boldsymbol{z}|\boldsymbol{x}) = P_{\text{ref}}(\boldsymbol{z}|\boldsymbol{x}, \mathcal{Y}_{\boldsymbol{x}})}$. In practice, this weighting downplays hard questions with small $P_{\text{ref}}(\mathcal{Y}_{\boldsymbol{x}}|\boldsymbol{x})$, biasing training toward easier ones. In contrast, our formulation in Eq. (9) treats all questions more evenly, ensuring that the objective remains attentive to difficult cases.

**Connection to binary reward RL.** In the case of RL training with a 0–1 binary reward, the training objective can be written in a form similar to Eq. (10), focusing only on the thinking trace $z$:

$$
\begin{aligned}
\nabla_\theta \mathcal{L}_{\text{bi-RL}}(\boldsymbol{x}, \pi_\theta) &\triangleq \nabla_\theta \mathbb{E}_{\pi_\theta(\boldsymbol{z}|\boldsymbol{x})} \mathbb{E}_{\pi_\theta(\boldsymbol{y}|\boldsymbol{x},\boldsymbol{z})} \left[ \mathbb{1}(\boldsymbol{y} \in \mathcal{Y}_{\boldsymbol{x}}) \right] \\
&\xrightarrow[\pi_\theta(\boldsymbol{z}|\boldsymbol{x})]{\text{only w.r.t.}} \nabla_\theta \mathbb{E}_{\pi_\theta^{\text{sg}}(\boldsymbol{z}|\boldsymbol{x})} \left[ \pi_\theta^{\text{sg}}(\mathcal{Y}_{\boldsymbol{x}}|\boldsymbol{x}, \boldsymbol{z}) \cdot \log \pi_\theta(\boldsymbol{z}|\boldsymbol{x}) \right] \\
&= -P_\theta^{\text{sg}}(\mathcal{Y}_{\boldsymbol{x}}|\boldsymbol{x}) \cdot \nabla_\theta \mathbb{D}_{\text{KL}}(P_\theta^{\text{sg}}(\boldsymbol{z}|\boldsymbol{x}, \mathcal{Y}_{\boldsymbol{x}}) || \pi_\theta(\boldsymbol{z}|\boldsymbol{x})).
\end{aligned}
\tag{11}
$$

Thus, the local gradient for $\pi_\theta(\boldsymbol{z}|\boldsymbol{x})$ in binary-reward RL is equivalent to minimizing the forward KL divergence between $\pi_\theta(\boldsymbol{z}|\boldsymbol{x})$ and the true posterior $P_\theta^{\text{sg}}(\boldsymbol{z}|\boldsymbol{x}, \mathcal{Y}_{\boldsymbol{x}}) = \frac{\pi_\theta(\mathcal{Y}_{\boldsymbol{x}}|\boldsymbol{x},\boldsymbol{z})\pi_\theta(\boldsymbol{z}|\boldsymbol{x})}{P_\theta(\mathcal{Y}_{\boldsymbol{x}}|\boldsymbol{x})}$, with the update further weighted by the model accuracy $P_\theta^{\text{sg}}(\mathcal{Y}_{\boldsymbol{x}}|\boldsymbol{x})$. Interestingly, the RL training objective itself is $\mathcal{L}_{\text{bi-RL}}(\boldsymbol{x}, \pi_\theta) = \mathbb{E}_{\pi_\theta(\boldsymbol{z}|\boldsymbol{x})} \mathbb{E}_{\pi_\theta(\boldsymbol{y}|\boldsymbol{x},\boldsymbol{z})} \left[ \mathbb{1}(\boldsymbol{y} \in \mathcal{Y}_{\boldsymbol{x}}) \right] = P_\theta(\mathcal{Y}_{\boldsymbol{x}}|\boldsymbol{x})$, whereas the MLE objective in Eq. (2) for our variational reasoning framework instead maximizes $\log P_\theta(\mathcal{Y}_{\boldsymbol{x}}|\boldsymbol{x})$.

Furthermore, in Group Relative Policy Optimization **(GRPO)** (Shao et al., 2024), one of the most widely used RL training objectives, the reward is normalized by the standard deviation of rewards within a group of rollouts. Under 0–1 binary reward, each rollout reward follows a Bernoulli distribution with mean $P_\theta(\mathcal{Y}_{\boldsymbol{x}}|\boldsymbol{x})$ and standard deviation $\sqrt{P_\theta(\mathcal{Y}_{\boldsymbol{x}}|\boldsymbol{x}) \cdot (1 - P_\theta(\mathcal{Y}_{\boldsymbol{x}}|\boldsymbol{x}))}$. The gradient of the GRPO objective can therefore be derived as (we omit the min and clip operations for brevity)

$$
\nabla_\theta \mathcal{L}_{\text{bi-GRPO}}(\boldsymbol{x}, \pi_\theta) \xrightarrow[\pi_\theta(\boldsymbol{z}|\boldsymbol{x})]{\text{only w.r.t.}} -\sqrt{\frac{P_\theta^{\text{sg}}(\mathcal{Y}_{\boldsymbol{x}}|\boldsymbol{x})}{1 - P_\theta^{\text{sg}}(\mathcal{Y}_{\boldsymbol{x}}|\boldsymbol{x})}} \cdot \nabla_\theta \mathbb{D}_{\text{KL}}(P_\theta^{\text{sg}}(\boldsymbol{z}|\boldsymbol{x}, \mathcal{Y}_{\boldsymbol{x}}) || \pi_\theta(\boldsymbol{z}|\boldsymbol{x})),
\tag{12}
$$

which follows from $\nabla_\theta \mathcal{L}_{\text{bi-GRPO}}(\boldsymbol{x}, \pi_\theta) = \nabla_\theta \mathcal{L}_{\text{bi-RL}}(\boldsymbol{x}, \pi_\theta) / \sqrt{P_\theta(\mathcal{Y}_{\boldsymbol{x}}|\boldsymbol{x}) \cdot (1 - P_\theta(\mathcal{Y}_{\boldsymbol{x}}|\boldsymbol{x}))}$. Thus, the local gradient for $\pi_\theta(\boldsymbol{z}|\boldsymbol{x})$ in GRPO is still equivalent to minimizing the forward KL divergence between $\pi_\theta(\boldsymbol{z}|\boldsymbol{x})$ and the true posterior $P_\theta^{\text{sg}}(\boldsymbol{z}|\boldsymbol{x}, \mathcal{Y}_{\boldsymbol{x}})$. Note that the per-instruction weight becomes $\sqrt{P_\theta(\mathcal{Y}_{\boldsymbol{x}}|\boldsymbol{x})/(1 - P_\theta(\mathcal{Y}_{\boldsymbol{x}}|\boldsymbol{x}))}$, which increases monotonically with model accuracy and therefore also emphasizes easier questions with higher $P_\theta(\mathcal{Y}_{\boldsymbol{x}}|\boldsymbol{x})$. In Appendix A.5, we extend these analyses and derive gradients for more general RL reward shaping, including cases with a *format reward*.

## 4 EXPERIMENTS

**Datasets.** We train on the Bespoke-Stratos-17k dataset curated by Li et al. (2025a), which combines math problems from Numina-Math (Li et al., 2024) with code problems from APPS (Hendrycks et al., 2021a) and TACO (Li et al., 2023). The dataset contains 16,710 samples, each paired with

Table 1: Performance of models trained from **Qwen3-4B-Base**. All models are trained on Bespoke-Stratos-17k except for General-Reasoner-4B. The best and second-best results are highlighted using **bold text** and underlined text, respectively.

| Method | MATH500 Avg@2 | AIME24 Avg@32 | AIME25 Avg@32 | AMC23 Avg@32 | OlympiadBench Avg@2 | Avg |
|---|---|---|---|---|---|---|
| Qwen3-4B-Base | 45.30 | 4.79 | 5.73 | 27.73 | 23.37 | 21.38 |
| General-Reasoner-4B | 71.70 | 19.06 | 16.77 | 55.00 | 45.18 | 41.54 |
| Bespoke-Stratos-4B[†] | 84.70 | 27.29 | 24.17 | 70.16 | 50.45 | 51.35 |
| Ours-PB-GML-4B | 87.30 | **33.54** | 26.77 | 74.06 | 54.45 | 55.23 |
| Ours-PB-Acc-4B | **88.30** | 31.67 | **27.29** | **75.63** | **55.71** | **55.72** |

| Method | GPQA-D Avg@8 | LCB-E Avg@8 | LCB-M Avg@8 | LCB-H Avg@8 | MMLU-Pro Avg@1 | Avg |
|---|---|---|---|---|---|---|
| Qwen3-4B-Base | 29.10 | 18.54 | 5.46 | 1.32 | 36.89 | 18.26 |
| General-Reasoner-4B | 40.97 | 61.40 | 17.90 | 2.85 | 61.36 | 36.90 |
| Bespoke-Stratos-4B[†] | 44.95 | 71.22 | 19.54 | 3.25 | 63.03 | 40.40 |
| Ours-PB-GML-4B | **45.52** | 79.53 | 31.25 | **6.20** | 65.52 | 45.60 |
| Ours-PB-Acc-4B | 45.33 | **80.29** | **33.68** | 5.79 | **65.53** | **46.12** |

Table 2: Performance of models trained from **Qwen3-8B-Base** trained on Bespoke-Stratos-17k.

| Method | MATH500 Avg@2 | AIME24 Avg@32 | AIME25 Avg@32 | AMC23 Avg@32 | OlympiadBench Avg@2 | Avg |
|---|---|---|---|---|---|---|
| Qwen3-8B-Base | 65.20 | 11.46 | 10.10 | 45.00 | 34.72 | 33.30 |
| Bespoke-Stratos-8B[†] | 89.70 | 39.58 | 28.85 | 78.91 | 55.64 | 58.54 |
| Ours-PB-GML-8B | 91.60 | 44.06 | 31.67 | 83.59 | 58.23 | 61.83 |
| Ours-PB-Acc-8B | **91.80** | **45.63** | **31.98** | **85.47** | **58.98** | **62.77** |

| Method | GPQA-D Avg@8 | LCB-E Avg@8 | LCB-M Avg@8 | LCB-H Avg@8 | MMLU-Pro Avg@1 | Avg |
|---|---|---|---|---|---|---|
| Qwen3-8B-Base | 35.42 | 41.14 | 13.65 | 1.42 | 45.62 | 27.45 |
| Bespoke-Stratos-8B[†] | 53.03 | 81.53 | 36.89 | 7.11 | 68.74 | 49.46 |
| Ours-PB-GML-8B | 52.72 | **87.36** | 45.51 | **13.82** | **70.76** | 54.03 |
| Ours-PB-Acc-8B | **53.66** | 86.47 | **49.33** | 13.21 | **70.76** | **54.69** |

a long-thinking trace generated by DeepSeek-R1 (Guo et al., 2025) or QwQ-32B-Preview (Team, 2024). To ensure fair evaluation, all training data is strictly separated from the test sets.

**Baselines.** We compare against a broad set of baselines for rigorous evaluation. For Qwen2.5-Instruct models, we include Bespoke-Stratos (Labs, 2025) and RLT (Cetin et al., 2025) at the 7B and 32B scales, where RLT trains teachers via RL to improve distillation. For Qwen3-Base models, we compare with Bespoke-Stratos-4B/8B[†] and General-Reasoner-4B (Ma et al., 2025), the latter trained with GRPO (Shao et al., 2024) and a model-based verifier. Models marked [†] are trained by us; others are official releases. All Bespoke-Stratos models are distilled on Bespoke-Stratos-17k. This ensures fair comparison, as all methods (except General-Reasoner) follow the same training recipes and datasets.

**Evaluation.** We assess our models on a broad set of challenging benchmarks: **MATH500** (Hendrycks et al., 2021b), a subset of competition math curated by Lightman et al. (2024); **AIME24&25** (MAA, 2025); **AMC23** (MAA, 2023); **OlympiadBench** (He et al., 2024); **LiveCodeBench** (Jain et al., 2025), with Easy, Medium, and Hard subsets (**LCB-E**, **LCB-M**, **LCB-H**) for fine-grained coding evaluation; **GPQA-Diamond** (Rein et al., 2024) (**GPQA-D**), graduate-level natural science questions; and **MMLU-Pro** (Wang et al., 2024), a diverse multiple-choice benchmark. Among these, GPQA-D and MMLU-Pro are out-of-distribution (OOD) relative to our training data.

**Decoding and average accuracy.** Following Hochlehnert et al. (2025), we sample responses with `temperature=0.7` and report average accuracy over $k$ responses per question (Avg@$k$). To reduce randomness and ensure fair comparison, we use larger $k$ for smaller datasets: Avg@32 for AIME24, AIME25, and AMC23; Avg@8 for GPQA-Diamond, LCB-E, LCB-M, and LCB-H; Avg@2 for MATH500 and OlympiadBench; and Avg@1 for the large MMLU-Pro (12k+ questions). All evaluations are conducted with SkyThought (Team, 2025a), with additional details in Appendix D.

Table 3: Performance of models trained from Qwen2.5-32B-Instruct trained on Bespoke-Stratos-17k. The best and second-best results are highlighted using **bold text** and underlined text, respectively.

| Method | MATH500 Avg@2 | AIME24 Avg@32 | AIME25 Avg@32 | AMC23 Avg@32 | OlympiadBench Avg@2 | Avg |
|---|---|---|---|---|---|---|
| Qwen2.5-32B-Instruct | 80.70 | 15.83 | 12.08 | 61.95 | 46.96 | 43.51 |
| Bespoke-Stratos-32B | 92.60 | 55.42 | 46.88 | 92.19 | **64.68** | 70.34 |
| RLT-32B | **93.50** | 56.77 | 47.19 | 91.48 | 63.21 | 70.43 |
| Ours-PA-GML-32B | 93.20 | 56.56 | 48.13 | **93.98** | 64.24 | 71.22 |
| Ours-PA-Acc-32B | **93.50** | **58.85** | **50.31** | 92.97 | 64.39 | **72.01** |

| Method | GPQA-D Avg@8 | LCB-E Avg@8 | LCB-M Avg@8 | LCB-H Avg@8 | MMLU-Pro Avg@1 | Avg |
|---|---|---|---|---|---|---|
| Qwen2.5-32B-Instruct | 46.28 | 79.88 | 40.60 | 9.76 | 59.19 | 47.14 |
| Bespoke-Stratos-32B | 57.57 | 94.78 | **73.54** | 30.48 | 75.22 | 66.32 |
| RLT-32B | 59.09 | 93.20 | 72.15 | 29.78 | 74.88 | 65.82 |
| Ours-PA-GML-32B | **60.92** | **95.19** | 72.21 | **35.57** | **75.57** | **67.89** |
| Ours-PA-Acc-32B | 60.73 | 94.78 | 73.18 | 31.81 | 75.55 | 67.21 |

Figure 1: Training loss and gradient norm of different methods during Qwen3-Base model training.

## 4.1 TRAINING DETAILS FOR VARIATIONAL REASONING

We conduct experiments on Qwen2.5-7B-Instruct, Qwen2.5-32B-Instruct (Yang et al., 2024), Qwen3-4B-Base, and Qwen3-8B-Base (Team, 2025b). Following Algorithm 1, we first train an *initial* reasoning model $\pi_{\theta_0}$ on Bespoke-Stratos-17k using the recipe from Labs (2025), and then a variational posterior $q_\phi$ with the forward KL divergence (Eq. (9)) on the same dataset. These models are later used to compute the weights $\widetilde{\rho}_k$. For $\pi_{\theta_0}$, we adopt the prompt template from Labs (2025); for $q_\phi$, we test two alternative templates ("**-PA**" and "**-PB**", see Appendix E). Both $\pi_{\theta_0}$ and $q_\phi$ are finetuned independently from the same base model without weight sharing (Appendix C.1).

Next, we use the trained $q_\phi$ to generate 8 responses (thinking traces and final answers) per training sample. For each response, the weight $\widetilde{\rho}_k$ in Eq. (6) is computed from $q_\phi$, $\pi_{\theta_0}$, and, when using the accuracy-based estimator, math/code verifiers from SkyThought. To estimate $\pi_\theta(\mathcal{Y}_x|x, z)$ in $\widetilde{\rho}_k$, we compare three options: a naive likelihood method ("**-L**"), an accuracy-based method ("**-Acc**", Section 2.2), and a geometric mean of token-level probabilities ("**-GML**"), as detailed in Appendix C.2.

We train the final reasoning model $\pi_\theta$ following Eq. (6) under two data settings. **17K:** the full Bespoke-Stratos-17k dataset. To enhance efficiency, we create a mixed dataset containing, for each original sample, the $q_\phi$-generated response with the highest $\widetilde{\rho}_k$ and the original sample itself. **1K:** a fixed 1,000-sample subset uniformly drawn from the full dataset, where all 8 $q_\phi$-generated responses per sample are used for weighted SFT with $\widetilde{\rho}_k$. The same 1K subset is reused across related experiments. Main results are reported with 17K, while ablations use both 17K and 1K configurations (Appendix C.3).

## 4.2 MAIN RESULTS

We evaluate our method across four model variants: Qwen3-4B/8B-Base (Tables 1 and 2) and Qwen2.5-7B/32B-Instruct (Tables 3 and 6). Extended results are provided in Appendix F.1.

**Variational reasoning performance.** All methods substantially improve the reasoning ability of the base model, but our approach consistently achieves the best results. As shown in Tables 1 and 2, variational reasoning yields substantial improvements in math, code, and other general domains compared to the base model (e.g., over 160% improvement in math and over 152% in other domains). It also surpasses all baselines in average accuracy (e.g., over 8.5% higher than the strong baseline Bespoke-Stratos-4B[†] that uses the same training data, and over 14% in other domains).

Table 4: Ablation study on the effect of conditioning the proposal distribution on $y'$.

| Method | MATH500 Avg@2 | AIME24 Avg@32 | AIME25 Avg@32 | AMC23 Avg@32 | OlympiadBench Avg@2 | Avg |
|---|---|---|---|---|---|---|
| Qwen3-4B-Base | 45.30 | 4.79 | 5.73 | 27.73 | 23.37 | 21.38 |
| Ours-4B | **88.30** | **31.67** | **27.29** | **75.63** | **55.71** | **55.72** |
| w/o $y'$ | 81.20 | 23.44 | 23.96 | 65.70 | 46.59 | 48.18 |

| Method | GPQA-D Avg@8 | LCB-E Avg@8 | LCB-M Avg@8 | LCB-H Avg@8 | MMLU-Pro Avg@1 | Avg |
|---|---|---|---|---|---|---|
| Qwen3-4B-Base | 29.10 | 18.54 | 5.46 | 1.32 | 36.89 | 18.26 |
| Ours-4B | **45.33** | **80.29** | **33.68** | **5.79** | **65.53** | **46.12** |
| w/o $y'$ | 40.53 | 67.93 | 16.63 | 2.44 | 61.49 | 37.80 |

Figure 2: Pass@K comparison of baselines versus our method based on Qwen3-4B/8B-Base.

Notably, GPQA-Diamond and MMLU-Pro can be considered out-of-distribution test sets, as our training data only cover math and code, whereas they are in-domain for General-Reasoner-4B. Despite this, our method significantly outperforms General-Reasoner on these benchmarks, suggesting that the reasoning improvements from variational reasoning generalize effectively.

Additionally, our method demonstrates robustness across different prompt templates. Performance remains consistent between Prompt Template A and B (denoted as "-PA" and "-PB" in Table 6), with both outperforming baselines. Across four model scales, the accuracy-based estimator ("-Acc") and the geometric mean of token likelihood estimator ("-GML") exhibit similar performance, though the accuracy-based variant shows a slight advantage in math-related benchmarks.

**Pass@K analysis.** We report Pass@K results of experiments based on Qwen3-4B/8B-Base for different values of K. Figure 2 reveals two key trends: (1) Our method's advantage increases with larger K on complex benchmarks (e.g., LiveCodeBench-Hard), and (2) Performance gaps diminish on simpler tasks (e.g., LiveCodeBench-Easy) and multiple-choice questions (e.g., GPQA-Diamond). This aligns with expectations, as simpler tasks offer limited room for improvement, and multiple-choice formats inherently allow high Pass@K with sufficiently large K. These results underscore the strong potential of variational reasoning in tackling complex tasks.

Table 5: Ablation study on effects of different $\pi_\theta(\mathcal{Y}_x|x, z_k)$ estimators. Experiments are done in data 1k setting. Acc: accuracy; GML: geometric mean of token likelihood; L: naive likelihood.

| Method | MATH500 Avg@2 | AIME24 Avg@32 | AIME25 Avg@32 | AMC23 Avg@32 | OlympiadBench Avg@2 | Avg |
|---|---|---|---|---|---|---|
| Qwen2.5-7B-Instruct | 75.60 | 10.94 | 7.40 | 51.10 | 39.91 | 36.99 |
| Bespoke-Stratos-7B-1K[†] | 77.20 | 16.25 | 13.96 | 53.75 | 40.88 | 40.41 |
| Ours-Acc-7B-1K | **81.30** | **19.69** | **18.44** | 61.64 | **45.99** | **45.41** |
| Ours-GML-7B-1K | **81.30** | 19.27 | 18.33 | **62.50** | 45.48 | 45.38 |
| Ours-L-7B-1K | 79.90 | 17.81 | 14.17 | 59.53 | 43.62 | 43.01 |

| Method | GPQA-D Avg@8 | LCB-E Avg@8 | LCB-M Avg@8 | LCB-H Avg@8 | MMLU-Pro Avg@1 | Avg |
|---|---|---|---|---|---|---|
| Qwen2.5-7B-Instruct | 29.99 | 62.50 | 18.20 | **3.35** | 48.20 | 32.45 |
| Bespoke-Stratos-7B-1K[†] | 37.94 | 60.37 | 13.59 | 1.22 | 56.07 | 33.84 |
| Ours-Acc-7B-1K | 41.16 | 68.13 | 21.42 | 1.42 | 60.94 | 38.62 |
| Ours-GML-7B-1K | **41.35** | **68.41** | **23.30** | 2.74 | **61.31** | **39.42** |
| Ours-L-7B-1K | 39.90 | 66.42 | 19.90 | 1.93 | 58.61 | 37.35 |

Figure 3: Effects of scaling up the number of thinking traces ($K$ in Algorithm 1) sampled from variational posterior $q_\phi$ on the performance of the final reasoning model $\pi_\theta$.

**Training dynamics.** We monitor training loss and gradient norms during training for Qwen3-4B/8B-base models (see Figure 1). Compared to Bespoke-Stratos-4B/8B[†], our method yields lower average training loss and fewer gradient norm spikes, indicating greater training stability. We attribute this stability to the $\pi_\theta/q_\phi$ ratio in $\widetilde{\rho}_k$. Specifically, for a reasoning trace $z_k$, the weight $\widetilde{\rho}_k$ is large when the trace is both high-quality (high $\pi_\theta(\mathcal{Y}_x|x, z_k)$) and aligned with the reasoning policy (high likelihood ratio $\pi_\theta(z_k|x)/q_\phi(z_k|x, y')$). This adaptive weighting promotes stable and effective training.

## 4.3 ABLATION STUDIES

As mentioned in Section 4.1, we conduct ablation studies on both the 17k and 1k data settings to better analyze variational reasoning. Additional ablations are provided in Appendix F.2.

**Scaling the number of thinking traces $z_k$.** We investigate the effect of increasing the number of traces $z$ sampled from the variational posterior $q_\phi$ (i.e., $K$ in Algorithm 1) on the performance of the reasoning model $\pi_\theta$. Experiments are conducted under the 1k data setting and faithful to Algorithm 1. We scale $K$ exponentially from 1 to 32, adjusting the batch size to keep optimization steps consistent. Results in Figure 3 suggest that increasing $K$ can further enhance model performance. This implies a practical trade-off between training computational cost and reasoning accuracy when selecting $K$.

**Conditioning on $y'$.** We ablate the necessity of conditioning on $y'$ in data 17k setting. The variant w/o $y'$ samples thinking traces $z$ by the initial reasoning model instead of the variational posterior. Results (Table 4) show that removing $y'$ as the condition negatively affects the performance.

**Different $\pi_\theta(\mathcal{Y}_x|x, z_k)$ estimators.** We ablate different estimators for $\pi_\theta(\mathcal{Y}_x|x, z_k)$ used in the weight $\widetilde{\rho}_k$ in data 1k setting (Table 5). We find estimators based on accuracy or geometric mean of token likelihood outperform the naive likelihood by a large margin, validating our analysis in Sec. 2.2.

## 5 CONCLUSION

We introduced a variational reasoning framework as a principled and stable objective for training reasoning models, while clarifying biases in existing SFT/RFT and RL methods. Beyond consistent gains over strong baselines on diverse reasoning tasks, our analysis offers a probabilistic perspective for interpreting current approaches. A natural future direction is extending training beyond a single round ($T > 1$ in Algorithm 1) and exploring richer posterior design for the answer hint $y'$.

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

# A DETAILED DERIVATIONS

In this section, we provide detailed derivations of the conclusions presented in the main text, along with some additional results.

## A.1 DERIVATION FOR EQ. (4)

The ELBO objective induced from Eq. (2) can be rewritten as

$$
\begin{aligned}
\mathcal{L}_{\text{ELBO}}(\boldsymbol{x}, \mathcal{Y}_{\boldsymbol{x}}, \boldsymbol{y}'; \pi_\theta, q_\phi) &\triangleq \mathbb{E}_{q_\phi(\boldsymbol{z}|\boldsymbol{x},\boldsymbol{y}')}\left[\log \pi_\theta(\mathcal{Y}_{\boldsymbol{x}}|\boldsymbol{x}, \boldsymbol{z})\right] - \text{KL}\left(q_\phi(\boldsymbol{z}|\boldsymbol{x}, \boldsymbol{y}')||\pi_\theta(\boldsymbol{z}|\boldsymbol{x})\right) \\
&= \mathbb{E}_{q_\phi(\boldsymbol{z}|\boldsymbol{x},\boldsymbol{y}')}\left[\log \pi_\theta(\boldsymbol{z}, \mathcal{Y}_{\boldsymbol{x}}|\boldsymbol{x})\right] + \mathcal{H}\left(q_\phi(\boldsymbol{z}|\boldsymbol{x}, \boldsymbol{y}')\right) \\
&= \log P_\theta(\mathcal{Y}_{\boldsymbol{x}}|\boldsymbol{x}) - \mathbb{D}_{\text{KL}}\left(q_\phi(\boldsymbol{z}|\boldsymbol{x}, \boldsymbol{y}')||P_\theta(\boldsymbol{z}|\boldsymbol{x}, \mathcal{Y}_{\boldsymbol{x}})\right),
\end{aligned} \tag{13}
$$

where $\mathcal{H}(\cdot)$ is entropy function and $P_\theta(\boldsymbol{z}|\boldsymbol{x}, \mathcal{Y}_{\boldsymbol{x}}) = \frac{\pi_\theta(\mathcal{Y}_{\boldsymbol{x}}|\boldsymbol{x}, \boldsymbol{z})\pi_\theta(\boldsymbol{z}|\boldsymbol{x})}{P_\theta(\mathcal{Y}_{\boldsymbol{x}}|\boldsymbol{x})}$ is the true posterior distribution.

## A.2 DERIVATION FOR EQ. (6)

Given the IWAE-style lower bound $\mathcal{L}_{\text{ELBO}}^K(\boldsymbol{x}, \mathcal{Y}_{\boldsymbol{x}}, \boldsymbol{y}'; \pi_\theta, q_\phi)$ in Eq. (5), we can derive its gradient w.r.t. model parameters $\theta$ as:

$$
\begin{aligned}
&\nabla_\theta \mathcal{L}_{\text{ELBO}}^K(\boldsymbol{x}, \mathcal{Y}_{\boldsymbol{x}}, \boldsymbol{y}'; \pi_\theta, q_\phi) \\
&= \nabla_\theta \mathbb{E}_{\boldsymbol{z}_{1:K} \sim q_\phi(\boldsymbol{z}|\boldsymbol{x},\boldsymbol{y}')}\left[\log \frac{1}{K}\sum_{k=1}^K \frac{\pi_\theta(\boldsymbol{z}_k, \mathcal{Y}_{\boldsymbol{x}}|\boldsymbol{x})}{q_\phi(\boldsymbol{z}_k|\boldsymbol{x}, \boldsymbol{y}')}\right] \\
&= \mathbb{E}_{\boldsymbol{z}_{1:K} \sim q_\phi(\boldsymbol{z}|\boldsymbol{x},\boldsymbol{y}')}\left[\frac{\frac{1}{K}\sum_{k=1}^K \frac{\nabla_\theta \pi_\theta(\boldsymbol{z}_k, \mathcal{Y}_{\boldsymbol{x}}|\boldsymbol{x})}{q_\phi(\boldsymbol{z}_k|\boldsymbol{x}, \boldsymbol{y}')}}{\frac{1}{K}\sum_{k=1}^K \frac{\pi_\theta(\boldsymbol{z}_k, \mathcal{Y}_{\boldsymbol{x}}|\boldsymbol{x})}{q_\phi(\boldsymbol{z}_k|\boldsymbol{x}, \boldsymbol{y}')}}\right] \\
&= \mathbb{E}_{\boldsymbol{z}_{1:K} \sim q_\phi(\boldsymbol{z}|\boldsymbol{x},\boldsymbol{y}')}\left[\sum_{k=1}^K \widetilde{\rho}_k \nabla_\theta \log \pi_\theta(\boldsymbol{z}_k, \mathcal{Y}_{\boldsymbol{x}}|\boldsymbol{x})\right] \\
&\quad\quad \text{where}\quad \widetilde{\rho}_k = \frac{\rho_k}{\sum_{j=1}^K \rho_j}\quad \text{and}\quad \rho_k = \frac{\pi_\theta(\boldsymbol{z}_k, \mathcal{Y}_{\boldsymbol{x}}|\boldsymbol{x})}{q_\phi(\boldsymbol{z}_k|\boldsymbol{x}, \boldsymbol{y}')}.
\end{aligned} \tag{14}
$$

Using the notations of $\rho_k$ and $\widetilde{\rho}_k$, we can further derive the gradient w.r.t. $\phi$ as:

$$
\begin{aligned}
&\nabla_\phi \mathcal{L}_{\text{ELBO}}^K(\boldsymbol{x}, \mathcal{Y}_{\boldsymbol{x}}, \boldsymbol{y}'; \pi_\theta, q_\phi) \\
&= \nabla_\phi \mathbb{E}_{\boldsymbol{z}_{1:K} \sim q_\phi(\boldsymbol{z}|\boldsymbol{x},\boldsymbol{y}')}\left[\log \frac{1}{K}\sum_{k=1}^K \frac{\pi_\theta(\boldsymbol{z}_k, \mathcal{Y}_{\boldsymbol{x}}|\boldsymbol{x})}{q_\phi(\boldsymbol{z}_k|\boldsymbol{x}, \boldsymbol{y}')}\right] \\
&= \mathbb{E}_{\boldsymbol{z}_{1:K} \sim q_\phi(\boldsymbol{z}|\boldsymbol{x},\boldsymbol{y}')}\left[\left(\log \frac{1}{K}\sum_{k=1}^K \rho_k\right)\cdot \sum_{k=1}^K \nabla_\phi \log q_\phi(\boldsymbol{z}_k|\boldsymbol{x}, \boldsymbol{y}') - \sum_{k=1}^K \widetilde{\rho}_k \nabla_\phi \log q_\phi(\boldsymbol{z}_k|\boldsymbol{x}, \boldsymbol{y}')\right] \\
&= \mathbb{E}_{\boldsymbol{z}_{1:K} \sim q_\phi(\boldsymbol{z}|\boldsymbol{x},\boldsymbol{y}')}\left[\sum_{k=1}^K \left(-\widetilde{\rho}_k + \log \frac{1}{K}\sum_{k=1}^K \rho_k\right)\cdot \nabla_\phi \log q_\phi(\boldsymbol{z}_k|\boldsymbol{x}, \boldsymbol{y}')\right].
\end{aligned}
$$

## A.3 PROOF OF THEOREM 1

As to the computation of $\pi_\theta(\mathcal{Y}_{\boldsymbol{x}}|\boldsymbol{x}, \boldsymbol{z})$, there are two unbiased estimators:

$$
\begin{aligned}
\textbf{Likelihood-based estimator:}\quad & \pi_\theta(\mathcal{Y}_{\boldsymbol{x}}|\boldsymbol{x}, \boldsymbol{z}) = |\mathcal{Y}_{\boldsymbol{x}}|\cdot \mathbb{E}_{\boldsymbol{y} \sim \mathcal{U}(\mathcal{Y}_{\boldsymbol{x}})}\left[\pi_\theta(\boldsymbol{y}|\boldsymbol{x}, \boldsymbol{z})\right]; \\
\textbf{Accuracy-based estimator:}\quad & \pi_\theta(\mathcal{Y}_{\boldsymbol{x}}|\boldsymbol{x}, \boldsymbol{z}) = \mathbb{E}_{\boldsymbol{y} \sim \pi_\theta(\boldsymbol{y}|\boldsymbol{x}, \boldsymbol{z})}\left[\mathbb{1}(\boldsymbol{y} \in \mathcal{Y}_{\boldsymbol{x}})\right],
\end{aligned} \tag{15}
$$

where $|\mathcal{Y}_{\boldsymbol{x}}|$ is the cardinal (number of elements) of $\mathcal{Y}_{\boldsymbol{x}}$, $\mathcal{U}(\mathcal{Y}_{\boldsymbol{x}})$ is the uniform distribution on $\mathcal{Y}_{\boldsymbol{x}}$, and $\mathbb{1}(\cdot)$ is the indicator function. When $|\mathcal{Y}_{\boldsymbol{x}}| = 1$, i.e., there is a unique correct answer expression

$\boldsymbol{y}^*$, Zhou et al. (2025) show that the likelihood-based estimator has lower variance (in fact, zero) compared to the accuracy-based one. We now extend this comparison to general cases when $|\mathcal{Y}_{\boldsymbol{x}}| > 1$:

$$
\begin{aligned}
\text{Var}_{\text{like}} &= |\mathcal{Y}_{\boldsymbol{x}}|^2 \cdot \text{Var}_{\boldsymbol{y} \sim \mathcal{U}(\mathcal{Y}_{\boldsymbol{x}})} \left[ \pi_\theta(\boldsymbol{y}|\boldsymbol{x}, \boldsymbol{z}) \right]; \\
\text{Var}_{\text{acc}} &= \pi_\theta(\mathcal{Y}_{\boldsymbol{x}}|\boldsymbol{x}, \boldsymbol{z}) \cdot (1 - \pi_\theta(\mathcal{Y}_{\boldsymbol{x}}|\boldsymbol{x}, \boldsymbol{z})) .
\end{aligned}
\tag{16}
$$

Note that the variance $\text{Var}_{\text{acc}}$ of accuracy-based estimator is independent of $|\mathcal{Y}_{\boldsymbol{x}}|$ and the model distribution $\pi_\theta$ over different elements in $\mathcal{Y}_{\boldsymbol{x}}$. Assuming that in the worst case where only one element $\boldsymbol{y}^* \in \mathcal{Y}_{\boldsymbol{x}}$ has non-zero probability under $\pi_\theta$, i.e., $\pi_\theta(\boldsymbol{y}^*|\boldsymbol{x}, \boldsymbol{z}) = \pi_\theta(\mathcal{Y}_{\boldsymbol{x}}|\boldsymbol{x}, \boldsymbol{z})$, we have

$$
\begin{aligned}
\text{Var}_{\text{like}}^{\text{worst}} &\triangleq \max_{\pi_\theta} \text{Var}_{\text{like}} = (|\mathcal{Y}_{\boldsymbol{x}}| - 1) \cdot \pi_\theta(\mathcal{Y}_{\boldsymbol{x}}|\boldsymbol{x}, \boldsymbol{z})^2; \\
\text{Var}_{\text{acc}}^{\text{worst}} &\triangleq \max_{\pi_\theta} \text{Var}_{\text{acc}} = \pi_\theta(\mathcal{Y}_{\boldsymbol{x}}|\boldsymbol{x}, \boldsymbol{z}) \cdot (1 - \pi_\theta(\mathcal{Y}_{\boldsymbol{x}}|\boldsymbol{x}, \boldsymbol{z})) .
\end{aligned}
\tag{17}
$$

Here we slightly abuse the notation of $\max_{\pi_\theta}$, since the maximization is taken w.r.t. all $\pi_\theta$ *under fixed value $\pi_\theta(\mathcal{Y}_{\boldsymbol{x}}|\boldsymbol{x}, \boldsymbol{z})$*. As seen, $\text{Var}_{\text{like}}^{\text{worst}} \geq \text{Var}_{\text{acc}}^{\text{worst}}$ holds when the model accuracy (condition on $\boldsymbol{x}, \boldsymbol{z}$) satisfies

$$
\pi_\theta(\mathcal{Y}_{\boldsymbol{x}}|\boldsymbol{x}, \boldsymbol{z}) \geq \frac{1}{|\mathcal{Y}_x|},
\tag{18}
$$

which almost always holds for $|\mathcal{Y}_x| \gg 1$. $\qquad\qquad\qquad\qquad\qquad\qquad\qquad\qquad\square$

## A.4 Derivation for Eq. (9)

Now we derive the gradient of the forward KL divergence $\mathbb{D}_{\text{KL}}(P_\theta(\boldsymbol{z}|\boldsymbol{x}, \mathcal{Y}_{\boldsymbol{x}})||q_\phi(\boldsymbol{z}|\boldsymbol{x}, \boldsymbol{y}'))$ w.r.t. $\phi$:

$$
\begin{aligned}
&\nabla_\phi \mathbb{D}_{\text{KL}}(P_\theta(\boldsymbol{z}|\boldsymbol{x}, \mathcal{Y}_{\boldsymbol{x}})||q_\phi(\boldsymbol{z}|\boldsymbol{x}, \boldsymbol{y}')) \\
&= -\nabla_\phi \mathbb{E}_{P_\theta(\boldsymbol{z}|\boldsymbol{x}, \mathcal{Y}_{\boldsymbol{x}})} \left[ \log q_\phi(\boldsymbol{z}|\boldsymbol{x}, \boldsymbol{y}') \right] \\
&= -\nabla_\phi \mathbb{E}_{\pi_\theta(\boldsymbol{z}|\boldsymbol{x})} \left[ \frac{\pi_\theta(\mathcal{Y}_{\boldsymbol{x}}|\boldsymbol{x}, \boldsymbol{z})}{P_\theta(\mathcal{Y}_{\boldsymbol{x}}|\boldsymbol{x})} \log q_\phi(\boldsymbol{z}|\boldsymbol{x}, \boldsymbol{y}') \right] \\
&\simeq \mathbb{E}_{\boldsymbol{z}_{1:M} \sim \pi_\theta(\boldsymbol{z}|\boldsymbol{x})} \left[ \sum_{m=1}^M \widetilde{w}_m \nabla_\phi \log q_\phi(\boldsymbol{z}_m|\boldsymbol{x}, \boldsymbol{y}') \right] \triangleq \nabla_\phi \mathcal{L}_{\text{forward}}^M, \\
&\text{where} \quad \widetilde{w}_m = \frac{w_m}{\sum_{j=1}^M w_j} \quad \text{and} \quad w_m = \pi_\theta(\mathcal{Y}_{\boldsymbol{x}}|\boldsymbol{x}, \boldsymbol{z}_m).
\end{aligned}
\tag{19}
$$

## A.5 Connection to more general RL reward shaping

In the literature on reinforcement learning with verifiable rewards (RLVR), various strategies for reward shaping have been proposed, many of which can be expressed as

$$
\mathcal{R}(\boldsymbol{x}, \boldsymbol{y}) = \begin{cases} \alpha & \text{if } \boldsymbol{y} \in \mathcal{Y}_{\boldsymbol{x}}; \\ \beta & \text{if } \boldsymbol{y} \notin \mathcal{Y}_{\boldsymbol{x}} \wedge \boldsymbol{y} \in \mathcal{Y}_{\text{format}}; \\ \gamma & \text{otherwise}, \end{cases}
\tag{20}
$$

where $\alpha, \beta, \gamma$ are hyperparameters, $\mathcal{Y}_{\text{format}}$ is the set of answers that correctly follow required format (e.g., \boxed{}) and is typically independent of $\boldsymbol{x}$. Apparently, $\mathcal{Y}_{\boldsymbol{x}} \subset \mathcal{Y}_{\text{format}}$ holds for any $\boldsymbol{x}$. Then we can derive the gradient of training objective under the general reward shaping $\mathcal{R}(\boldsymbol{x}, \boldsymbol{y})$ as

$$
\begin{aligned}
&\nabla_\theta \mathcal{L}_{\text{general-RL}}(\boldsymbol{x}, \pi_\theta) \\
&\triangleq \nabla_\theta \mathbb{E}_{\pi_\theta(\boldsymbol{z}|\boldsymbol{x})} \mathbb{E}_{\pi_\theta(\boldsymbol{y}|\boldsymbol{x}, \boldsymbol{z})} \left[ \mathcal{R}(\boldsymbol{x}, \boldsymbol{y}) \right] \\
&= \mathbb{E}_{\pi_\theta(\boldsymbol{z}|\boldsymbol{x})} \mathbb{E}_{\pi_\theta(\boldsymbol{y}|\boldsymbol{x}, \boldsymbol{z})} \left[ \mathcal{R}(\boldsymbol{x}, \boldsymbol{y}) \cdot (\nabla_\theta \log \pi_\theta(\boldsymbol{z}|\boldsymbol{x}) + \nabla_\theta \log \pi_\theta(\boldsymbol{y}|\boldsymbol{x}, \boldsymbol{z})) \right] \\
&\xrightarrow[\pi_\theta(\boldsymbol{z}|\boldsymbol{x})]{\text{only w.r.t.}} \mathbb{E}_{\pi_\theta(\boldsymbol{z}|\boldsymbol{x})} \mathbb{E}_{\pi_\theta(\boldsymbol{y}|\boldsymbol{x}, \boldsymbol{z})} \left[ \mathcal{R}(\boldsymbol{x}, \boldsymbol{y}) \cdot \nabla_\theta \log \pi_\theta(\boldsymbol{z}|\boldsymbol{x}) \right] \\
&= \mathbb{E}_{\pi_\theta(\boldsymbol{z}|\boldsymbol{x})} \left[ ((\alpha - \beta) \cdot \pi_\theta(\mathcal{Y}_{\boldsymbol{x}}|\boldsymbol{x}, \boldsymbol{z}) + (\beta - \gamma) \cdot \pi_\theta(\mathcal{Y}_{\text{format}}|\boldsymbol{x}, \boldsymbol{z})) \cdot \nabla_\theta \log \pi_\theta(\boldsymbol{z}|\boldsymbol{x}) \right] \\
&= (\alpha - \beta) \cdot P_\theta(\mathcal{Y}_{\boldsymbol{x}}|\boldsymbol{x}) \cdot \nabla_\theta \mathbb{D}_{\text{KL}}(P_\theta^{\text{sg}}(\boldsymbol{z}|\boldsymbol{x}, \mathcal{Y}_{\boldsymbol{x}})||\pi_\theta(\boldsymbol{z}|\boldsymbol{x})) \\
&\quad + (\beta - \gamma) \cdot P_\theta(\mathcal{Y}_{\text{format}}|\boldsymbol{x}) \cdot \nabla_\theta \mathbb{D}_{\text{KL}}(P_\theta^{\text{sg}}(\boldsymbol{z}|\boldsymbol{x}, \mathcal{Y}_{\text{format}})||\pi_\theta(\boldsymbol{z}|\boldsymbol{x})),
\end{aligned}
\tag{21}
$$

where $P_\theta(\mathcal{Y}_{\text{format}}|\boldsymbol{x})$ is the probability that the output answers follow the required format. It is easy to know that the optimal solution for Eq. (21) can be written as:

$$\pi_\theta^*(\boldsymbol{z}|\boldsymbol{x}) = \frac{(\alpha - \beta) \cdot P_\theta^{\text{sg}}(\boldsymbol{z}, \mathcal{Y}_{\boldsymbol{x}}|\boldsymbol{x}) + (\beta - \gamma) \cdot P_\theta^{\text{sg}}(\boldsymbol{z}, \mathcal{Y}_{\text{format}}|\boldsymbol{x})}{(\alpha - \beta) \cdot P_\theta^{\text{sg}}(\mathcal{Y}_{\boldsymbol{x}}|\boldsymbol{x}) + (\beta - \gamma) \cdot P_\theta^{\text{sg}}(\mathcal{Y}_{\text{format}}|\boldsymbol{x})}. \tag{22}$$

**Remark.** When $\beta = \gamma$, i.e., there is no format reward, the optimization problem in Eq. (21) degrades to Eq. (11). When $\alpha > \beta > \gamma$, the model $\pi_\theta(\boldsymbol{z}|\boldsymbol{x})$ will tend to hack reward function on hard problems (i.e., low $P_\theta(\mathcal{Y}_{\boldsymbol{x}}|\boldsymbol{x})$) that can easily follow format (i.e., high $P_\theta(\mathcal{Y}_{\text{format}}|\boldsymbol{x})$), where $\pi_\theta(\boldsymbol{z}|\boldsymbol{x})$ will seek modes of $P_\theta^{\text{sg}}(\boldsymbol{z}|\boldsymbol{x}, \mathcal{Y}_{\text{format}})$. Besides, there may be an intuition that setting $\beta < 0$ could alleviate reward hacking, however, as shown in Eq. (21), the optimization only depends on the relative values of $\alpha - \beta$ and $\beta - \gamma$.

Now we show that it is straightforward to debias $P_\theta(\mathcal{Y}_{\boldsymbol{x}}|\boldsymbol{x})$ and $P_\theta(\mathcal{Y}_{\text{format}}|\boldsymbol{x})$ in Eq. (21). Specifically, we can rewrite the reward function as (note that reward functions are equivalent up to any constant):

$$\mathcal{R}(\boldsymbol{x}, \boldsymbol{y}) = (\alpha - \beta) \cdot \mathbb{1}(\boldsymbol{y} \in \mathcal{Y}_{\boldsymbol{x}}) + (\beta - \gamma) \cdot \mathbb{1}(\boldsymbol{y} \in \mathcal{Y}_{\text{format}}). \tag{23}$$

Then the debiased version of reward function is

$$\mathcal{R}^{\ddagger}(\boldsymbol{x}, \boldsymbol{y}) = \frac{(\alpha - \beta)}{P_\theta^{\text{sg}}(\mathcal{Y}_{\boldsymbol{x}}|\boldsymbol{x})} \cdot \mathbb{1}(\boldsymbol{y} \in \mathcal{Y}_{\boldsymbol{x}}) + \frac{(\beta - \gamma)}{P_\theta^{\text{sg}}(\mathcal{Y}_{\text{format}}|\boldsymbol{x})} \cdot \mathbb{1}(\boldsymbol{y} \in \mathcal{Y}_{\text{format}}), \tag{24}$$

where in practice $P_\theta^{\text{sg}}(\mathcal{Y}_{\boldsymbol{x}}|\boldsymbol{x})$ and $P_\theta^{\text{sg}}(\mathcal{Y}_{\text{format}}|\boldsymbol{x})$ can be approximated by the ratio of correct answers (i.e., model accuracy) and the ratio of correct format for each batch of RL rollouts (larger rollout number could lead to more accurate estimation). After using the debiased reward function $\mathcal{R}^{\ddagger}(\boldsymbol{x}, \boldsymbol{y})$, the optimal solution of $\pi_\theta^*(\boldsymbol{z}|\boldsymbol{x})$ becomes

$$\pi_\theta^*(\boldsymbol{z}|\boldsymbol{x}) = \frac{(\alpha - \beta) \cdot P_\theta^{\text{sg}}(\boldsymbol{z}|\boldsymbol{x}, \mathcal{Y}_{\boldsymbol{x}}) + (\beta - \gamma) \cdot P_\theta^{\text{sg}}(\boldsymbol{z}|\boldsymbol{x}, \mathcal{Y}_{\text{format}})}{\alpha - \gamma}. \tag{25}$$

## A.6 SPECIAL CASES IN EQ. (6)

**Special case I:** $q_\phi(\boldsymbol{z}|\boldsymbol{x}, \boldsymbol{y}') = \pi_\theta^{\text{sg}}(\boldsymbol{z}|\boldsymbol{x})$. In this case, we can simplify the gradient estimation as

$$\nabla_\theta \mathcal{L}_{\text{ELBO}}^K(\boldsymbol{x}, \mathcal{Y}_{\boldsymbol{x}}; \pi_\theta, \pi_\theta^{\text{sg}}) = \mathbb{E}_{\boldsymbol{z}_{1:K} \sim \pi_\theta(\boldsymbol{z}|\boldsymbol{x})} \left[ \sum_{k=1}^K \widetilde{w}_k \nabla_\theta \log \pi_\theta(\boldsymbol{z}_k, \mathcal{Y}_{\boldsymbol{x}}|\boldsymbol{x}) \right]$$
$$\text{where} \quad \widetilde{w}_k = \frac{w_k}{\sum_{j=1}^K w_j} \quad \text{and} \quad w_k = \pi_\theta(\mathcal{Y}_{\boldsymbol{x}}|\boldsymbol{x}, \boldsymbol{z}_k), \tag{26}$$

which can be regarded as a normalized version of VeriFree (Zhou et al., 2025).

**Special case II:** $K = 1$. In this case $\mathcal{L}_{\text{ELBO}}^1 = \mathcal{L}_{\text{ELBO}}$ and we can simplify the gradient estimation as

$$\nabla_\theta \mathcal{L}_{\text{ELBO}}(\boldsymbol{x}, \mathcal{Y}_{\boldsymbol{x}}, \boldsymbol{y}'; \pi_\theta, q_\phi) = \mathbb{E}_{\boldsymbol{z} \sim q_\phi(\boldsymbol{z}|\boldsymbol{x}, \boldsymbol{y}')} \left[ \nabla_\theta \log \pi_\theta(\boldsymbol{z}, \mathcal{Y}_{\boldsymbol{x}}|\boldsymbol{x}) \right];$$
$$\nabla_\phi \mathcal{L}_{\text{ELBO}}(\boldsymbol{x}, \mathcal{Y}_{\boldsymbol{x}}, \boldsymbol{y}'; \pi_\theta, q_\phi) = \mathbb{E}_{\boldsymbol{z} \sim q_\phi(\boldsymbol{z}|\boldsymbol{x}, \boldsymbol{y}')} \left[ \left( \log \frac{\pi_\theta(\boldsymbol{z}, \mathcal{Y}_{\boldsymbol{x}}|\boldsymbol{x})}{q_\phi(\boldsymbol{z}|\boldsymbol{x}, \boldsymbol{y}')} \right) \cdot \nabla_\phi \log q_\phi(\boldsymbol{z}|\boldsymbol{x}, \boldsymbol{y}') \right]. \tag{27}$$

## A.7 MORE DERIVATIONS FOR EQ. (11)

Now we investigate the gradient of binary reward RL w.r.t. $\pi_\theta(\boldsymbol{y}|\boldsymbol{x}, \boldsymbol{z})$:

$$\nabla_\theta \mathcal{L}_{\text{bi-RL}}(\boldsymbol{x}, \pi_\theta) \triangleq \nabla_\theta \mathbb{E}_{\pi_\theta(\boldsymbol{z}|\boldsymbol{x})} \mathbb{E}_{\pi_\theta(\boldsymbol{y}|\boldsymbol{x}, \boldsymbol{z})} \left[ \mathbb{1}(\boldsymbol{y} \in \mathcal{Y}_{\boldsymbol{x}}) \right]$$
$$= \mathbb{E}_{\pi_\theta(\boldsymbol{z}|\boldsymbol{x})} \mathbb{E}_{\pi_\theta(\boldsymbol{y}|\boldsymbol{x}, \boldsymbol{z})} \left[ \mathbb{1}(\boldsymbol{y} \in \mathcal{Y}_{\boldsymbol{x}}) \cdot (\nabla_\theta \log \pi_\theta(\boldsymbol{z}|\boldsymbol{x}) + \nabla_\theta \log \pi_\theta(\boldsymbol{y}|\boldsymbol{x}, \boldsymbol{z})) \right]$$
$$\xrightarrow[\pi_\theta(\boldsymbol{y}|\boldsymbol{x}, \boldsymbol{z})]{\text{only w.r.t.}} \mathbb{E}_{\pi_\theta(\boldsymbol{z}|\boldsymbol{x})} \mathbb{E}_{\pi_\theta(\boldsymbol{y}|\boldsymbol{x}, \boldsymbol{z})} \left[ \mathbb{1}(\boldsymbol{y} \in \mathcal{Y}_{\boldsymbol{x}}) \cdot \nabla_\theta \log \pi_\theta(\boldsymbol{y}|\boldsymbol{x}, \boldsymbol{z}) \right]$$
$$= \mathbb{E}_{\pi_\theta(\boldsymbol{z}|\boldsymbol{x})} \left[ \nabla_\theta \pi_\theta(\mathcal{Y}_{\boldsymbol{x}}|\boldsymbol{x}, \boldsymbol{z}) \right], \tag{28}$$

where the optimal solution is straightforward that $\forall \boldsymbol{z}$, there is $\pi_\theta^*(\mathcal{Y}_{\boldsymbol{x}}|\boldsymbol{x}, \boldsymbol{z}) = 1$. However, this optimal solution is usually unachievable, since it requires the model to return $100\%$ correct answers independent of the thinking process $\boldsymbol{z}$.

## A.8 APPLYING VARIATIONAL POSTERIOR FOR RL

Recall that the objective function of binary reward RL is defined as

$$
\begin{aligned}
\mathcal{L}_{\text{bi-RL}}(\boldsymbol{x}, \pi_\theta) &\triangleq \mathbb{E}_{\pi_\theta(\boldsymbol{z}|\boldsymbol{x})} \mathbb{E}_{\pi_\theta(\boldsymbol{y}|\boldsymbol{x},\boldsymbol{z})} \left[ \mathbb{1}(\boldsymbol{y} \in \mathcal{Y}_{\boldsymbol{x}}) \right] \\
&= \mathbb{E}_{\pi_\theta(\boldsymbol{z}|\boldsymbol{x})} \left[ \pi_\theta(\mathcal{Y}_{\boldsymbol{x}}|\boldsymbol{x}, \boldsymbol{z}) \right].
\end{aligned}
\tag{29}
$$

Suppose the data points are drawn from a behavior policy $q(\boldsymbol{z}|\boldsymbol{x})$, the RL objective can be reformulated using an importance sampling correction term as follows:

$$
\mathcal{L}_{\text{bi-RL}}(\boldsymbol{x}, \pi_\theta) = \mathbb{E}_{q(\boldsymbol{z}|\boldsymbol{x})} \left[ \frac{\pi_\theta(\boldsymbol{z}|\boldsymbol{x})}{q(\boldsymbol{z}|\boldsymbol{x})} \pi_\theta(\mathcal{Y}_{\boldsymbol{x}}|\boldsymbol{x}, \boldsymbol{z}) \right].
\tag{30}
$$

Then, a natural question arises: *what is the optimal behavior policy $q(\boldsymbol{z}|\boldsymbol{x})$ that minimizes the variance of estimating $\mathcal{L}_{bi\text{-}RL}(\boldsymbol{x}, \pi_\theta)$?* Specifically, we can compute

$$
\begin{aligned}
\text{Var}_{q(\boldsymbol{z}|\boldsymbol{x})} \left[ \frac{\pi_\theta(\boldsymbol{z}|\boldsymbol{x})}{q(\boldsymbol{z}|\boldsymbol{x})} \pi_\theta(\mathcal{Y}_{\boldsymbol{x}}|\boldsymbol{x}, \boldsymbol{z}) \right] &= \text{Var}_{q(\boldsymbol{z}|\boldsymbol{x})} \left[ \frac{P_\theta(\boldsymbol{z}|\boldsymbol{x}, \mathcal{Y}_{\boldsymbol{x}})}{q(\boldsymbol{z}|\boldsymbol{x})} P_\theta(\mathcal{Y}_{\boldsymbol{x}}|\boldsymbol{x}) \right] \\
&= P_\theta(\mathcal{Y}_{\boldsymbol{x}}|\boldsymbol{x})^2 \cdot \text{Var}_{q(\boldsymbol{z}|\boldsymbol{x})} \left[ \frac{P_\theta(\boldsymbol{z}|\boldsymbol{x}, \mathcal{Y}_{\boldsymbol{x}})}{q(\boldsymbol{z}|\boldsymbol{x})} \right] \\
&= P_\theta(\mathcal{Y}_{\boldsymbol{x}}|\boldsymbol{x})^2 \cdot \left( \sum_{\boldsymbol{z}} \frac{P_\theta(\boldsymbol{z}|\boldsymbol{x}, \mathcal{Y}_{\boldsymbol{x}})^2}{q(\boldsymbol{z}|\boldsymbol{x})} - 1 \right),
\end{aligned}
\tag{31}
$$

which is equivalent to minimizing

$$
\min_{q(\boldsymbol{z}|\boldsymbol{x})} \sum_{\boldsymbol{z}} \frac{P_\theta(\boldsymbol{z}|\boldsymbol{x}, \mathcal{Y}_{\boldsymbol{x}})^2}{q(\boldsymbol{z}|\boldsymbol{x})} \quad \text{s.t.} \quad \sum_{\boldsymbol{z}} q(\boldsymbol{z}|\boldsymbol{x}) = 1, \, q(\boldsymbol{z}|\boldsymbol{x}) \geq 0.
\tag{32}
$$

Using calculus of variations with a Lagrange multiplier $\lambda$, we obtain

$$
\begin{aligned}
\delta \left[ \sum_{\boldsymbol{z}} \frac{P_\theta(\boldsymbol{z}|\boldsymbol{x}, \mathcal{Y}_{\boldsymbol{x}})^2}{q(\boldsymbol{z}|\boldsymbol{x})} + \lambda \left( \sum_{\boldsymbol{z}} q(\boldsymbol{z}|\boldsymbol{x}) - 1 \right) \right] &= 0 \\
\Rightarrow q^*(\boldsymbol{z}|\boldsymbol{x}) = P_\theta(\boldsymbol{z}|\boldsymbol{x}, \mathcal{Y}_{\boldsymbol{x}}) \text{ and } \lambda &= 1 \\
\Rightarrow P_\theta(\boldsymbol{z}|\boldsymbol{x}, \mathcal{Y}_{\boldsymbol{x}}) = \arg\min_{q(\boldsymbol{z}|\boldsymbol{x})} \text{Var}_{q(\boldsymbol{z}|\boldsymbol{x})} &\left[ \frac{\pi_\theta(\boldsymbol{z}|\boldsymbol{x})}{q(\boldsymbol{z}|\boldsymbol{x})} \pi_\theta(\mathcal{Y}_{\boldsymbol{x}}|\boldsymbol{x}, \boldsymbol{z}) \right].
\end{aligned}
\tag{33}
$$

Therefore, we show that optimizing the variational posterior $q_\phi$ to approximate the true posterior $P_\theta(\boldsymbol{z}|\boldsymbol{x}, \mathcal{Y}_{\boldsymbol{x}})$ in Eq. (9) naturally yields an (approximately) optimal behavior policy for RL, one that minimizes the variance of the objective estimator. In practice, the trained variational posterior $q_\phi$ can thus be employed as the behavior policy to reduce variance, which is fully compatible with actor-critic frameworks that incorporate advantage estimation with baselines.

## A.9 CONNECTION TO REINFORCEMENT LEARNING TEACHERS

In the derivation of our method, the ELBO objective in Eq. (4) minimizes *reverse* KL divergence $\mathbb{D}_{\text{KL}}(q_\phi(\boldsymbol{z}|\boldsymbol{x}, \boldsymbol{y}')||P_\theta(\boldsymbol{z}|\boldsymbol{x}, \mathcal{Y}_{\boldsymbol{x}}))$. As analyzed in Section 2.3, we propose to optimize $q_\phi(\boldsymbol{z}|\boldsymbol{x}, \boldsymbol{y}')$ using the *forward* KL divergence $\mathbb{D}_{\text{KL}}(P_\theta(\boldsymbol{z}|\boldsymbol{x}, \mathcal{Y}_{\boldsymbol{x}})||q_\phi(\boldsymbol{z}|\boldsymbol{x}, \boldsymbol{y}'))$, which shares the same optimal solution.

Alternatively, we can also optimize the reverse KL divergence by policy gradient method as follows:

$$
\begin{aligned}
&\nabla_\phi \mathcal{L}_{\text{ELBO}} \\
&= \nabla_\phi \mathbb{D}_{\text{KL}} (q_\phi(\boldsymbol{z}|\boldsymbol{x}, \boldsymbol{y}')||P_\theta(\boldsymbol{z}|\boldsymbol{x}, \mathcal{Y}_{\boldsymbol{x}})) \\
&= \nabla_\phi \mathbb{E}_{q_\phi(\boldsymbol{z}|\boldsymbol{x}, \boldsymbol{y}')}[\log \pi_\theta(\boldsymbol{z}, \mathcal{Y}_{\boldsymbol{x}}|\boldsymbol{x}) - \log q_\phi(\boldsymbol{z}|\boldsymbol{x}, \boldsymbol{y}')] \\
&= \mathbb{E}_{q_\phi(\boldsymbol{z}|\boldsymbol{x}, \boldsymbol{y}')} \Big[ \log \pi_\theta(\boldsymbol{z}, \mathcal{Y}_{\boldsymbol{x}}|\boldsymbol{x}) \nabla_\phi \log q_\phi(\boldsymbol{z}|\boldsymbol{x}, \boldsymbol{y}') - \log q_\phi(\boldsymbol{z}|\boldsymbol{x}, \boldsymbol{y}') \nabla_\phi \log q_\phi(\boldsymbol{z}|\boldsymbol{x}, \boldsymbol{y}') \Big] \\
&= \mathbb{E}_{q_\phi(\boldsymbol{z}|\boldsymbol{x}, \boldsymbol{y}')} \left[ \underbrace{\left( \log \pi_\theta(\mathcal{Y}_{\boldsymbol{x}}|\boldsymbol{x}, \boldsymbol{z}) - \log \frac{q_\phi(\boldsymbol{z}|\boldsymbol{x}, \boldsymbol{y}')}{\pi_\theta(\boldsymbol{z}|\boldsymbol{x})} \right)}_{\text{reward}} \nabla_\phi \log q_\phi(\boldsymbol{z}|\boldsymbol{x}, \boldsymbol{y}') \right].
\end{aligned}
\tag{34}
$$

More concisely, the reverse KL divergence can be alternatively minimized via reinforcement learning using $\log \pi_\theta(\mathcal{Y}_{\boldsymbol{x}}|\boldsymbol{x}, \boldsymbol{z}) - \log \frac{q_\phi(\boldsymbol{z}|\boldsymbol{x}, \boldsymbol{y}')}{\pi_\theta(\boldsymbol{z}|\boldsymbol{x})}$ as the reward function.

This derivation establishes a connection to Reinforcement Learning Teachers (RLTs) (Cetin et al., 2025), who focus on training reasoning LLMs to act as teachers for distilling new students. Their approach introduces RLTs optimized specifically for effective student distillation. RLTs are trained by GRPO using dense rewards obtained by feeding each explanation to the student and evaluating its understanding of the solution.

Specifically, the dense reward in RLT combines two components: one measuring the student's likelihood of reaching the correct solution (analogous to $\log \pi_\theta(\mathcal{Y}_{\boldsymbol{x}}|\boldsymbol{x}, \boldsymbol{z})$), and another regularizing the teacher's explanation to remain coherent from the student's perspective given only its prior knowledge and the question (analogous to $-\log \frac{q_\phi(\boldsymbol{z}|\boldsymbol{x}, \boldsymbol{y}')}{\pi_\theta(\boldsymbol{z}|\boldsymbol{x})}$).

While RLT employs an intuitively designed reward, our work provides rigorous theoretical justification from a variational inference perspective. Furthermore, we enhance the method with a tighter IWAE-style lower bound and an accuracy-based estimator, as detailed in Section 2.2.

## B    RELATED WORK

**SFT and RL methods for reasoning.** Reasoning has emerged as a central capability of LLMs, driving advances in domains such as mathematics, programming, and scientific discovery (Jaech et al., 2024; Comanici et al., 2025; Team et al., 2025). Among the approaches developed to strengthen these abilities, SFT and RL have become the two dominant paradigms (Uesato et al., 2022; Rafailov et al., 2023; Guha et al., 2025; Hu et al., 2025; Hochlehnert et al., 2025). Building on the DeepSeek-R1 framework (Shao et al., 2024; Guo et al., 2025), a range of new RL algorithms have been proposed, including Dr. GRPO (Liu et al., 2025), DAPO (Yu et al., 2025a), REINFORCE++ (Hu, 2025), VinePPO (Kazemnejad et al., 2024), and VAPO (Yue et al., 2025b). In parallel, extensive empirical studies have explored the design space of RL for reasoning (Zeng et al., 2025; Team et al., 2025), focusing on dimensions such as curriculum learning (Wen et al., 2025; Luo et al., 2025) and reward design (Gao et al., 2024; Cui et al., 2025a; Ma et al., 2023; Qi et al., 2025). While early progress has centered on mathematical reasoning, recent work has extended RL-based methods to code and software engineering tasks (Liu & Zhang, 2025; Xie et al., 2025; Wei et al., 2025; Yang et al., 2025; Chen et al., 2025; Li et al., 2025b), as well as to agentic problem-solving scenarios (Wang et al., 2025; Jin et al., 2025; Jiang et al., 2025; Xue et al., 2025).

**Decomposing thinking and answering processes.** Traditional studies on LLM reasoning ability often treat model responses holistically. In contrast, a recent line of research explicitly decomposes the LLM response into a thinking trace $\boldsymbol{z}$ and a final answer $\boldsymbol{y}$, given a question $\boldsymbol{x}$ (Chen et al., 2024; Xiang et al., 2025; Zhou et al., 2025; Zhong et al., 2025). This decomposition offers several novel and useful perspectives.

Zhou et al. (2025) propose VeriFree, which directly optimizes $P_\theta(\boldsymbol{y}|\boldsymbol{x}) = \mathbb{E}_{\pi_\theta(\boldsymbol{z}|\boldsymbol{x})}[\pi_\theta(\boldsymbol{y}|\boldsymbol{x}, \boldsymbol{z})]$ using policy gradient. Their algorithm simultaneously optimizes $\pi_\theta(\boldsymbol{z}|\boldsymbol{x})$ via policy gradient methods with $\pi_\theta(\boldsymbol{y}|\boldsymbol{x}, \boldsymbol{z})$ as a reward, and performs weighted SFT on $\pi_\theta(\boldsymbol{y}|\boldsymbol{x}, \boldsymbol{z})$. This approach demonstrates strong performance in general domains where rule-based verifiers are typically unavailable. Subsequent works (Yu et al., 2025b; Xu et al., 2025) further improve upon this by reshaping the reward, e.g., intuitively replacing the product of token probabilities with the mean when computing $\pi_\theta(\boldsymbol{y}|\boldsymbol{x}, \boldsymbol{z})$.

Chen et al. (2024) introduce LaTRO, formulating reasoning as sampling from a latent distribution $q(\boldsymbol{z}|\boldsymbol{x})$ and optimizing $\log P_\theta(\boldsymbol{y}|\boldsymbol{x})$ via a variational manner. Their derived lower bound is:
$$\log P_\theta(\boldsymbol{y}|\boldsymbol{x}) \geq \mathbb{E}_{q(\boldsymbol{z}|\boldsymbol{x})}\left[\log \pi_\theta(\boldsymbol{y}|\boldsymbol{x}, \boldsymbol{z})\right] - \mathbb{D}_{\mathrm{KL}}\left(q(\boldsymbol{z}|\boldsymbol{x}) \| \pi_\theta(\boldsymbol{z}|\boldsymbol{x})\right).$$
They set the proposal distribution $q(\boldsymbol{z}|\boldsymbol{x})$ to $\pi_\theta(\boldsymbol{z}|\boldsymbol{x})$, resulting in a reinforcement learning algorithm where $\log \pi_\theta(\boldsymbol{y}|\boldsymbol{x}, \boldsymbol{z})$ serves as the reward. Tang et al. (2025) and Ruan et al. (2025) tighten this bound using ideas similar to IWAE (Burda et al., 2015).

A more natural choice for the variational distribution is the true posterior $P_\theta(\boldsymbol{z}|\boldsymbol{x}, \boldsymbol{y})$, though it is intractable. Hoffman et al. (2023) use MCMC to sample from the posterior, while Hu et al. (2024) employ GFlowNets (Bengio et al., 2023) to fine-tune an LLM to approximate it. Both methods use an EM-like algorithm to optimize the ELBO of $\log P_\theta(\boldsymbol{y}|\boldsymbol{x})$.

Our approach uses forward KL divergence to train a variational posterior and derives a novel objective based on a tighter IWAE-style bound. Additionally, we propose an accuracy-based estimator for $\pi_\theta(\mathcal{Y}_x|x, z_k)$, instead of the likelihood-based estimator used in Zhou et al. (2025). We also build connections to other mainstream finetuning algorithms that enhance reasoning, such as RFT and GRPO.

**Reinforcement learning as probabilistic inference.** Previous works have also explored connections between reinforcement learning and probabilistic inference. Notably, Levine (2018) discuss RL and control from a probabilistic inference perspective. Their approach begins from a reinforcement learning standpoint, where the goal is to search for an optimal policy, and constructs a probabilistic graphical model (PGM) in which the posterior distribution over actions corresponds to an optimal policy. This is achieved by defining observations in the PGM based on rewards, and the resulting inference problem is then solved via variational inference. This formulation differs from our approach.

In contrast, our method starts from the objective of maximizing $\log P_\theta(\mathcal{Y}_x|x)$, treats the reasoning process as a discrete latent variable $z$, and optimizes the ELBO to train the reasoning model. We subsequently draw connections between our framework and other RL algorithms (Section 3). From an application perspective, our work focuses on the language domain and explicitly models reasoning traces as discrete latent variables, which further distinguishes it from the aforementioned work.

## C   TRAINING DETAILS

In this section, we detail the training procedure used in our method. Our framework builds on LLaMA-Factory (Zheng et al., 2024). By default, SFT averages token-level cross-entropy over all valid tokens in a batch. However, as shown in Section 2, our variables $z$ and $y$ are defined at the sentence level. To align with this, we modify the objective: instead of normalizing by the number of valid tokens, we sum the loss over all tokens and divide by a constant equal to the average response length in the training set (precomputed offline). This change parallels the difference between GRPO (Shao et al., 2024) and Dr. GRPO (Liu et al., 2025), thus we name this slight modification as **Dr. SFT**. Both our models and Bespoke-Stratos-4B/8B[†] are trained with this modified objective. We further extend the framework with weighted SFT, as the original LLaMA-Factory does not support weighting. This feature is essential for parts of our method that require weighted training. All experiments are conducted on NVIDIA H100 GPUs.

### C.1   TRAINING THE INITIAL REASONING MODEL AND VARIATIONAL POSTERIOR

For training, we use the following settings:

1. **Initial reasoning model $\pi_{\theta_0}$ (following Labs (2025)):**
   - `batch_size=96`, `cutoff_len=16384`
   - Optimizer: AdamW (Loshchilov & Hutter, 2019) with `adam_beta1=0.9`, `adam_beta2=0.999`, `adam_epsilon=1.0e-8`, `weight_decay=0`
   - Learning rate schedule: `cosine` with `warmup_ratio=0.1`
   - `learning_rate=1.0e-5`, `max_grad_norm=1.0`
   - Training for 3 epochs
   - Precision: `bfloat16`
   - Baselines (Bespoke-Stratos-7B/32B and Bespoke-Stratos-4B/8B[†]) are trained with the same setup

2. **Variational posterior $q_\phi$:**
   - `batch_size=16`
   - Optimizer: AdamW with `adam_beta1=0.9`, `adam_beta2=0.95`, `adam_epsilon=1.0e-8`, `weight_decay=1.0e-4`
   - `warmup_ratio=0.05`
   - Training for 10 epochs
   - All other hyperparameters follow those used for $\pi_{\theta_0}$

## C.2 DETAILS OF SAMPLING FROM VARIATIONAL POSTERIOR

Using the trained variational posterior $q_\phi$, we sample 8 reasoning traces (including final answers) for each question in Bespoke-Stratos-17k with vLLM (Kwon et al., 2023), using `temperature=0.7`, `top_p=1.0`, `top_k=-1`, `max_tokens=32764`, and `dtype=bfloat16`. After obtaining the sampled reasoning traces, we compute the importance weight $\widetilde{\rho}_k$ for each question-thinking-answer triplet using the pre-trained initial reasoning model $\pi_{\theta_0}$ and the variational posterior $q_\phi$. This is done in forward mode by evaluating the log-likelihoods under both models, without requiring backpropagation, which is efficient.

To estimate the term $\pi_\theta(\mathcal{Y}_{\boldsymbol{x}} \mid \boldsymbol{x}, \boldsymbol{z})$ used in $\widetilde{\rho}_k$, we adopt an *accuracy-based estimator*. Specifically, for each question and each sampled thinking trace, we use $\pi_\theta$ to generate 8 answers under the same sampling configuration as above. The correctness of these answers is evaluated using the math/code verifiers from SkyThought, and the average accuracy is taken as the estimate of $\pi_\theta(\mathcal{Y}_{\boldsymbol{x}} \mid \boldsymbol{x}, \boldsymbol{z})$. Experiments utilizing the accuracy-based estimator are labeled as **"-Acc"**.

Additionally, we employ the geometric mean of token-level probabilities under $\pi_\theta(\mathcal{Y}_{\boldsymbol{x}} \mid \boldsymbol{x}, \boldsymbol{z})$ as an alternative and intuitive estimator. This approach mitigates the inherent length bias present in the strict definition of $\pi_\theta(\mathcal{Y}_{\boldsymbol{x}} \mid \boldsymbol{x}, \boldsymbol{z})$, which computes the product of token probabilities and consequently assigns excessively small values to longer reasoning traces. This estimator provides an intuitive approximation without requiring an external verifier. Experiments utilizing the estimator based on geometric mean are labeled as **"-GML"**, while those with the naive estimator are labeled as **"-L"**.

This process results in a weighted dataset where each sample consists of a question-thinking-answer triplet along with its corresponding weight $\widetilde{\rho}_k$, which will be utilized in subsequent training stages.

## C.3 DETAILS OF TRAINING FINAL REASONING MODEL

To train the final reasoning model $\pi_\theta$, we adopt the following procedure. For the 17k data setting, we select, for each question, the reasoning trace with the highest importance weight $\widetilde{\rho}_k$ among the 8 samples sampled from the variational posterior. In experiments using the accuracy-based estimator, we pair the selected reasoning trace with a randomly chosen verified answer generated by the initial reasoning model. For other estimators, the original answer from the dataset is retained. The resulting synthetic data is then mixed with the original Bespoke-Stratos-17k dataset.

We maintain the same training configuration as used for the initial reasoning model, with one exception: the batch size is increased to `batch_size=192`. This adjustment ensures that the total number of optimization steps remains consistent with baseline models (e.g., Bespoke-Stratos-32B), as the mixed dataset is twice the size of the original.

For the 1k data setting, the baseline model (e.g., Bespoke-Stratos-7B-1K[†]) is trained with the following configuration: We adopt `batch_size=16` and `cutoff_len=32768`. We use the AdamW optimizer (Loshchilov & Hutter, 2019) with parameters `adam_beta1=0.9`, `adam_beta2=0.95`, `adam_epsilon=1.0e-8`, and `weight_decay=1.0e-4`. A `cosine` learning rate schedule is applied with `warmup_ratio=0.1`, alongside a learning rate of `1.0e-5` and gradient clipping at `max_grad_norm=1.0`. Training is conducted for 5 epochs.

In our method under the 1k setting, we do not combine with the original dataset. Instead, we use all 8 reasoning traces, weighted by $\widetilde{\rho}_k$, which is faithful to Algorithm 1. To match the number of optimization steps in the baseline, we proportionally adjust the batch size while keeping all other hyperparameters unchanged.

## D DETAILS OF EVALUATION

We conduct all evaluations using SkyThought, specifically at commit `0d190f1`.[3] Team (2025b) suggest avoiding greedy decoding for models with long thinking traces. Thus, responses are sampled from the models using `temperature=0.7` and `top_p=1.0`. A generous token budget of `max_tokens=38912` is allocated to accommodate lengthy outputs.

---

[3] https://github.com/NovaSky-AI/SkyThought

To maximize reproducibility, we perform inference using `dtype=float32`, accepting a potential decrease in speed for improved consistency. For model parallelism, we configure `tensor_parallel_size=4` for 4B/7B/8B models and `tensor_parallel_size=8` for the 32B models. We choose vLLM (Kwon et al., 2023) as the inference backend. For models based on Qwen2.5, we use `vllm==0.7.0`, while for Qwen3-based models, we use `vllm==0.8.4`. Although we anticipate that these version differences have negligible impact on evaluation accuracy, we document them here to ensure full reproducibility.

## E  PROMPT TEMPLATES

In Section 2, we abstractly introduced how we define the prompt patterns used in the reasoning model $\pi_\theta$ and variational posterior $q_\phi$. In this section, we provide details of the prompt templates used in practice, as shown below.

> **Prompt template A (PA) for variational posterior $q_\phi$**
>
> Your role as an assistant involves providing precise and accurate solutions before providing detailed explanations with your full work showing your systematic thinking process leading to each solution. Your explanations should show how you engaged in a comprehensive cycle of analysis, summarizing, exploration, reassessment, reflection, backtracing, and iteration to develop well-considered thinking process. Please structure your response into two main sections: Solution and Explanation. In the Solution section, present your well-thought solution that accurately answers the question. The solution should remain a logical, accurate, concise expression style and detail necessary step needed to reach the conclusion, formatted as follows: <|begin_of_solution|> {final formatted, precise, and clear solution} <|end_of_solution|>. In the Explanation section, comprehensively detail your reasoning process using the specified format: <|begin_of_explanation|> {explanation with steps separated with '\\n\\n'} <|end_of_explanation|> Each step should show detailed considerations leading to your solutions such as analisying questions, summarizing relevant findings, brainstorming new ideas, verifying the accuracy of the current steps, refining any errors, and revisiting previous steps.

> **Prompt template B (PB) for variational posterior $q_\phi$**
>
> Your role as an assistant involves reconstructing the internal reasoning process that connects a provided question to its correct answer. Your task is to methodically reverse-engineer the logical steps, demonstrating a full cycle of analysis, summarization, idea generation, verification, error correction, and iterative refinement. Please structure your response into two distinct parts: Solution and Thought. In the Solution section, present the given correct answer in a precise and clear format: <|begin_of_solution|> {provided correct solution} <|end_of_solution|>. In the Thought section, articulate the step-by-step cognitive journey that leads to the solution. Use the specified format: <|begin_of_thought|> {detailed thought process with steps separated by '\\n\\n'} <|end_of_thought|>. Each step should reflect analytical breakdowns, synthesis of key points, generation of logical pathways, validation of each step's accuracy, refinement of any missteps, and reassessment of previous conclusions. The focus is solely on depicting the internal, structured thinking that arrives at the provided solution.

> **Prompt template for reasoning model $\pi_\theta$**
>
> ```
> Your role as an assistant involves thoroughly exploring questions
> through a systematic long thinking process before providing the
> final precise and accurate solutions. This requires engaging in a
> comprehensive cycle of analysis, summarizing, exploration,
> reassessment, reflection, backtracing, and iteration to develop
> well-considered thinking process. Please structure your response
> into two main sections: Thought and Solution. In the Thought
> section, detail your reasoning process using the specified format:
> <|begin_of_thought|> {thought with steps separated with '\\n\\n'}
> <|end_of_thought|> Each step should include detailed
> considerations such as analyzing questions, summarizing relevant
> findings, brainstorming new ideas, verifying the accuracy of the
> current steps, refining any errors, and revisiting previous steps.
> In the Solution section, based on various attempts, explorations,
> and reflections from the Thought section, systematically present
> the final solution that you deem correct. The solution should
> remain a logical, accurate, concise expression style and detail
> necessary step needed to reach the conclusion, formatted as
> follows: <|begin_of_solution|> {final formatted, precise, and
> clear solution} <|end_of_solution|> Now, try to solve the
> following question through the above guidelines:
> ```

Table 6: Performance of models trained from **Qwen2.5-7B-Instruct** trained on Bespoke-Stratos-17k. The best and second-best results are highlighted using **bold text** and underlined text, respectively.

| Method | MATH500 Avg@2 | AIME24 Avg@32 | AIME25 Avg@32 | AMC23 Avg@32 | OlympiadBench Avg@2 | Avg |
|---|---|---|---|---|---|---|
| Qwen2.5-7B-Instruct | 75.60 | 10.94 | 7.40 | 51.10 | 39.91 | 36.99 |
| Bespoke-Stratos-7B | 82.20 | 19.58 | 19.48 | 63.28 | 45.03 | 45.91 |
| RLT-7B | 84.30 | 22.81 | 19.48 | 64.84 | 46.43 | 47.57 |
| Ours-PA-GML-7B | **85.30** | **24.17** | 20.42 | **68.20** | 46.88 | **48.99** |
| Ours-PA-Acc-7B | 83.40 | 22.50 | **20.83** | 65.39 | 47.55 | 47.94 |
| Ours-PB-GML-7B | 84.00 | 22.08 | 20.42 | 66.80 | 46.29 | 47.92 |
| Ours-PB-Acc-7B | 84.80 | 23.96 | 19.69 | 65.00 | **48.15** | 48.32 |

| Method | GPQA-D Avg@8 | LCB-E Avg@8 | LCB-M Avg@8 | LCB-H Avg@8 | MMLU-Pro Avg@1 | Avg |
|---|---|---|---|---|---|---|
| Qwen2.5-7B-Instruct | 29.99 | 62.50 | 18.20 | 3.35 | 48.2 | 32.45 |
| Bespoke-Stratos-7B | 39.02 | 69.30 | 23.06 | 2.95 | 60.59 | 38.98 |
| RLT-7B | 41.60 | 72.32 | 25.06 | 3.66 | **61.28** | 40.78 |
| Ours-PA-GML-7B | **43.62** | 74.52 | 27.79 | **4.78** | 60.88 | 42.32 |
| Ours-PA-Acc-7B | 43.56 | 74.45 | 29.43 | 4.37 | 60.72 | **42.51** |
| Ours-PB-GML-7B | 41.60 | 74.73 | 28.46 | 4.57 | 61.14 | 42.10 |
| Ours-PB-Acc-7B | 41.67 | **75.07** | **29.49** | 4.68 | **61.28** | 42.44 |

# F    EXTENDED RESULTS

## F.1    EXTENDED MAIN RESULTS

Due to space constraints in the main paper, we present extended evaluation results in this section.

We report the evaluation results for models fine-tuned from Qwen2.5-7B-Instruct on the Bespoke-Stratos-7B dataset in Table 6. All variants of our method outperform all baselines in terms of average accuracy, demonstrating the superiority of the variational reasoning approach. Notably, the two prompt templates used for the variational posterior $q_\phi$ yield similar results, indicating that our method is robust to the choice of template. We attribute this robustness to the fact that the posterior is obtained by fine-tuning the model $q_\phi$, rather than through prompt engineering alone, thereby reducing the sensitivity to specific prompt formulations.

Additionally, we plot the distributions of the thinking token length versus the log-likelihood ratio $\log \frac{\pi_\theta(z_k|x)}{q_\phi(z_k|x,y')}$ and the answer token length versus the log-likelihood of the answer $\log \pi_\theta(\mathcal{Y}_x \mid x, z_k)$ in Figure 4. The results reveal strong correlations between these variables, indicating the presence of length biases. This observation further justifies the use of estimators based on accuracy or the geometric mean of token likelihood, rather than the naive likelihood.

Another noteworthy observation is that our evaluation results for General-Reasoner-4B (see Table 1) differ from those reported by Ma et al. (2025) in their original paper, despite using their officially released checkpoints and provided prompt template. This discrepancy can be attributed to several factors: (1) different evaluation frameworks: we employ SkyThought whereas they utilize simple-evals[4]; (2) different sampling configurations: we use `temperature=0.7` and `max_tokens=38912`, while they primarily employ greedy decoding (i.e., `temperature=0`) except for AIME24 and AIME25, along with a more constrained token budget of `max_tokens=8192`.

To facilitate a fair comparison, we provide results on the common benchmarks that we and Ma et al. (2025) both utilize, comparing our model accuracy with their officially reported values (see Table 7). As demonstrated, our method continues to outperform General-Reasoner-4B[‡] by a significant margin (where ‡ indicates their officially reported accuracy).

---

[4] https://github.com/openai/simple-evals

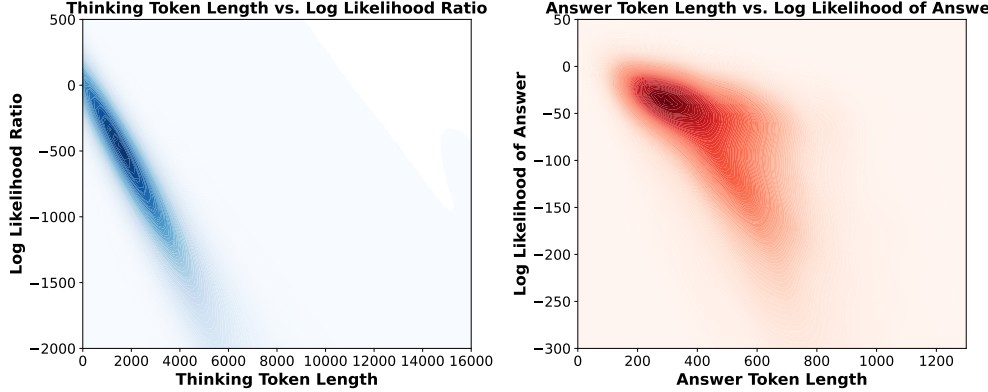

Figure 4: Density maps of the thinking token length versus the log-likelihood ratio $\log \frac{\pi_\theta(z_k|x)}{q_\phi(z_k|x,y')}$ (left), and the answer token length versus the log-likelihood of the answer $\log \pi_\theta(\mathcal{Y}_x \mid x, z_k)$ (right).

Table 7: Performance of models trained from **Qwen3-4B-Base**. All models are trained on Bespoke-Stratos-17k except for General-Reasoner-4B. Here, [‡] denotes accuracy values officially reported by Ma et al. (2025), rather than results obtained through our own evaluation. The best and second-best results are highlighted using **bold text** and underlined text, respectively.

| Method | MATH500 | AIME24 | AIME25 | AMC23 |
|---|---|---|---|---|
| Qwen3-4B-Base | 45.30 | 4.79 | 5.73 | 27.73 |
| General-Reasoner-4B | 71.70 | 19.06 | 16.77 | 55.00 |
| General-Reasoner-4B[‡] | 80.6 | 20.0 | 15.4 | 60.0 |
| Bespoke-Stratos-4B[†] | 84.70 | 27.29 | 24.17 | 70.16 |
| Ours-PB-GML-4B | 87.30 | **33.54** | 26.77 | 74.06 |
| Ours-PB-Acc-4B | **88.30** | 31.67 | **27.29** | **75.63** |

| Method | OlympiadBench | GPQA-D | MMLU-Pro |
|---|---|---|---|
| Qwen3-4B-Base | 23.37 | 29.10 | 36.89 |
| General-Reasoner-4B | 45.18 | 40.97 | 61.36 |
| General-Reasoner-4B[‡] | 47.7 | 42.9 | 62.8 |
| Bespoke-Stratos-4B[†] | 50.45 | 44.95 | 63.03 |
| Ours-PB-GML-4B | 54.45 | **45.52** | 65.52 |
| Ours-PB-Acc-4B | **55.71** | 45.33 | **65.53** |

## F.2 EXTENDED ABLATION STUDIES

This section presents extended ablation studies analyzing: the impact of different training data sources and different ways of data usage (Appendix F.3); the effect of data overlap between variational posterior training and reasoning model training (Appendix F.4); the comparison between Dr. SFT and naive SFT (Appendix F.4); and the influence of cutoff length during training (Appendix F.6).

## F.3 EFFECTS OF DIFFERENT FINAL DATA SOURCES AND WAYS OF DATA USAGE

In our main experiments, we prioritize training efficiency by using the 17k data setting, selecting the variational reasoning trace $z_k$ with the highest weight $\widetilde{\rho}_k$ and mixing it with the original Bespoke-Stratos-17k data (which results in double data sizes), rather than using all eight traces sampled from the variational posterior. To evaluate the impact of this simplification, we conduct ablation studies under the 1k data setting. We compare variants that either mix or do not mix with the original data, and that use either single-best trace selection ("-S") or weighted multiple traces ("-M").

Results are shown in Table 8. The best performance is achieved by the variant that uses weighted multiple reasoning traces without mixing with the original data. This suggests that, when computational cost is not a constraint, the optimal approach is to utilize all reasoning traces from the

Table 8: Ablation study on the effects of different final data sources (only sampled from variational posterior vs. mixed) and different ways to use samples from variational posterior to train reasoning models (single best reasoning trace selection ("-S") vs. weighted multiple reasoning traces ("-M")). This ablation is done in data 1k setting. The best and second-best results are highlighted using **bold text** and underlined text, respectively.

| Method | MATH500 Avg@2 | AIME24 Avg@32 | AIME25 Avg@32 | AMC23 Avg@32 | OlympiadBench Avg@2 | Avg |
|---|---|---|---|---|---|---|
| Qwen2.5-7B-Instruct | 75.60 | 10.94 | 7.40 | 51.10 | 39.91 | 36.99 |
| Bespoke-Stratos-7B-1K[†] | 77.20 | 16.25 | 13.96 | 53.75 | 40.88 | 40.41 |
| Ours-M-7B-1K | 79.80 | 18.65 | 16.98 | 60.55 | 44.81 | 44.16 |
|   w/o Mix | **81.30** | 19.69 | **18.44** | **61.64** | **45.99** | **45.41** |
| Ours-S-7B-1K | 81.10 | **19.90** | 17.08 | 60.39 | 43.92 | 44.48 |
|   w/o Mix | 80.40 | 17.92 | 17.81 | 59.06 | 43.18 | 43.67 |

| Method | GPQA-D Avg@8 | LCB-E Avg@8 | LCB-M Avg@8 | LCB-H Avg@8 | MMLU-Pro Avg@1 | Avg |
|---|---|---|---|---|---|---|
| Qwen2.5-7B-Instruct | 29.99 | 62.50 | 18.20 | **3.35** | 48.2 | 32.45 |
| Bespoke-Stratos-7B-1K[†] | 37.94 | 60.37 | 13.59 | 1.22 | 56.07 | 33.84 |
| Ours-M-7B-1K | **42.80** | 65.73 | 18.99 | 1.93 | 60.49 | 37.99 |
|   w/o Mix | 41.16 | **68.13** | **21.42** | 1.42 | **60.94** | **38.62** |
| Ours-S-7B-1K | 40.34 | 64.84 | 18.20 | 2.13 | 59.54 | 37.01 |
|   w/o Mix | 39.96 | 65.04 | 18.27 | 1.32 | 59.87 | 36.89 |

variational posterior, weighted by $\widetilde{\rho}_k$, for training the final reasoning model. Another interesting observation is that for the single-trace method, data mixing improves performance, whereas for the weighted multi-trace method, mixing slightly degrades performance. This may indicate that the weighted ensemble of variational traces already provides sufficient information, making the original data redundant in this scenario.

### F.4 EFFECTS OF DATA OVERLAP

In the 17k data setting, the variational posterior $q_\phi$ is trained on all 17k samples and generates thinking traces for the same set of 17k samples, which are subsequently used to train the final reasoning model $\pi_\theta$. In other words, both the variational posterior and the final reasoning model are trained on the same set of question–answer pairs.

An interesting question is how our method performs in the absence of data overlap. To investigate this, we design two experimental settings: the first is the same 1k data setting introduced earlier; the second is constructed by splitting Bespoke-Stratos-17k into two non-overlapping subsets: one contains 15,710 samples (approximately 16k) and the other contains 1k samples. In the latter setting, we train the variational posterior $q_\phi$ and the initial reasoning model $\pi_{\theta_0}$ on the 16k subset. We then use the trained $q_\phi$ to sample thinking traces for the 1k subset, and employ both $q_\phi$ and $\pi_{\theta_0}$ to compute $\widetilde{\rho}_k$. Finally, the final reasoning model $\pi_\theta$ is trained on the 1k subset using weighted multiple reasoning traces. This setting is referred to as "w/o Overlap".

The results are presented in Table 9. Both the overlap and non-overlap variants exhibit similar performance in terms of average accuracy, and both outperform the baseline, Bespoke-Stratos-7B-1K[†]. This suggests that the trained variational posterior generalizes reasonably well and can be applied to broader scenarios.

### F.5 COMPARING DR. SFT WITH NAIVE SFT

As detailed in Appendix C, we employ a slightly modified objective function, Dr. SFT, where the loss is defined as the sum of all valid token losses normalized by a constant, rather than the mean loss across valid tokens in the batch. We conduct an ablation study comparing this Dr. SFT approach against naive SFT when training the final reasoning model $\pi_\theta$.

Table 9: Ablation study on the effects of data overlap between variational posterior training and reasoning model training. This ablation is done in data 1k setting. The best and second-best results are highlighted using **bold text** and underlined text, respectively.

| Method | MATH500 Avg@2 | AIME24 Avg@32 | AIME25 Avg@32 | AMC23 Avg@32 | OlympiadBench Avg@2 | Avg |
|---|---|---|---|---|---|---|
| Qwen2.5-7B-Instruct | 75.60 | 10.94 | 7.40 | 51.10 | 39.91 | 36.99 |
| Bespoke-Stratos-7B-1K[†] | 77.20 | 16.25 | 13.96 | 53.75 | 40.88 | 40.41 |
| Ours-7B-1K | 79.80 | 18.65 | 16.98 | 60.55 | 44.81 | 44.16 |
|    w/o Mix | 81.30 | 19.69 | **18.44** | 61.64 | 45.99 | 45.41 |
|    w/o Overlap | 80.60 | **20.83** | **18.44** | 61.17 | 44.81 | 45.17 |
|    w/o Mix w/o Overlap | **82.00** | 18.75 | 16.88 | **63.52** | **47.11** | **45.65** |

| Method | GPQA-D Avg@8 | LCB-E Avg@8 | LCB-M Avg@8 | LCB-H Avg@8 | MMLU-Pro Avg@1 | Avg |
|---|---|---|---|---|---|---|
| Qwen2.5-7B-Instruct | 29.99 | 62.50 | 18.20 | **3.35** | 48.2 | 32.45 |
| Bespoke-Stratos-7B-1K[†] | 37.94 | 60.37 | 13.59 | 1.22 | 56.07 | 33.84 |
| Ours-7B-1K | **42.80** | 65.73 | 18.99 | 1.93 | 60.49 | 37.99 |
|    w/o Mix | 41.16 | 68.13 | **21.42** | 1.42 | **60.94** | **38.62** |
|    w/o Overlap | 38.19 | 66.00 | 19.84 | 1.12 | 60.65 | 37.16 |
|    w/o Mix w/o Overlap | 39.65 | **68.82** | 19.96 | 2.44 | 60.36 | 38.25 |

The results are presented in Table 10. Both variants demonstrate comparable performance, with less than 2% difference in average accuracy, and both outperform the baseline. This allows us to conclude that the primary performance improvement stems from the variational reasoning mechanism rather than from this minor modification to the objective function.

## F.6 ABLATION STUDY ON THE EFFECTS OF CUTOFF LENGTH WHEN TRAINING

In our main experiments (17k data setting), we use a cutoff length of `cutoff_len=16384`. To investigate the impact of this hyperparameter, we conduct an ablation study comparing two variants: one using the default `cutoff_len=16384` (denoted as "-Len16k") and another with `cutoff_len=32768` (denoted as "-Len32k").

The results are presented in Table 11. Both variants exhibit similar performance. We further analyze the average completion token lengths on several evaluation benchmarks (see Table 12). All methods produce significantly longer reasoning traces compared to the Qwen2.5-7B-Instruct, with our methods generating slightly longer thinking traces. Notably, the 16k and 32k cutoff variants result in similar generation lengths during inference. This indicates that increasing the cutoff length beyond 16k has a minimal effect on the model's output. Therefore, we can confidently use the 16k setting for better training efficiency without sacrificing performance.

Table 10: Ablation study comparing Dr. SFT and Naive SFT. This ablation is done in data 17k setting. The best and second-best results are highlighted using **bold text** and underlined text, respectively.

| Method | MATH500 Avg@2 | AIME24 Avg@32 | AIME25 Avg@32 | AMC23 Avg@32 | OlympiadBench Avg@2 | Avg |
|---|---|---|---|---|---|---|
| Qwen2.5-7B-Instruct | 75.60 | 10.94 | 7.40 | 51.10 | 39.91 | 36.99 |
| Bespoke-Stratos-7B | 82.20 | 19.58 | 19.48 | 63.28 | 45.03 | 45.91 |
| RLT-7B | 84.30 | 22.81 | 19.48 | 64.84 | 46.43 | 47.57 |
| Ours-PA-GML-7B | **85.30** | **24.17** | 20.42 | **68.20** | 46.88 | **48.99** |
|   w/ naive SFT | 84.00 | 22.60 | 20.10 | 65.31 | **49.11** | 48.23 |
| Ours-PA-Acc-7B | 83.40 | 22.50 | 20.83 | 65.39 | 47.55 | 47.94 |
|   w/ naive SFT | 84.10 | 23.02 | **21.04** | 67.66 | 46.96 | 48.56 |

| Method | GPQA-D Avg@8 | LCB-E Avg@8 | LCB-M Avg@8 | LCB-H Avg@8 | MMLU-Pro Avg@1 | Avg |
|---|---|---|---|---|---|---|
| Qwen2.5-7B-Instruct | 29.99 | 62.50 | 18.20 | 3.35 | 48.2 | 32.45 |
| Bespoke-Stratos-7B | 39.02 | 69.30 | 23.06 | 2.95 | 60.59 | 38.98 |
| RLT-7B | 41.60 | 72.32 | 25.06 | 3.66 | 61.28 | 40.78 |
| Ours-PA-GML-7B | **43.62** | 74.52 | 27.79 | 4.78 | 60.88 | 42.32 |
|   w/ naive SFT | 42.49 | 74.73 | 25.85 | 3.66 | **61.41** | 41.63 |
| Ours-PA-Acc-7B | 43.56 | 74.45 | **29.43** | 4.37 | 60.72 | **42.51** |
|   w/ naive SFT | 42.11 | **75.69** | 27.79 | **5.39** | 61.22 | 42.44 |

Table 11: Ablation study on effects of cutoff length used in training. This ablation is done in data 17k setting. Len16k: `cutoff_len=16384`; Len32k: `cutoff_len=32768`. The best and second-best results are highlighted using **bold text** and underlined text, respectively.

| Method | MATH500 Avg@2 | AIME24 Avg@32 | AIME25 Avg@32 | AMC23 Avg@32 | OlympiadBench Avg@2 | Avg |
|---|---|---|---|---|---|---|
| Qwen2.5-7B-Instruct | 75.60 | 10.94 | 7.40 | 51.10 | 39.91 | 36.99 |
| Bespoke-Stratos-7B | 82.20 | 19.58 | 19.48 | 63.28 | 45.03 | 45.91 |
| RLT-7B | 84.30 | 22.81 | 19.48 | 64.84 | 46.43 | 47.57 |
| Ours-PA-GML-7B-Len16k | **85.30** | **24.17** | 20.42 | **68.20** | 46.88 | **48.99** |
| Ours-PA-GML-7B-Len32k | 84.10 | 22.08 | **20.94** | 66.80 | **48.37** | 48.46 |

| Method | GPQA-D Avg@8 | LCB-E Avg@8 | LCB-M Avg@8 | LCB-H Avg@8 | MMLU-Pro Avg@1 | Avg |
|---|---|---|---|---|---|---|
| Qwen2.5-7B-Instruct | 29.99 | 62.50 | 18.20 | 3.35 | 48.2 | 32.45 |
| Bespoke-Stratos-7B | 39.02 | 69.30 | 23.06 | 2.95 | 60.59 | 38.98 |
| RLT-7B | 41.60 | 72.32 | 25.06 | 3.66 | 61.28 | 40.78 |
| Ours-PA-GML-7B-Len16k | **43.62** | 74.52 | 27.79 | **4.78** | 60.88 | 42.32 |
| Ours-PA-GML-7B-Len32k | 42.49 | **74.93** | **28.58** | 4.37 | **61.64** | **42.40** |

Table 12: Average completion token length of models trained from Qwen2.5-7B-Instruct.

| Method | MATH500 Avg@2 | AIME24 Avg@32 | AIME25 Avg@32 | AMC23 Avg@32 | MMLU-Pro Avg@1 |
|---|---|---|---|---|---|
| Qwen2.5-7B-Instruct | 564 | 1270 | 1027 | 849 | 531 |
| Bespoke-Stratos-7B | 5801 | 18413 | 15769 | 10921 | 3889 |
| RLT-7B | 5508 | 18143 | 15769 | 10986 | 3942 |
| Ours-PA-GML-7B | 5677 | 18299 | 16471 | 11338 | 3924 |
| Ours-PA-GML-7B-Len32k | 5688 | 18170 | 16747 | 11531 | 3965 |
| Ours-PA-Acc-7B | 5688 | 18170 | 16747 | 11531 | 3965 |
| Ours-PB-GML-7B | 5803 | 18479 | 16615 | 11080 | 4052 |
| Ours-PB-Acc-7B | 5787 | 18651 | 16696 | 11591 | 3974 |

## F.7 EFFECTS OF WEIGHTS FOR TRAINING THE REASONING MODEL

Our framework employs weighted supervised fine-tuning (SFT) for the reasoning model, as formalized in Eq. 6. The weight $\rho_k = \frac{\pi_\theta(\boldsymbol{z}_k|\boldsymbol{x})}{q_\phi(\boldsymbol{z}_k|\boldsymbol{x},\boldsymbol{y}')} \cdot \pi_\theta(\mathcal{Y}_{\boldsymbol{x}}|\boldsymbol{x},\boldsymbol{z}_k)$ is derived from an IWAE-style evidence lower bound (ELBO). To analyze the contribution of the weighting scheme, we conduct an ablation study with two variants: (1) uniform weighting ($\rho_k = 1$), which reduces the method to standard SFT on traces sampled from the variational posterior; and (2) correctness-only weighting, which uses only the final answer probability $\pi_\theta(\mathcal{Y}_{\boldsymbol{x}}|\boldsymbol{x},\boldsymbol{z}_k)$ and omits the likelihood ratio. Results in Table 13 show that both components of the full weighting scheme contribute positively to performance. These findings align with our theoretical derivation and validate the design of the objective.

Table 13: Ablation study on the effects of different SFT weights for training the reasoning model. The best and second-best results are highlighted using **bold text** and underlined text, respectively.

| Method | MATH500 Avg@2 | AIME24 Avg@32 | AIME25 Avg@32 | AMC23 Avg@32 | OlympiadBench Avg@2 | Avg |
|---|---|---|---|---|---|---|
| Qwen3-4B-Base | 45.30 | 4.79 | 5.73 | 27.73 | 23.37 | 21.38 |
| General-Reasoner-4B | 71.70 | 19.06 | 16.77 | 55.00 | 45.18 | 41.54 |
| Bespoke-Stratos-4B[†] | 84.70 | 27.29 | 24.17 | 70.16 | 50.45 | 51.35 |
| Ours-PB-Acc-4B | **88.30** | **31.67** | **27.29** | **75.63** | **55.71** | **55.72** |
| w/ $\rho_k = 1$ | 87.00 | 31.04 | 26.04 | 72.89 | 52.52 | 53.90 |
| w/ $\rho_k = \pi_\theta(\mathcal{Y}_{\boldsymbol{x}}|\boldsymbol{x},\boldsymbol{z}_k)$ | 88.20 | 31.45 | 26.56 | 73.43 | 54.45 | 54.82 |

| Method | GPQA-D Avg@8 | LCB-E Avg@8 | LCB-M Avg@8 | LCB-H Avg@8 | MMLU-Pro Avg@1 | Avg |
|---|---|---|---|---|---|---|
| Qwen3-4B-Base | 29.10 | 18.54 | 5.46 | 1.32 | 36.89 | 18.26 |
| General-Reasoner-4B | 40.97 | 61.40 | 17.90 | 2.85 | 61.36 | 36.90 |
| Bespoke-Stratos-4B[†] | 44.95 | 71.22 | 19.54 | 3.25 | 63.03 | 40.40 |
| Ours-PB-Acc-4B | **45.33** | **80.29** | **33.68** | **5.79** | **65.53** | **46.12** |
| w/ $\rho_k = 1$ | 45.07 | 78.09 | 28.82 | 5.38 | 64.24 | 44.32 |
| w/ $\rho_k = \pi_\theta(\mathcal{Y}_{\boldsymbol{x}}|\boldsymbol{x},\boldsymbol{z}_k)$ | 45.07 | 78.43 | 30.27 | 5.48 | 64.41 | 44.73 |

## F.8 EFFECTS OF NOISY HINTS

As a sanity check, we conduct an experiment to confirm a basic expectation: sampling reasoning traces from the well-trained variational posterior should depend critically on the quality of the conditioning hint $\boldsymbol{y}'$. We generate noisy hints by using an LLM (Qwen3-4B-Instruct-2507) to rewrite the original hints, introducing errors, and and use these noisy hints for sampling reasoning traces using the variational posterior. The results, shown in Table 14, confirm that providing noisy hints significantly degrades the quality of the sampled reasoning traces. This outcome validates our basic assumption, as it demonstrates the sensitivity of the posterior to its conditioning input, which is consistent with its theoretical role.

## F.9 EFFECTS OF THE VERIFIER'S ACCURACY

To estimate $q_\phi(\boldsymbol{z}_k|\boldsymbol{x},\boldsymbol{y}')$ for the weight $\rho_k$, we have two options: a likelihood-based estimator (and its variant with a geometric mean modification) and an accuracy-based estimator. The likelihood-based approach does not require a verifier but relies on a reference answer, whereas the accuracy-based estimator depends on a verifier.

To examine the robustness of our method to verifier accuracy, we conduct an ablation study using a simulated, highly inaccurate "dummy verifier" by setting all $q_\phi(\boldsymbol{z}_k|\boldsymbol{x},\boldsymbol{y}')$ values to 0.5. As shown in Table 15, while an inaccurate verifier can degrade performance, our method still outperforms the baselines. We attribute this robustness to the fact that the variational posterior is conditioned on a reasoning hint, which maintains a high overall correctness. Nevertheless, these results confirm that a more accurate verifier is still preferable in practice.

Table 14: Ablation study on the effects of the quality of hints used for sampling from the variational posterior. The best and second-best results are highlighted using **bold text** and underlined text, respectively.

| Method | MATH500 Avg@2 | AIME24 Avg@32 | AIME25 Avg@32 | AMC23 Avg@32 | OlympiadBench Avg@2 | Avg |
|---|---|---|---|---|---|---|
| Qwen3-4B-Base | 45.30 | 4.79 | 5.73 | 27.73 | 23.37 | 21.38 |
| General-Reasoner-4B | 71.70 | 19.06 | 16.77 | 55.00 | 45.18 | 41.54 |
| Bespoke-Stratos-4B[†] | 84.70 | 27.29 | 24.17 | 70.16 | 50.45 | 51.35 |
| Ours-PB-Acc-4B | **88.30** | **31.67** | **27.29** | **75.63** | **55.71** | **55.72** |
| w/ noisy hints | 86.10 | 27.29 | 24.27 | 70.00 | 50.52 | 51.64 |

| Method | GPQA-D Avg@8 | LCB-E Avg@8 | LCB-M Avg@8 | LCB-H Avg@8 | MMLU-Pro Avg@1 | Avg |
|---|---|---|---|---|---|---|
| Qwen3-4B-Base | 29.10 | 18.54 | 5.46 | 1.32 | 36.89 | 18.26 |
| General-Reasoner-4B | 40.97 | 61.40 | 17.90 | 2.85 | 61.36 | 36.90 |
| Bespoke-Stratos-4B[†] | 44.95 | 71.22 | 19.54 | 3.25 | 63.03 | 40.40 |
| Ours-PB-Acc-4B | **45.33** | **80.29** | **33.68** | **5.79** | **65.53** | **46.12** |
| w/ noisy hints | 44.82 | 74.24 | 26.21 | 4.98 | 64.18 | 42.88 |

## F.10 Effects of Geometric Mean Modification on Likelihood Ratio

As derived in Section 2.2, the weight $\rho_k$ used for training the reasoning model in Eq. (6) can be decomposed as $\rho_k = \frac{\pi_\theta(z_k|x)}{q_\phi(z_k|x,y')} \cdot \pi_\theta(\mathcal{Y}_x|x, z_k)$, where the first term, $\frac{\pi_\theta(z_k|x)}{q_\phi(z_k|x,y')}$, is the likelihood ratio of the thinking trace $z_k$.

The standard likelihood ratio is unbounded, which can lead to high variance, and exhibits a clear length bias as demonstrated in Figure 4 (left). To address this, we heuristically introduce a geometric mean modification to the likelihood ratio, as defined in Eq. 8. While this modification introduces estimation bias, we evaluate its practical utility by comparing our method against a variant that removes this modification. Results presented in Table 16 indicate that the geometric mean is indeed beneficial. The principled variant (strictly derived from variational inference) performs slightly worse than our modified version, yet still surpasses all baseline methods, confirming the overall robustness of the framework.

## F.11 Comparison Against Baselines With Matched Computational Budget

Our method involves training both a variational posterior and a reasoning model, and also includes additional sampling and forward passes to compute token probabilities, which incurs extra computational cost. To further ensure a fair comparison, we scaled the compute of the baseline, Bespoke-Stratos-4B, to exceed the total GPU hours used by our framework. This is achieved by increasing both its training epochs and batch size. Consequently, the baseline's total number of training tokens also largely exceeds that of our method. The results, presented in Table 17, indicate that even under a similar computational budget, our approach achieves better performance. This demonstrates the practical value of variational reasoning.

Table 15: Ablation study on the effects of the verifier's accuracy. The best and second-best results are highlighted using **bold text** and underlined text, respectively.

| Method | MATH500 Avg@2 | AIME24 Avg@32 | AIME25 Avg@32 | AMC23 Avg@32 | OlympiadBench Avg@2 | Avg |
|---|---|---|---|---|---|---|
| Qwen3-4B-Base | 45.30 | 4.79 | 5.73 | 27.73 | 23.37 | 21.38 |
| General-Reasoner-4B | 71.70 | 19.06 | 16.77 | 55.00 | 45.18 | 41.54 |
| Bespoke-Stratos-4B[†] | 84.70 | 27.29 | 24.17 | 70.16 | 50.45 | 51.35 |
| Ours-PB-Acc-4B | **88.30** | **31.67** | **27.29** | **75.63** | **55.71** | **55.72** |
| w/ dummy verifier | 88.00 | 31.25 | 26.45 | 74.21 | 53.64 | 54.71 |

| Method | GPQA-D Avg@8 | LCB-E Avg@8 | LCB-M Avg@8 | LCB-H Avg@8 | MMLU-Pro Avg@1 | Avg |
|---|---|---|---|---|---|---|
| Qwen3-4B-Base | 29.10 | 18.54 | 5.46 | 1.32 | 36.89 | 18.26 |
| General-Reasoner-4B | 40.97 | 61.40 | 17.90 | 2.85 | 61.36 | 36.90 |
| Bespoke-Stratos-4B[†] | 44.95 | 71.22 | 19.54 | 3.25 | 63.03 | 40.40 |
| Ours-PB-Acc-4B | **45.33** | **80.29** | **33.68** | **5.79** | **65.53** | **46.12** |
| w/ dummy verifier | 45.26 | 79.67 | 31.12 | 4.98 | 64.90 | 45.18 |

Table 16: Ablation study on the effects of geometric mean modification of the likelihood ratio. The best and second-best results are highlighted using **bold text** and underlined text, respectively.

| Method | MATH500 Avg@2 | AIME24 Avg@32 | AIME25 Avg@32 | AMC23 Avg@32 | OlympiadBench Avg@2 | Avg |
|---|---|---|---|---|---|---|
| Qwen3-4B-Base | 45.30 | 4.79 | 5.73 | 27.73 | 23.37 | 21.38 |
| General-Reasoner-4B | 71.70 | 19.06 | 16.77 | 55.00 | 45.18 | 41.54 |
| Bespoke-Stratos-4B[†] | 84.70 | 27.29 | 24.17 | 70.16 | 50.45 | 51.35 |
| Ours-PB-Acc-4B | **88.30** | 31.67 | 27.29 | **75.63** | **55.71** | **55.72** |
| w/o geometric mean | 88.00 | **32.60** | **27.60** | 74.14 | 52.30 | 54.93 |

| Method | GPQA-D Avg@8 | LCB-E Avg@8 | LCB-M Avg@8 | LCB-H Avg@8 | MMLU-Pro Avg@1 | Avg |
|---|---|---|---|---|---|---|
| Qwen3-4B-Base | 29.10 | 18.54 | 5.46 | 1.32 | 36.89 | 18.26 |
| General-Reasoner-4B | 40.97 | 61.40 | 17.90 | 2.85 | 61.36 | 36.90 |
| Bespoke-Stratos-4B[†] | 44.95 | 71.22 | 19.54 | 3.25 | 63.03 | 40.40 |
| Ours-PB-Acc-4B | **45.33** | **80.29** | **33.68** | 5.79 | **65.53** | **46.12** |
| w/o geometric mean | 45.20 | 78.57 | 31.18 | **5.99** | 65.22 | 45.23 |

Table 17: Performance comparison under a similar computational budget. The best and second-best results are highlighted using **bold text** and underlined text, respectively.

| Method | MATH500 Avg@2 | AIME24 Avg@32 | AIME25 Avg@32 | AMC23 Avg@32 | OlympiadBench Avg@2 | Avg |
|---|---|---|---|---|---|---|
| Qwen3-4B-Base | 45.30 | 4.79 | 5.73 | 27.73 | 23.37 | 21.38 |
| Bespoke-Stratos-4B[†] | 84.70 | 27.29 | 24.17 | 70.16 | 50.45 | 51.35 |
| w/ scaled compute | 86.20 | 29.79 | 25.52 | 72.81 | 52.30 | 53.32 |
| Ours-PB-GML-4B | 87.30 | **33.54** | 26.77 | 74.06 | 54.45 | 55.23 |
| Ours-PB-Acc-4B | **88.30** | 31.67 | **27.29** | **75.63** | **55.71** | **55.72** |

| Method | GPQA-D Avg@8 | LCB-E Avg@8 | LCB-M Avg@8 | LCB-H Avg@8 | MMLU-Pro Avg@1 | Avg |
|---|---|---|---|---|---|---|
| Qwen3-4B-Base | 29.10 | 18.54 | 5.46 | 1.32 | 36.89 | 18.26 |
| Bespoke-Stratos-4B[†] | 44.95 | 71.22 | 19.54 | 3.25 | 63.03 | 40.40 |
| w/ scaled compute | 45.01 | 75.61 | 27.91 | 4.57 | 63.99 | 43.42 |
| Ours-PB-GML-4B | **45.52** | 79.53 | 31.25 | **6.20** | 65.52 | 45.60 |
| Ours-PB-Acc-4B | 45.33 | **80.29** | **33.68** | 5.79 | **65.53** | **46.12** |

## G  EXAMPLES OF GENERATED RESPONSES

In this section, we provide examples of responses from the variational posterior $q_\phi$ and the final reasoning model $\pi_\theta$ trained based on Qwen3-4B-Base. The provided example of the variational posterior $q_\phi$ is based on the variant with prompt template B, and the question-answer pair is one of the samples from the Bespoke-Stratos-17k dataset. The provided example of the final reasoning model $\pi_\theta$ is based on one of the samples in MATH500. See the provided examples below.

---

**Example of variational posterior $q_\phi$**

**Prompt:**

```
<|im_start|>system
Your role as an assistant involves reconstructing the internal
reasoning process that connects a provided question to its correct
answer. Your task is to methodically reverse-engineer the logical
steps, demonstrating a full cycle of analysis, summarization, idea
generation, verification, error correction, and iterative
refinement. Please structure your response into two distinct
parts: Solution and Thought. In the Solution section, present the
given correct answer in a precise and clear format:
<|begin_of_solution|> {provided correct solution}
<|end_of_solution|>. In the Thought section, articulate the
step-by-step cognitive journey that leads to the solution. Use the
specified format: <|begin_of_thought|> {detailed thought process
with steps separated by '\n\n'} <|end_of_thought|>. Each step
should reflect analytical breakdowns, synthesis of key points,
generation of logical pathways, validation of each step's
accuracy, refinement of any missteps, and reassessment of previous
conclusions. The focus is solely on depicting the internal,
structured thinking that arrives at the provided
solution.<|im_end|>
<|im_start|>user
Return your final response within \boxed{}. A figure is an
equiangular parallelogram if and only if it is a
$\textbf{(A)}\ \text{rectangle}\qquad \textbf{(B)}\ \text{regular
polygon}\qquad \textbf{(C)}\ \text{rhombus}\qquad \textbf{(D)}\
\text{square}\qquad \textbf{(E)}\ \text{trapezoid}$<|im_end|>
<|im_start|>assistant
<|begin_of_solution|>

To determine the correct answer, we need to analyze the
definitions and properties of the given options in relation to an
equiangular parallelogram.

1. **Definitions and Properties**:
- A **parallelogram** is a quadrilateral with both pairs of
opposite sides parallel.
- **Equiangular** means all angles are equal. In a quadrilateral,
this means each angle is 90 degrees (since the sum of angles in a
quadrilateral is 360 degrees, and 360/4 = 90).
- A **rectangle** is a parallelogram with all angles equal to 90
degrees.
- A **regular polygon** (for quadrilaterals) is a square, which is
both equiangular and equilateral.
- A **rhombus** is a parallelogram with all sides equal, but not
necessarily equiangular.
- A **square** is a special case of both a rectangle and a
rhombus, being both equiangular and equilateral.
- A **trapezoid** has at least one pair of parallel sides, but not
necessarily two pairs.

2. **Analysis**:
```

---

– An equiangular parallelogram must have all angles equal to 90 degrees, making it a rectangle.
– The "if and only if" condition requires that the converse is also true: a rectangle is an equiangular parallelogram.
– A regular polygon (quadrilateral) is a square, which is too specific since the problem does not mention sides being equal.
– A rhombus does not require all angles to be equal, so it is not necessarily equiangular.
– A square is a special case of a rectangle but requires both equiangular and equilateral properties, which are not both specified here.
– A trapezoid does not necessarily have two pairs of parallel sides, hence not a parallelogram.

3. **Conclusion**:
– The correct answer is a rectangle, as it satisfies both the equiangular condition and the properties of a parallelogram.

\boxed{A}

<|end_of_solution|>

<|begin_of_thought|>

**Response:**

Okay, let me try to figure out this problem. The question is: "A figure is an equiangular parallelogram if and only if it is a..." with options given. The answer choices are rectangle, regular polygon, rhombus, square, trapezoid. Hmm. So, first, let's break down the terms here.

An equiangular parallelogram. Let's recall definitions. A parallelogram is a quadrilateral with both pairs of opposite sides parallel. Equiangular means that all the angles are equal. In a quadrilateral, if all angles are equal, each angle must be 90 degrees because the sum of angles in a quadrilateral is 360 degrees. So, 360 divided by 4 is 90. So, an equiangular quadrilateral has four right angles. Wait, but a parallelogram by definition already has opposite angles equal. So, if a parallelogram is equiangular, that forces all angles to be 90 degrees, right? So, that would make it a rectangle. Because a rectangle is a parallelogram with all angles equal to 90 degrees. So, then, an equiangular parallelogram is a rectangle. Then the "if and only if" part. So, the statement is saying that a figure is an equiangular parallelogram exactly when it is a [answer]. So, the converse would also have to be true. That is, if it's a [answer], then it's an equiangular parallelogram, and vice versa.

So, let's check the options. Option A is rectangle. If a figure is a rectangle, then it's a parallelogram with all angles equal (to 90 degrees), so it's an equiangular parallelogram. Conversely, if it's an equiangular parallelogram, then as we saw, all angles are 90 degrees, so it's a rectangle. So, that seems to satisfy the "if and only if" condition.

But wait, let's check the other options just to be thorough. Option B is regular polygon. A regular polygon is a polygon that is both equiangular and equilateral. In the case of a quadrilateral, a regular polygon would be a square. So, a square is a regular quadrilateral. But a parallelogram that is equiangular is a rectangle, which isn't necessarily a square unless it's also equilateral (all sides equal). So, regular polygon is too specific here. Moreover, the question is about a

parallelogram, which is a quadrilateral, but "regular polygon" could refer to any regular polygon, not necessarily a quadrilateral, but in the answer choices, it's listed as an option. But since a regular polygon (quadrilateral) is a square, but the question is about an equiangular parallelogram, which is a rectangle. So, unless rectangles are considered regular polygons, but rectangles are not regular polygons unless they are squares. So, regular polygon is not the correct answer here. So, B is out.

Option C is rhombus. A rhombus is a parallelogram with all sides equal. A rhombus doesn't have to be equiangular. In fact, a rhombus can have angles of, say, 60 and 120 degrees. So, a rhombus is not necessarily equiangular. So, an equiangular parallelogram is a rectangle, not necessarily a rhombus. So, C is out.

Option D is square. A square is a parallelogram that is both equiangular (all angles 90 degrees) and equilateral (all sides equal). So, the problem here is that an equiangular parallelogram is a rectangle, which is only a square if the sides are equal. But the question doesn't specify that the sides are equal. So, an equiangular parallelogram is a rectangle, but a rectangle is a square only if it's equilateral. Since the problem is only specifying equiangular (all angles equal) and parallelogram, the figure is a rectangle, which is a type of parallelogram but not necessarily a square. Therefore, the "if and only if" would require that being a square is equivalent to being an equiangular parallelogram. But that's not true because a square is a special case. So, D is not correct.

Option E is trapezoid. A trapezoid is a quadrilateral with at least one pair of parallel sides. However, a parallelogram has two pairs of parallel sides, so a parallelogram is a special case of a trapezoid (depending on the definition; in some definitions, trapezoids have exactly one pair). But even so, a trapezoid isn't necessarily a parallelogram. Moreover, an equiangular trapezoid would have all angles equal, but in a trapezoid, which is not necessarily a parallelogram, having all angles equal doesn't guarantee that the sides are parallel. For example, an isosceles trapezoid can be equiangular (with two pairs of equal angles), but that's a different case. However, the question specifies an equiangular parallelogram. So, if it's a parallelogram, then it's already a quadrilateral with two pairs of parallel sides, and if it's equiangular, all angles are 90 degrees, making it a rectangle. A trapezoid isn't required to have two pairs of parallel sides, so an equiangular trapezoid isn't necessarily a parallelogram. Therefore, E is out.

So, the only remaining option is A, rectangle. Let me just verify again. If it's a rectangle, then it's a parallelogram with all angles equal, so equiangular. Conversely, if it's an equiangular parallelogram, all angles are 90 degrees, hence a rectangle. The "if and only if" holds here. So, the answer should be A.

Wait, but hold on. The problem says "a figure is an equiangular parallelogram if and only if it is a [answer]". So, this is a biconditional statement. So, the statement and the statement "it is a [answer]" must be equivalent. Therefore, the [answer] must be such that every equiangular parallelogram is a [answer], and every [answer] is an equiangular parallelogram.

But as I thought earlier, a rectangle is an equiangular parallelogram. However, the reverse – is every rectangle an equiangular parallelogram? Well, yes, because a rectangle by

definition has four right angles, so it's equiangular. So, the equivalence holds. Therefore, the if and only if is satisfied.

But wait, the answer options include square. So, maybe there's a confusion here. Let me think again. The question is asking for the figure that is an equiangular parallelogram if and only if it is a [answer]. So, the figure is a parallelogram that is equiangular, and this happens exactly when it's a [answer]. So, since we concluded that this [answer] is rectangle, then A is correct.

But let me check if any of the other options could also satisfy the biconditional. For example, square. So, is a square an equiangular parallelogram? Yes, because a square is a parallelogram with all sides equal and all angles equal (90 degrees). However, the problem states "if and only if". So, the figure is an equiangular parallelogram exactly when it is a square. But that's not true. Because a rectangle is also an equiangular parallelogram, but a square is a special case of a rectangle. So, if the statement was "if and only if it is a square", then it would be too restrictive, because there are rectangles that are equiangular parallelograms which are not squares. Therefore, the "if and only if" with square would not hold. Similarly, regular polygon is too broad or specific. So, the correct answer is rectangle.

Therefore, the answer should be A. But wait, the original problem says "equiangular parallelogram". Wait, in some contexts, a parallelogram being equiangular is the same as being a rectangle, so equiangular parallelogram = rectangle. Then the question is asking, "An equiangular parallelogram is a [answer] if and only if it is a..." Hmm, perhaps the wording is confusing. Let me parse the sentence again.

"A figure is an equiangular parallelogram if and only if it is a [answer]"

So, the figure is defined as an equiangular parallelogram, and this is equivalent (if and only if) it is a [answer]. So, the definition of an equiangular parallelogram is "it is a [answer]". Therefore, the [answer] must be the name given to the figure that is exactly the equiangular parallelogram. Since we've concluded that an equiangular parallelogram is a rectangle, then the definition would be "a rectangle is an equiangular parallelogram". But the answer options have rectangle as A. However, in logic, the if and only if is symmetric. So, saying "a figure is an equiangular parallelogram if and only if it is a rectangle" is correct. So, that would be answer A.

But then why is square an option? Maybe I made a mistake here. Let's consider that perhaps the question is from a country where parallelogram definitions differ. For example, in some definitions, a parallelogram is a quadrilateral with both pairs of opposite sides parallel, and in those cases, a parallelogram is only a rectangle if it has right angles. But in standard definitions, a rectangle is a parallelogram with four right angles, so that's correct.

Alternatively, maybe there is a different interpretation. Wait, the problem is in the "if and only if" part. So, if we take "equiangular parallelogram" as the definition, then the "if and only if" it is a [answer]. So, the [answer] must be co-extensive with the equiangular parallelogram. In other words, the set of equiangular parallelograms is the same as the set of [answer]. So,

```
since we know that an equiangular parallelogram is a rectangle,
but a rectangle is not necessarily a square (since rectangles can
have sides of different lengths), then the set of equiangular
parallelograms is exactly the set of rectangles. Therefore, the
answer is rectangle.

But maybe the problem is trying to trick us into thinking that a
square is the answer because it's a special case. But since the
problem doesn't mention sides being equal, just angles being
equal, it's a rectangle. So, the answer should be A.

Alternatively, perhaps there's a different approach. Let's recall
that in a parallelogram, opposite angles are equal. If it's
equiangular, then all angles are equal. So, in a parallelogram,
this forces all angles to be 90 degrees. Therefore, it's a
rectangle. Therefore, equiangular parallelogram is another name
for rectangle. So, if the question is asking for the term that
describes such a figure, the answer is rectangle. So, A.

But maybe the options are labeled differently. Let me check the
options again:

A) rectangle

B) regular polygon

C) rhombus

D) square

E) trapezoid

Yes. So, given that, the answer is A. But let me verify once more.
Suppose someone thinks that "equiangular" refers to both angles
and sides. But no, equiangular just means all angles are equal.
Equilateral would mean all sides are equal. So, the problem is
only mentioning angles. So, a parallelogram that is equiangular
must have all angles 90 degrees, hence a rectangle. Therefore, the
answer is A.

I think that's solid. So, I think the answer is A, rectangle.

**Final Answer**
\boxed{A}
```

---

**Example of reasoning model $\pi_\theta$**

**Prompt:**

```
<|im_start|>system
Your role as an assistant involves thoroughly exploring questions
through a systematic long thinking process before providing the
final precise and accurate solutions. This requires engaging in a
comprehensive cycle of analysis, summarizing, exploration,
reassessment, reflection, backtracing, and iteration to develop
well-considered thinking process. Please structure your response
into two main sections: Thought and Solution. In the Thought
section, detail your reasoning process using the specified format:
<|begin_of_thought|> {thought with steps separated with '\n\n'}
<|end_of_thought|> Each step should include detailed
considerations such as analisying questions, summarizing relevant
findings, brainstorming new ideas, verifying the accuracy of the
current steps, refining any errors, and revisiting previous steps.
```

In the Solution section, based on various attempts, explorations,
and reflections from the Thought section, systematically present
the final solution that you deem correct. The solution should
remain a logical, accurate, concise expression style and detail
necessary step needed to reach the conclusion, formatted as
follows: <|begin_of_solution|> {final formatted, precise, and
clear solution} <|end_of_solution|> Now, try to solve the
following question through the above guidelines:<|im_end|>
<|im_start|>user
If $f(x) = \frac{3x-2}{x-2}$, what is the value of $f(-2)
+f(-1)+f(0)$? Express your answer as a common fraction.<|im_end|>
<|im_start|>assistant
<|begin_of_thought|>

**Response:**

Okay, so I need to find the value of f(-2) + f(-1) + f(0) for the
function f(x) = (3x − 2)/(x − 2). Hmm, let me start by recalling
how to evaluate a function at a specific point. For each x-value
given, I substitute it into the function and simplify. Let me
break this dow
n step by step.

First, let's compute f(-2). Substituting x = -2 into the function:

f(-2) = [3*(-2) − 2]/[(-2) − 2] = [(-6) − 2]/[-4] = (-8)/(-4) = 2.
Wait, that seems straightforward. The numerator becomes -8 and the
denominator becomes -4, and a negative divided by a negative is
positive 2. Okay, so f(-2) is 2.

Next, let's find f(-1). Substituting x = -1:

f(-1) = [3*(-1) − 2]/[(-1) − 2] = [(-3) − 2]/[-3] = (-5)/(-3) =
5/3. Hmm, negative divided by negative is positive, so 5/3. That
makes sense.

Now, moving on to f(0). Substitute x = 0:

f(0) = [3*0 − 2]/[0 − 2] = [0 − 2]/[-2] = (-2)/(-2) = 1. Okay, so
f(0) is 1.

Now, adding these three results together: 2 + 5/3 + 1. Let me
convert the whole numbers to fractions to make addition easier. 2
is 6/3, and 1 is 3/3. So adding them up:

6/3 + 5/3 + 3/3 = (6 + 5 + 3)/3 = 14/3.

Wait, let me double-check each calculation to make sure I didn't
make any mistakes. Starting with f(-2):

Numerator: 3*(-2) = -6, minus 2 is -8. Denominator: -2 − 2 = -4.
So -8/-4 is indeed 2. Correct.

For f(-1):

Numerator: 3*(-1) = -3, minus 2 is -5. Denominator: -1 − 2 = -3.
So -5/-3 is 5/3. Correct.

For f(0):

Numerator: 3*0 = 0, minus 2 is -2. Denominator: 0 − 2 = -2. So
-2/-2 is 1. Correct.

Adding them up: 2 + 5/3 + 1. Converting 2 and 1 to thirds: 2 is 6/3, 1 is 3/3. So 6/3 + 5/3 + 3/3 = 14/3. That seems right.

Alternatively, maybe I can add them in decimal form to verify. 2 is 2.0, 5/3 is approximately 1.666..., and 1 is 1.0. Adding those: 2.0 + 1.666... + 1.0 = 4.666..., which is 14/3 (since 14 divided by 3 is approximately 4.666...). So that checks out.

I don't think I made any errors here. Each substitution was done correctly, and the arithmetic adds up. Therefore, the final answer should be 14/3.

<end_of_thought>

<begin_of_solution>

To find $f(-2) + f(-1) + f(0)$ for $f(x) = \frac{3x-2}{x-2}$:

1. **Calculate $f(-2)$:**
$$
f(-2) = \frac{3(-2) - 2}{-2 - 2} = \frac{-6 - 2}{-4} = \frac{-8}{-4} = 2
$$

2. **Calculate $f(-1)$:**
$$
f(-1) = \frac{3(-1) - 2}{-1 - 2} = \frac{-3 - 2}{-3} = \frac{-5}{-3} = \frac{5}{3}
$$

3. **Calculate $f(0)$:**
$$
f(0) = \frac{3(0) - 2}{0 - 2} = \frac{-2}{-2} = 1
$$

4. **Sum the results:**
$$
2 + \frac{5}{3} + 1 = \frac{6}{3} + \frac{5}{3} + \frac{3}{3} = \frac{14}{3}
$$

The final answer is $\boxed{\dfrac{14}{3}}$.

<end_of_solution>

