# OpenReview forum: "Variational Reasoning for Language Models"
_ICLR.cc/2026/Conference — ICLR 2026 Poster_

### Official Review · Reviewer_xe5w · 2025-10-31

**Soundness:** 2
**Presentation:** 1
**Contribution:** 3
**Rating:** 6
**Confidence:** 3

**Summary:**

This paper proposes a framework for fine-tuning a reasoning LLM to generate better “thinking traces”. Specifically, this paper first trains an auxiliary LLM to use the answer as a hint to generate better thinking traces, then trains the original LLM to generate similar high-quality thinking traces without seeing any hint. The idea is put in a variational framework: the thinking traces are formulated as latent variables and the auxiliary LLM and original LLM are alternately optimized for ELBO. The auxiliary LLM is viewed as a variational posterior, and a forward-KL objective is used to optimize it to avoid collapsing to shortcut reasoning. A tighter IWAE‑style multi‑trace bound is used, and practical tricks are proposed for it to work. Besides, The authors show that RFT and binary‑reward RL including GRPO are local forward‑KL updates implicitly weighted by model accuracy, explaining a bias towards easier questions. On math/code/general‑knowledge benchmarks with Qwen 2.5/3 models, the method outperforms strong baselines and trains more stably.

**Strengths:**

* The proposed method is intuitive at a high level: to train an auxiliary LLM which can make use of a hint to generate better thinking trace, and use this model to improve the original one.
* Many techniques are proposed to make the method work in practice, including the tighter IWAE-style lower bound, accuracy-based estimator for lower worst-case variances, forward-KL to avoid the variational posterior collapsing to trivial solution, etc.
* The connection between RFT/binary reward RL and forward-KL is revealed, which also reveals their bias toward easier questions.

**Weaknesses:**

* The paper states that "our work provides a principled probabilistic perspective that unifies variational inference with RL-style methods". However, tricks like using geometric mean as a surrogate for the likelihood ratio and using forward-KL to optimize the variational posterior are not compatible with VI as far as I can see. I appreciate the method for its practical impact and clarity of intent, but I think connecting it to VI to make it "principled" may be overstating without additional justification or ablations.
* The presentation is difficult to follow on a first read. Algorithm 1 greatly clarifies the pipeline, but it appears after substantial technical detail. Reordering the exposition (high‑level pipeline, objective, implementation details) could make the paper more accessible to newcomers.

**Questions:**

* Have you tried to use the variational posterior to generate high-quality thinking traces and use them to do simple SFT on the original LLM? How crucial are the VI/ELBO components in that case?
* Is it possible to extend this method to scenarios where a rule-based verifier doesn't exist?
* As a lightweight qualitative check, it would be interesting to compare the auxiliary model’s traces when conditioned on a correct vs. incorrect answer hint.

---

> ### Author Response · Authors · 2025-11-21
> **Rebuttal by Authors**
>
> Thank you for your supportive review and suggestions. Below we respond to the comments in **Weaknesses (W)** and **Questions (Q)**.
>
> ---
>
>
> ***W1: Tricks like using geometric mean and forward KL are not compatible with VI.***
>
> The use of forward KL divergence is a well-principled choice. The evidence lower bound (ELBO) holds for **any** distribution $q\_\\phi$, and forward KL is a standard objective used in established methods like wake-sleep algorithms [1]. Both forward and reverse KL divergence converge to the same optimum: the true posterior distribution. We further provide a theoretical proof that the variational posterior is the optimal variance-minimizing behavior policy ($\\textrm{\\color{blue}Appendix A.8}$), offering additional justification for our use of forward KL.
>
> We acknowledge that the geometric mean modification applied to the ratio term in $\\rho\_k$ is a heuristic adjustment that introduces bias. However, our ablation study ($\\textrm{\\color{blue}Table 16}$) indicates that it leads to a slight performance improvement. Even without this modification, the framework outperforms the baselines, confirming the robustness of the core approach. The modification can thus be viewed as a practical trade-off that balances theoretical bias against empirical performance.
>
>
> [1] Reweighted Wake-Sleep, ICLR 2015
>
> ---
>
> ***W2: The presentation is difficult to follow on a first read. Reordering could make it more accessible to newcomers.***
>
> Thank you for the suggestion. We will reorganize the paper presentation to first introduce the overall pipeline, then the objective, and finally the implementation details in the future version.
>
> ---
>
> ***Q1: Ablation of simple SFT on thinking traces sampled from variational posterior.***
>
> We conducted an ablation study using a variant with $\\tilde{\\rho}\_k = 1$, which is equivalent to simple supervised fine-tuning (SFT) without weighting. The results are presented in $\\textrm{\\color{blue}Table 13}$.
>
> The results indicate that replacing the variational-inference-derived weighting $\\tilde{\\rho}\_k$ with uniform weighting (i.e., simple SFT) leads to degraded performance compared to the complete variational reasoning framework. This underscores the importance of the VI/ELBO-based weighting scheme when training the reasoning model. Additionally, we observe that even simple SFT applied to traces sampled from the variational posterior still outperforms the baseline. This suggests that the learned posterior effectively approximates the true posterior, thereby contributing to improved performance over the baseline method.
>
> ---
>
>
> ***Q2: Extending this method to scenarios where a rule-based verifier doesn't exist.***
>
> Our framework does not fundamentally rely on a verifier, as it supports two distinct approaches for estimating $\\log \\pi\_\\theta(\\mathcal{Y}\_x|x,z)$: an accuracy-based estimator that requires a verifier, and a geometric-mean-likelihood-based estimator that operates without a verifier while achieving comparable performance. These variants are denoted as ''-Acc'' and ''-GML'' in our experiments. It is worth noting that most reinforcement learning algorithms typically require some form of verification mechanism, making our verifier-free variant particularly advantageous.
>
>
> ---
>
>
> ***Q3: Comparison against the variants with traces conditioned on incorrect hints.***
>
> For this comparison, we generate noisy hints by instructing an LLM (Qwen3-4B-Instruct-2507) to rewrite the original hints, thereby introducing errors. When these corrupted hints are used to sample reasoning traces from the variational posterior, the quality of the generated traces is markedly reduced, as evidenced by the results in $\\textrm{\\color{blue}Table 14}$. This degradation confirms a fundamental property of the posterior: its dependence on the quality of the conditioning input, which aligns perfectly with its theoretical formulation.

---

### Official Review · Reviewer_nKtU · 2025-11-01

**Soundness:** 3
**Presentation:** 4
**Contribution:** 3
**Rating:** 6
**Confidence:** 3

**Summary:**

This paper applies variational inference to enhance language models' chain-of-thought reasoning by treating reasoning traces as latent variables and optimizing an ELBO-based objective. The authors further extend the objective to an IWAE-style multi-trace formulation for a tighter bound and introduce a stabilized training procedure using a forward-KL term. Experiments on math, coding, and general reasoning benchmarks show that this variational reasoning framework consistently outperforms SFT, RFT, and RL baselines, while also generalizing well to out-of-distribution tasks.

**Strengths:**

- Leverages a principled variational inference framework to improve reasoning by treating thought traces as latent variables, providing a clean probabilistic formulation.
- Uses a forward-KL based objective to address instability and collapse issues in training the variational posterior, leading to more stable optimization.
- Algorithmic pipeline is clearly presented and appears reproducible, with explicit sampling, weighting, and update procedures.
- Strong empirical evaluation across diverse reasoning tasks (math, coding, and general benchmarks), including out-of-distribution settings.
- Comprehensive ablations that assess key design choices, such as the effect of the answer hint, different estimators for correctness probability, and varying the number of sampled traces.

**Weaknesses:**

- The forward-KL update is largely heuristic and does not come with formal guarantees on bias or convergence. While it stabilizes training in practice, the theoretical trade-offs and conditions under which the approximation holds are not fully analyzed.

- The geometric-mean modification of the importance ratio reduces variance for long reasoning traces but may introduce systematic bias. The paper does not characterize the direction or magnitude of this bias, nor discuss potential failure modes.

- The method relies on external verifiers to estimate correctness probabilities. The overall stability and performance depend on verifier quality; imperfect or incomplete verifiers may mislabel valid/invalid reasoning paths and influence learning dynamics.

**Questions:**

Please refer to the weaknesses section.

---

> ### Author Response · Authors · 2025-11-21
> **Rebuttal by Authors**
>
> Thank you for your supportive review and suggestions. Below we respond to the comments in **Weaknesses (W)**.
>
> ---
>
> ***W1: Analysis of forward KL.***
>
> The forward KL is not heuristic. The evidence lower bound (ELBO) holds regardless of the specific form of the variational distribution $q\_\\phi$.
>
> Both forward and reverse KL divergence share the same optimum: the true posterior distribution. We have provided a proof demonstrating that the posterior constitutes the optimal variance-minimizing behavior policy ($\\textrm{\\color{blue}Appendix A.8}$), which provides theoretical justification for employing forward KL divergence in our framework.
>
> ---
>
> ***W2: Bias of geometric-mean modification.***
>
> We acknowledge that the geometric mean modification applied to the ratio term in $\\rho\_k$ introduces a bias. The primary motivation for this modification is the unbounded nature of the likelihood ratio, which can range from zero to extremely large values. This numerical challenge is also encountered in the sequence-level importance weighting of GSPO [1], where geometric mean normalization is similarly employed.
>
> The geometric mean modification serves to reduce variance and mitigate length biases, as evidenced by the log-linear relationship between the unmodified ratio and token length shown in Figure 4 (left). In terms of bias direction, the unmodified $\\rho\_k$ empirically favors shorter reasoning traces, while the geometric mean variant alleviates this tendency. The magnitude of this bias depends on the token length distribution of reasoning traces for each specific question.
>
> Our ablation study ($\\textrm{\\color{blue}Table 16}$) demonstrates that despite introducing bias, the geometric mean modification improves performance, validating its practical utility.
>
> ---
>
> ***W3: The overall stability and performance depend on verifier quality.***
>
> Our framework does not fundamentally rely on a verifier, as it supports two variants for estimating $\\log \\pi\_\\theta(\\mathcal{Y}\_x|x,z)$: an accuracy-based estimator, which uses a verifier, and a geometric-mean-likelihood-based estimator, which does not require a verifier and achieves comparable performance. These variants are labeled “-Acc” and “-GML” in our main experiments, respectively.
>
> To study sensitivity to verifier quality, we introduce a “dummy verifier” that assigns $\\pi\_\\theta(\\mathcal{Y}\_x|x,z) = 0.5$ to all samples, simulating an extreme case of a completely inaccurate verifier. Results of this ablation are provided in $\\textrm{\\color{blue}Table 15}$. The results show that the performance degenerates slightly.

---

> > ### Comment · Reviewer_nKtU · 2025-11-28
> >
> > Thank you for the detailed responses. The clarifications and additional analyses address my questions. I have no further points to raise.

---

### Official Review · Reviewer_b5vB · 2025-11-04

**Soundness:** 3
**Presentation:** 3
**Contribution:** 3
**Rating:** 8
**Confidence:** 3

**Summary:**

The paper reframes reasoning in language models as a latent variable problem, where the intermediate “thinking trace” is treated as a hidden variable optimized through variational inference. Starting from the ELBO, the authors derive a tighter multi-trace objective and introduce a forward-KL formulation that stabilizes training of the variational posterior. This yields a unified probabilistic view that connects supervised finetuning, rejection sampling finetuning, and binary-reward RL as approximate inference objectives under the same framework. Empirically, the method consistently improves reasoning performance across math, code, and general benchmarks on Qwen models, showing both higher stability and generalization. Overall, it offers a principled bridge between likelihood-based reasoning and RL-style optimization.

**Strengths:**

The paper offers a clean and principled unification of reasoning-oriented training methods for LLMs under a single probabilistic framework. By treating reasoning traces as latent variables and optimizing an IWAE-style ELBO, it grounds both RL and supervised finetuning as approximate inference procedures, clarifying their implicit assumptions. The forward-KL formulation is a well-motivated and technically strong addition that improves stability without sacrificing generality. The paper is clearly written and empirically supported with competitive results across diverse reasoning domains. Overall, it provides both conceptual clarity and practical value, advancing the theoretical understanding of reasoning optimization in large models.

There is a nice connection to older works of RL for non-LLM domains: see https://arxiv.org/pdf/1805.00909. They also use latent optimization via ELBO.

**Weaknesses:**

I believe the authors haven't done a full literature search on prior RL works, such as https://arxiv.org/pdf/1805.00909, https://openreview.net/pdf?id=a147pIS2Co that discuss a similar idea in the non-LLM setting. Could the authors explain the differences to these works.

**Questions:**

How does the variational posterior q_{\phi} scale with model size? Is there evidence that a separate learned posterior remains necessary for very large models, or can it be amortized within the base model?

How sensitive is the proposed method to the quality or accuracy of the “hint” used in the variational posterior? Would the framework still hold when the hint is noisy, partially correct, or unavailable?

---

> ### Author Response · Authors · 2025-11-21
> **Rebuttal by Authors**
>
> Thank you for your supportive review and suggestions. Below we respond to the comments in **Weaknesses (W)** and **Questions (Q)**.
>
> ---
>
> ***W1: Missing literature on prior RL works [1,2].***
>
> Thank you for pointing out these references. We have included a discussion of related work in Appendix B, which already covers reference [2] in our original submission. The key distinction is that while this reference [2] employs MCMC to sample from the intractable posterior, we propose using forward KL divergence to fine-tune a variational posterior. Additionally, our method utilizes a tighter IWAE-style evidence lower bound.
>
> Regarding the tutorial [1], it begins from a reinforcement learning perspective (where the goal is to search for an optimal policy) and provides a view on how to construct a probabilistic graphical model (PGM), where the posterior (of actions) corresponds to an optimal solution derived by defining observations in the PGM based on rewards. Then this can be solved via variational inference. This differs from our approach.
>
> Our method, in contrast, starts from the objective of maximizing $\\log P\_\\theta(\\mathcal{Y}\_x \\mid x)$, treats the reasoning process as a discrete latent variable $z$, and optimizes the ELBO to train the reasoning model. We subsequently build connections between our framework and other RL algorithms (Section 3 and Appendix A). From an application perspective, we focus on the language domain and model reasoning traces as discrete latent variables, which also differs from the above work.
>
> We have included a discussion about these related works in the paper revision.
>
> **References:**\
> [1] Reinforcement Learning and Control as Probabilistic Inference: Tutorial and Review, 2018\
> [2] Training Chain-of-Thought via Latent-Variable Inference, NeurIPS 2023
>
>
> ---
>
> ***Q1: How does the variational posterior $q\_{\\phi}$ scale with model size? Is there evidence that a separate learned posterior remains necessary for very large models, or can it be amortized within the base model?***
>
> In all experiments, both the variational posterior $q\_\\phi$ and the reasoning model $\\pi\_\\theta$ are initialized from the same base model, sharing identical model sizes. We opted not to use amortization as it offers no expected advantage in our framework. Crucially, employing a separately learned posterior incurs minimal computational overhead: the variational posterior is only used during training (with much lower model loading frequency compared to typical RL methods) and is discarded during deployment. The only additional cost is disk storage for the variational posterior parameters, which is negligible given modern storage capabilities.
>
> ---
>
> ***Q2: How sensitive is the proposed method to the quality of the “hint” used in the variational posterior?***
>
> To validate this, we generate noisy hints by using an LLM (Qwen3-4B-Instruct-2507) to rewrite the original hints, introducing errors, and use these hints for sampling reasoning traces using the variational posterior. The results, shown in $\\textrm{\\color{blue}Table 14}$, confirm that providing noisy hints significantly degrades the quality of the sampled reasoning traces. This outcome validates our basic assumption, as it demonstrates the sensitivity of the posterior to its conditioning input, which is consistent with its theoretical role.

---

### Official Review · Reviewer_7MH7 · 2025-11-08

**Soundness:** 3
**Presentation:** 3
**Contribution:** 3
**Rating:** 4
**Confidence:** 4

**Summary:**

This paper proposes a framework for training reasoning models by treating thinking traces as latent variables and optimizing the log-likelihood of correct answers using variational inference (VI). The authors derive an IWAE-style multi-trace objective (ELBO) to update the reasoning model $\pi_{\theta}$ and a forward-KL objective (Eq. 9) to stabilize the training of the variational posterior $q_{\phi}$.

A key theoretical contribution is showing that existing methods like Rejection Sampling Finetuning (RFT) and GRPO can be interpreted as local forward-KL objectives, revealing an implicit bias toward easier questions. Empirically, the method demonstrates strong performance, outperforming baselines like Bespoke-Stratos on several reasoning benchmarks.

**Strengths:**

1. Novel and Insightful Theoretical Framework: The paper's primary strength is its principled unification of several reasoning-as-inference methods (SFT, RL) under a single probabilistic lens. Framing reasoning as a latent variable problem is elegant.

2. Valuable Unification and Bias Discovery: Showing that RFT and GRPO can be derived as forward-KL minimizations (Eq. 10 & 11) is a significant insight. The discovery of the implicit accuracy-weighting, and its resulting bias toward easier questions, is a valuable, non-obvious finding for the community that could influence future method design.

3. Strong Empirical Performance: The method achieves impressive and consistent gains across multiple model scales (4B to 32B). It surpasses strong baselines, for instance, by over 8.5% on average accuracy compared to Bespoke-Stratos-4B on the same training data (Table 1), demonstrating its effectiveness.

**Weaknesses:**

1. Extreme Methodological and Computational Complexity: The proposed training pipeline (Algorithm 1) is substantially more complex than the baselines. As detailed in Appendix C, the full process requires (at minimum) three separate, full finetuning runs ($\pi_{\theta_0}$, $q_{\phi}$, and $\pi_{\theta}$) plus a massive offline data generation and re-weighting pass. This re-weighting step (for the "-Acc" estimator) is itself a huge computational burden, requiring lots of verifications.

2. The final algorithm essentially boils down to two complex, weighted SFT steps. This raises the critical question: is the VI formalism necessary? The argument for its necessity is weakened by the authors' own admission that they must modify the "principled" importance weight (Eq. 8, using a geometric mean) to mitigate high variance, thereby introducing bias and moving away from the pure ELBO derivation. The paper is missing the important ablation: a comparison against a simpler weighted SFT (e.g., using only verifier accuracy as a weight) to prove that the complex $\pi_{\theta} / q_{\phi}$ importance-weighting is truly beneficial.

3. While Appendix C provides training details, the paper presents its strong empirical results without a clear accounting of the total compute. The performance gains are achieved with what is likely an orders-of-magnitude increase in total compute over the SFT baseline. Without a compute-matched baseline (e.g., SFT on more data or for more epochs), it is impossible to assess the actual efficiency of the method.

**Questions:**

The author mentions directly ELBO formulation with reverse KL will have mode collapse. How bad is it? Do you have small scale observations?

---

> ### Author Response · Authors · 2025-11-21
> **Rebuttal by Authors**
>
> Thank you for your valuable review and suggestions. Below we respond to the comments in **Weaknesses (W)** and **Questions (Q)**.
>
> ---
>
> ***W1: Extreme Methodological and Computational Complexity.***
>
>
> Our method is derived from a straightforward and well-motivated objective: maximizing $\\log P\_\\theta(\\mathcal{Y}\_x|x)$. It consists of two clear steps: learning a variational posterior and optimizing the evidence lower bound (ELBO). We have provided the code in the supplementary material. The implementation is simple.
>
> We acknowledge the higher computational cost compared to baselines. However, this cost is justified by the significant performance gains achieved using the same training data, demonstrating superior data efficiency. Furthermore, as shown in $\\textrm{\\color{blue}Table 17}$, even when the baseline's compute is scaled to exceed our method's GPU hours, our approach still outperforms, underscoring its effectiveness.
>
> ---
>
> ***W2: The necessity of VI formalism is weakened by its reliance on a modified importance weight (a geometric mean in Eq. 8) to reduce variance; The comparison against a simpler weighted SFT is missing.***
>
> The weight $\\rho\_k=\\frac{\\pi\_\\theta (z\_k|x)}{q\_\\phi(z\_k|x,y')} \\cdot \\pi\_\\theta(\\mathcal{Y}\_x|x,z\_k)$ is derived from an IWAE-style evidence lower bound (ELBO). Its two components have distinct interpretations: the likelihood ratio $\\frac{\\pi\_\\theta (z\_k|x)}{q\_\\phi(z\_k|x,y')}$ measures the likelihood ratio of the reasoning trace $z\_k$, while $\\pi\_\\theta(\\mathcal{Y}\_x|x,z\_k)$ represents the likelihood of arriving at the correct answer given $x$ and $z\_k$.
>
> The second term is bounded in $[0,1]$, but the first term might become large, especially for lengthy reasoning traces. This issue is also noted in GSPO [1]. To mitigate this, following GSPO, we adopt a similar geometric mean normalization. Although this modification introduces a slight bias within the variational inference framework, the terms retain their original semantic meaning. An ablation study on this geometric mean modification is provided in $\\textrm{\\color{blue}Table 16}$.
>
> To further analyze the role of the likelihood ratio, we conduct an ablation where the weight is set to $\\rho\_k=\\pi\_\\theta(\\mathcal{Y}\_x|x,z\_k)$, effectively removing the ratio term. Results in $\\textrm{\\color{blue}Table 13}$ confirm that including the likelihood ratio leads to better performance, justifying its importance in our framework.
>
> **References:**\
> [1] Group Sequence Policy Optimization, 2025
>
> ---
>
> ***W3: Comparison considering the total compute.***
>
> We acknowledge that our method incurs a higher computational overhead than the Bespoke-Stratos baseline, although both use identical training data. Despite this cost, our approach achieves significantly stronger performance. To ensure a fair comparison, we scaled the compute allocated to Bespoke-Stratos to exceed our method's total GPU hours. As shown in $\\textrm{\\color{blue}Table 17}$, our method maintains superior performance under a comparable computational budget.
>
> Furthermore, our method outperforms the General-Reasoner-4B baseline, which itself requires substantially more computational resources than our approach.
>
> Theoretical support for our method's efficacy is provided in $\\textrm{\\color{blue}Appendix A.8}$, where we prove that the variational posterior serves as the optimal variance-minimizing behavior policy.
>
> ---
>
> ***Q1: Optimizing the ELBO formulation with reverse KL will have mode collapse. How bad is it? Do you have small-scale observations?***
>
>
>
>
> To optimize the reverse KL divergence, it is necessary to sample reasoning traces $z$ from the variational posterior $q\_\\phi(z|x,y')$ and then compute the corresponding loss. However, when initialized from a base model, $q\_\\phi(z|x,y')$ tends to produce shortcut reasoning traces. Our pilot experiments confirmed this issue: sampled traces $z$ frequently exhibited answer leakage, meaning $z$ did not reflect natural reasoning but instead prematurely revealed the answer. Such low-quality traces lack genuine logical reasoning, causing the optimization to perform poorly and become trapped in a shortcut mode.
>
> We attempted this approach using Qwen3-4B-Base, but the model consistently generated shortcut reasoning traces and showed no sign of convergence. Due to the complete failure of this experiment, we did not include it in the paper.
>
> To overcome this limitation, we instead employ the forward KL divergence, which shares the same global optimum as the reverse KL divergence but exhibits more favorable training behavior. The key distinction is that optimizing the forward KL relies on sampling $z$ from the reasoning model $\\pi\_\\theta(z|x)$, which does not suffer from the shortcut bias. This leads to more stable and effective optimization.

---

### Official Review · Reviewer_xKMW · 2025-11-08

**Soundness:** 2
**Presentation:** 2
**Contribution:** 2
**Rating:** 6
**Confidence:** 4

**Summary:**

The paper proposes a variational inference framework for training LLM reasoning model by treating thinking traces as latent variables and maximizing evidence lower bound. Key innovations include an IWAE-style multi-trace objective that tightens the bound with multiple samples, as well as optimization via forward KL divergence to avoid mode collapse. Authors show existing methods like rejection sampling finetuning and GRPO can be treated as local forward KL objectives with implicit reweighting. Experiments on Qwen models show consistent improvements over baselines on math, coding and general reasoning benchmarks.

**Strengths:**

* Authors address the important topic of injecting reasoning into LLMs by providing a probabilistic perspective. Theorems and proofs are clear to understand. There are some parts of the method that require additional discussion (see weakness), but it is overall a well-polished work.
* The paper provides a novel perspective on RFT/GRPO and can have impact on future works.
* Experiments are well conducted on serveral benchmarks spanning different aspects of reasoning. Results show consistent gains across multiple model scales.

**Weaknesses:**

* The mode collapse issue mentioned in section 2.3 brings up an important research question for introducing latent reasoning in LLMs but may require additional elaboration. The proposed forward KL is claimed to prevent mode collapse, but weighted SFT still concentrate probability on high-accuracy traces. My understanding is that the model will still collapse to the "correct" modes while sacrificing diversity. I wonder if the authors can provide more insights and experiment results on this part.
* Authors mention that the method currently is applied to only single iteration, while standard variational EMs alternate until close to convergence. Is there any reason why T>1 wasn't specifically tested? Will we observe certain collapse or divergence, as commonly observed in experiments with EM algorithms? Additional results can improve the credibility especially given that the paper's claim focuses on mode collapse/ diverse reasoning.
* The paper is missing discussion on computational overhead from the method as compared to other baselines, especially with multiple rollouts and multiple answer samples per trace.
* Could the authors provide some insights on the scaling properties of the method, especially with different model sizes and different K?

**Questions:**

See questions in weakness part

---

> ### Author Response · Authors · 2025-11-21
> **Rebuttal by Authors**
>
> Thank you for your supportive review and suggestions. Below we respond to the comments in **Weaknesses (W)**.
>
> ---
>
> ***W1: Additional elaboration on the collapse issue.***
>
> The "collapse" issue mentioned in Section 2.3 refers to the *model collapsing into shortcut reasoning*. The original objective for the variational posterior is the reverse KL divergence: $\\mathbb{D}\_{KL}(q\_\\phi(z|x,y’) || p\_\\theta (z|x,\\mathcal{Y}\_x) )$.
>
> To optimize this reverse KL, we must sample $z$ from $q\_\\phi$ and then compute the loss. However, when initialized from a base model, $q\_\\phi(z|x,y’)$ tends to generate shortcut reasoning. Our pilot experiments confirmed this: the sampled $z$ often exhibits answer leakage. This means that upon inspection, $z$ does not resemble natural thinking; it appears as if the answer is provided before any reasoning occurs. Such $z$ is low-quality because it lacks true logical reasoning, causing the optimization to perform poorly and become stuck in the shortcut mode. Since this experiment totally failed, we did not report it in the paper.
>
> To tackle this, we use the forward KL divergence, which shares the same optimal solution as the reverse KL, though with different training dynamics. The key difference is that optimizing the forward KL relies on sampling $z$ from the trained reasoning model $p\_\\theta(z|x)$, which does not suffer from the shortcut phenomenon. This leads to more stable and effective optimization.
>
> ---
>
> ***W2: Is there any reason why T>1 wasn't specifically tested?***
>
> Our decision to use $T=1$ is based on two main reasons. First, this setting ensures a fair and direct comparison with the baseline methods. Second, even with $T=1$, our approach consistently outperforms existing baselines, thereby validating the effectiveness of the proposed framework. We therefore plan to explore the effect of $T > 1$ and conduct related analysis in future work.
>
>
> ---
>
> ***W3: Discussion on computational overhead.***
>
>
> We acknowledge that our method involves additional computational steps, including training both a variational posterior and a reasoning model, as well as sampling traces and forward passes for probability estimation. However, our empirical results demonstrate that it enables more effective learning from the same data.
>
> Additionally, we add a new experiment (in $\\textrm{\\color{blue}Table 17}$), where we scale up the compute of the Bespoke-Stratos baseline to slightly exceed our method's total GPU hours. The results confirm that our approach maintains a performance advantage under an equivalent computational budget, highlighting its superiority.
>
>
>
> ---
>
> ***W4: Insights on the scaling properties of the method, especially with different model sizes and different K.***
>
> Our main experiments demonstrate that our method consistently outperforms the baselines across four different model scales: Qwen3-4B/8B-Base and Qwen2.5-7B/32B-Instruct (Figures 1, 2, 7, and 3). This consistent improvement suggests the approach is broadly effective and scalable.
>
> We further analyze the impact of $K$ in our 1k-data ablation study (Figure 3). The results indicate that performance improves as $K$ increases, although the relationship is not a simple function due to the use of benchmark accuracy for evaluation (unlike the smoother scaling laws observed with training loss). This trend suggests that performance can be further improved with a larger $K$. While we expect gains to eventually plateau, our experiments have not reached this saturation point, indicating significant potential for improvement in variational reasoning. The performance gain from increasing $K$ is consistent with theoretical expectations, as it leads to a tighter IWAE-style lower bound.

---

### Author Response · Authors · 2025-11-21
**Summary of Paper Revision**

We thank all reviewers for their constructive feedback, and we have responded to each reviewer individually. We have also uploaded a **Paper Revision** including additional results:


- $\\textrm{\\color{blue}Appendix A.8}$: Proof of the variational posterior's optimality as a variance-minimizing behavior policy.

- $\\textrm{\\color{blue}Appendix B}$: Discussion about more related works.

- $\\textrm{\\color{blue}Table 13}$: Ablation study on the effects of different SFT weights for training the reasoning model.

- $\\textrm{\\color{blue}Table 14}$: Ablation study on the effects of the quality of hints used for sampling from the variational posterior.

- $\\textrm{\\color{blue}Table 15}$: Ablation study on the effects of the verifier's accuracy.

- $\\textrm{\\color{blue}Table 16}$: Ablation study on the effects of geometric mean modification of the likelihood ratio.

- $\\textrm{\\color{blue}Table 17}$: Performance comparison with the baseline under a similar computational budget.

---

### Meta-Review · Area_Chair_qNK3 · 2025-12-31

**Summary:**

In this paper, the authors propose a variational approach to enhance the reasoning capabilities of LLMs. All the reviewers value the proposed method and remark that the results are competitive with existing solutions. However, there are a few limitations, such as greater complexity and a single training round. It is unclear whether this method can be used for retraining, as the others typically are; repeated training to maintain the same computational complexity may reduce the gain reported in the paper. This paper is a truly borderline paper with an interesting and principle VI derivation with some limitations (higher complexity and repeated training) that might limit the applicability of the method in general.

**Reviewer Concerns:**

The reviewers were primarily concerned with the method's computational complexity and the feasibility of using multiple rounds of training.  The authors acknowledge the additional complexity and pointed out their better results. I do not know if the authors would have convinced the reviewers.

I still think the paper would be a good addition to the program because of the derived VI approach. However, it can be rejected if there is not space for it.

**Reviewer Scores:**

Would the reviewers have significantly increased the score? The authors' arguments were not particularly strong and did not address the core weaknesses. I would assume they would not. But still the paper is well-evaluated.

---

### Decision · Program_Chairs · 2026-01-26

Accept (Poster)